# ON THE COLLAPSE OF GENERATIVE PATHS: A CRITERION AND CORRECTION FOR DIFFUSION STEERING

## ABSTRACT

Inference-time steering enables pretrained diffusion/flow models to be adapted to new tasks without retraining. A widely used approach is the ratio-of-densities method, which defines a time-indexed target path by reweighting probability-density trajectories from multiple models with positive, or in some cases, negative exponents. This construction, however, harbors a critical and previously unformalized failure mode: Marginal Path Collapse, where intermediate densities become non-normalizable even though endpoints remain valid. Collapse arises systematically when composing heterogeneous models trained on different noise schedules or datasets, including a common setting in molecular design where de-novo, conformer, and pocket-conditioned models must be combined for tasks such as flexible-pose scaffold decoration. We provide a novel and complete solution for the problem. First, we derive a simple path existence criterion that predicts exactly when collapse occurs from noise schedules and exponents alone. Second, we introduce Adaptive path Correction with Exponents (ACE), which extends Feynman–Kac steering to time-varying exponents and guarantees a valid probability path. On a synthetic 2D benchmark and on flexible-pose scaffold decoration, ACE eliminates collapse and enables high-guidance compositional generation, improving distributional and docking metrics over constant-exponent baselines and even specialized task-specific scaffold decoration models. Our work turns ratio-of-densities steering with heterogeneous experts from an unstable heuristic into a reliable tool for controllable generation.

## 1 INTRODUCTION

Generative models based on stochastic interpolants, such as diffusion models (Ho et al., 2020; Song et al., 2021) and flow matching (Lipman et al., 2023), have become state-of-the-art systems for creation and scientific discovery Rombach et al. (2022); Schiff et al. (2025); Xie et al. (2024). These models learn to transform noise at $t = 0$ into complex data at $t = 1$ by following a learned probability path defined by ordinary or stochastic differential equations. A key to their practical success is *inference-time control*: the ability to steer pretrained models toward new goals without retraining. Concretely, we are interested in problems where one wishes to impose several constraints by reusing separate models for each constraint rather than retraining a single monolithic model. This increasingly involves composing multiple pretrained experts, making robust steering a central bottleneck for progress (Skreta et al., 2025b; Mark et al., 2025).

A widely used steering mechanism is the *ratio-of-densities* $p(x)^{\gamma_1}/q(x)^{\gamma_2}$, which reweights the probability landscape to favor modes of $p$ while suppressing those of $q$. This construction arises naturally in Bayesian model composition and admits many existing guidance techniques as special cases. For example, classifier-free guidance (Ho & Salimans, 2021) can be written as targeting a distribution proportional to $p(x \mid y)^{\gamma}/p(x)^{\gamma-1}$. In Appendix D.1, we show that reward-tilted sampling, product-of-experts conditioning, and contrastive decoding also fit a common ratio-of-densities template whose intermediate densities must remain normalizable along the flow. Reviewer gBQy, naNy

In settings where experts share the same noise schedule (e.g., CFG with jointly trained conditional/unconditional pairs), this approach is typically stable. However, many scientific applications involve composing *heterogeneous* models (i.e., trained on diverse datasets, modalities, and noise schedules) where modularity is highly advantageous for flexibility and cost-efficiency. For exam-

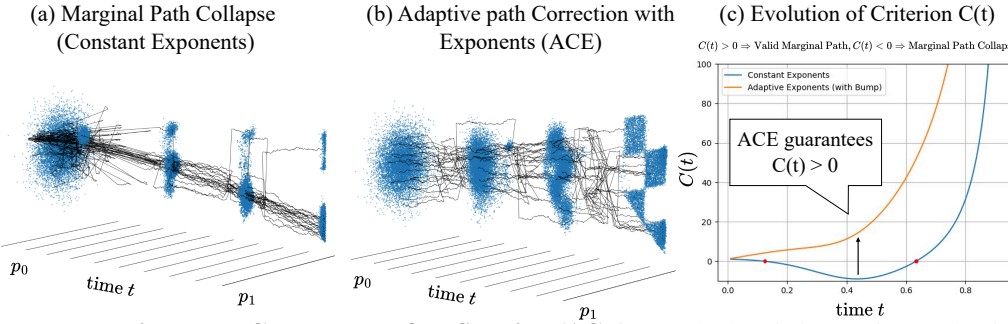

Figure 1: **Marginal Path Collapse and Our Solution (ACE).** (a) The heuristic steering path using constant exponents starts correctly but then diverges into nonsense towards the end. (b) Using the same pretrained models, ACE uses adaptive exponent schedules (Bump= $30$ visualized) to guarantee the path existence criterion $C(t) > 0$ for all $t$, enabling smooth transport to the target. (c) Graph of $C(t)$ for the constant schedule dips below zero, while ACE guarantees positivity for all time.

ple, in molecular design, *scaffold decoration* (Guan et al., 2009) starts from a known "scaffold" fragment that binds a protein and asks the model to generate side chains and poses that improve properties while preserving binding. This task naturally combines unconditional de-novo (DN) models (Hoogeboom et al., 2022a; Ketata et al., 2024), conformer (CONF) models for 3D geometry (Xu et al., 2022; Hassan et al., 2024), and pocket-conditioned structure-based drug-design (SBDD) models (Schneuing et al., 2024; Huang et al., 2024b) all trained on different datasets and noise schedules.

Moreover, a growing body of work on task-specific and adaptive noise scheduling (Vignac et al., 2023; Lee et al., 2024; Choi et al., 2025; Seo et al., 2025) and our own survey of DN/CONF/SBDD models (Appendix E.3, Tables E.2–E.4) show that different tasks and modalities require different allocations of noise across the trajectory, reinforcing that heterogeneous scheduler combinations are not only common but often *necessary* in practice.

Reviewer gBQy, naNy

**The Hidden Flaw: Uncovering Marginal Path Collapse.** Despite its apparent simplicity, ratio-of-densities steering can fail in heterogeneous settings. We identify a critical and frequent failure mode, *Marginal Path Collapse*, which occurs when the intermediate densities along the stochastic interpolant become non-normalizable and cease to exist, even though the endpoint distributions at $t = 0$ and $t = 1$ are perfectly valid (Figure. 1). Collapse arises from a mathematical imbalance between the steering exponents and the shrinking variance of the interpolant as $t \to 1$. When experts have different noise schedules or dimensionalities, their contributions contract at incompatible rates, causing the resulting ratio to grow rather than decay at infinity, violating normalizability. We show that this scenario is *systematic* when composing heterogeneous models trained with common schedules such as linear, cosine, or sigmoids, and, combined with the task-specific scheduler trends above, this implies that many realistic scientific workflows will routinely enter such invalid regions.

SDE/ODE samplers and Feynman–Kac correctors assume that the guided path defines a normalizable density at every timestep. When an intermediate density becomes non-normalizable, then its score ceases to exist. The sampler still evolves particles under a well-posed SDE/ODE, but the resulting flow transports a different density path rather than the intended one. Consequently, the terminal distribution at $t = 1$ no longer matches the target distribution.

Reviewer gBQy, naNy

**A Framework for Guaranteed Path Stability.** In this work, we provide a comprehensive solution to *Marginal Path Collapse*, transforming ratio-of-densities steering into a reliable and principled framework. Our contributions follow a clear "Diagnosis-and-Solution" pipeline:

1. **Diagnosis: Path Existence Criterion.** We derive a rigorous and easy-to-compute criterion that predicts exactly when collapse occurs from the noise schedules and exponents alone. This criterion explains why constant-exponent heuristics appear stable in shared-schedule settings yet fail under heterogeneous compositions.

2. **Solution: Adaptive Path Correction with Exponents (ACE).** Building on this diagnosis, we extend the Feynman–Kac PDE to support time-varying exponents. This allows ACE to dynamically adjust steering weights throughout generation to ensure the path existence criterion is satisfied at all times. ACE provably guarantees a valid probability path whenever the endpoints are valid.

**Validation.** On a synthetic 2D checkerboard benchmark, ACE eliminates collapse and reduces distributional error by over $5\times$. On flexible-pose scaffold decoration (Chen et al., 2025; Xie et al., 2024; Ghorbani et al., 2024), a practical molecular design task requiring heterogeneous DN/CONF/SBDD

Table 1: Comparison of inference-time control methodologies. Unlike heuristics (e.g., CFG) or FKC (Skreta et al., 2025a), our framework provides a principled criterion with adaptive correction (ACE), guaranteeing valid paths even under heterogeneous or time-varying settings.

| Methodology | Primary Goal | Handles Heterogeneity? | Handles Time-Varying / Negative Exponents? | Guarantees Path Existence? |
|---|---|---|---|---|
| Heuristic Guidance (e.g., CFG) | Improve sample quality | heuristically | heuristically | ✗ |
| Feynman–Kac Correctors (FKC) | Provide unbiased samples | ✗ | ✗ | ✗ |
| **Our Work (Criterion + ACE)** | Guarantee a valid, stable path | ✓ | ✓ | ✓ |

composition, standard steering fails at high guidance scales due to path collapse, whereas ACE remains stable and discovers molecules with substantially improved docking scores. On composi- tional image generation, we show that even when the path-existence criterion holds, time-varying exponents can sharpen intermediate distributions and improve sample quality in practice ($+9.57\%p$ on COCO-MIG), suggesting ACE as a general steering paradigm rather than only a fix for collapse.

Reviewer naNy

Our work provides a simple safeguard against a common failure mode and establishes the theoretical foundations for safely composing heterogeneous generative models, as summarized in Table 1.

## 2 METHOD

### 2.1 PRELIMINARIES: HETEROGENEOUS PRODUCT/RATIO-OF-DENSITIES

**Stochastic Interpolants and Probability Paths.** We begin by formalizing the notion of a probabil- ity path. A *probability path* (or *expert*) $\{q_t\}_{t\in[0,1]}$ on $\mathbb{R}^d$ is a family of densities with respect to the Lebesgue measure such that $q_t$ is normalizable to 1 for every $t$. In practice, paths are often gener- ated by a *stochastic interpolant* of the form $X_t = \alpha_t X_0 + \beta_t X_1$, $t \in [0, 1]$, where $X_0 \sim \mathcal{N}(0, I_d)$, $X_1 \sim q_1$ (target density), and $\alpha_t, \beta_t$ are nonnegative differentiable noise schedules satisfying bound- ary conditions $\alpha_0 = \beta_1 = 1, \alpha_1 = \beta_0 = 0$. The law of $X_t$ defines $q_t$. Stochastic interpolants can express both diffusion and flow matching by choosing proper $\alpha_t, \beta_t$ (Albergo et al., 2023).

**Heterogeneous Dimensionalities.** Given $n$ heterogeneous experts $\{q_t^{(i)}\}_{t\in[0,1]}$, $i = 1, \ldots, n$ each supported on $\mathbb{R}^{d_i}$ and exponents $\gamma_i : [0, 1] \to \mathbb{R}$, we embed them into a common ambient space $\mathbb{R}^d$ with $d = \max_{1 \le i \le n} d_i$, via coordinate projections and canonical embeddings. For each expert $i$, let $I_i \subset \{1, \ldots, d\}$ denote the coordinates it acts on. The projection $\pi_i : \mathbb{R}^d \to \mathbb{R}^{d_i}$ selects components in $I_i$, and the canonical embedding $\iota_i : \mathbb{R}^{d_i} \to \mathbb{R}^d$ inserts them back (zeroing the rest), satisfying $\pi_i \circ \iota_i = \mathrm{Id}_{\mathbb{R}^{d_i}}$. The lifted densities are $\tilde{q}_t^{(i)} := q_t^{(i)} \circ \pi_i$, and the heterogeneous *product/ratio-of-densities* is

$$h_t(x) := \prod_{i=1}^{n} \big(\tilde{q}_t^{(i)}(x)\big)^{\gamma_i(t)}, \qquad x \in \mathbb{R}^d. \tag{1}$$

We say that the family $\{h_t\}_{t\in[0,1]}$ has the *path existence property* if $h_t \in L^1(\mathbb{R}^d)$, $\forall t \in [0, 1]$. When so, $p_t^* = h_t/Z_t$ is the corresponding normalized probability path with $Z_t = \int_{\mathbb{R}^d} h_t(x)\, dx$.

**Heterogeneous Noise Schedules.** Throughout the paper, we will assume that each expert path $\{q_t^{(i)}\}_{t\in[0,1]}$ is generated by its own interpolant

$$X_t^{(i)} = \alpha_t^{(i)} X_0^{(i)} + \beta_t^{(i)} X_1^{(i)} \qquad t \in [0, 1] \tag{2}$$

where $X_0^{(i)} \sim q_0^{(i)} = \mathcal{N}(0, I_{d_i})$, $X_1^{(i)} \sim q_1^{(i)}$. When $\alpha_t^{(i)} \neq \alpha_t^{(j)}$ for some $i \neq j$, we refer to $h_t$ as having *heterogeneous noise schedules*.

**Path existence requirement.** SDE/ODE samplers and Feynman–Kac correctors assume that $p_t^* = h_t/Z_t$ exists at every timestep, requiring the $h_t$ to be integrable. If $h_t$ becomes non-normalizable (Marginal Path Collapse), then $p_t^*$ and its score cease to exist. The sampler still solves a well-posed ODE/SDE, but the resulting flow transports a different density path $\{p_t'\}$ rather than the intended $\{p_t^*\}$. The terminal distribution $p_1'$ produced by the sampler no longer matches the desired target $p_1^*$ (see Remark in Appendix A.2).

Reviewr gBQy, naNy

### 2.2 MARGINAL PATH COLLAPSE

Even if both endpoints $h_0$ and $h_1$ are integrable, *path-existence* is not guaranteed. A simple Gaussian example demonstrates the phenomenon (Figure 2).

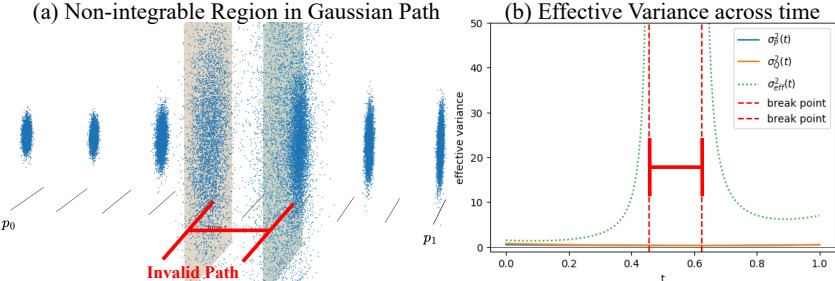

Figure 2: **Non-integrable Region in the ratio-of-Gaussians example (Eq. 3).** Although the ratio is well-defined at the endpoints, the intermediate variance explodes, causing *Marginal Path Collapse.*

**Why collapse occurs.** Let $h_t(x) = q_t^{(1)} q_t^{(2)} / q_t^{(3)} q_t^{(4)}$, where each component is a Gaussian path[1] under the linear interpolant $X_t = (1-t)X_0 + tX_1$:

$$q_t^{(1)} = \mathcal{N}\Big(0, \ ((1-t)^2 + \tfrac{1}{2}t^2)I\Big), \qquad q_t^{(2)} = \mathcal{N}\Big(0, \ ((1-t)^2 + 7t^2)I\Big),$$

$$q_t^{(3)} = \mathcal{N}\Big(0, \ (\tfrac{3}{2}(1-t)^2 + t^2)I\Big), \qquad q_t^{(4)} = \mathcal{N}\Big(0, \ (\tfrac{3}{2}(1-t)^2 + t^2)I\Big). \tag{3}$$

At $t = 0$ and $t = 1$, the ratio $h_t \propto \mathcal{N}(0, \sigma_{\text{eff}}^2(t)I)$ for some finite $\sigma_{\text{eff}}^2(t)$, making it integrable. However, by directly computing the ratio at $t = 0.5$, $h_t(x) \geq C \cdot \exp(+0.01\|x\|^2)$, for some constant $C > 0$, which is not integrable on $\mathbb{R}^d$. In fact, we can plot the probability path $p_t^*$ and its effective variance $\sigma_{\text{eff}}^2(t)$ across time as in Figure 2, showing that intermediate paths do not exist despite valid endpoints. We analyze the more general Gaussian mixture case in Appendix B.4.

The Gaussian example provides a crucial intuition: *Marginal Path Collapse* occurs when the variances of the numerator terms shrink "slower" than the variances of the denominator terms. This creates a temporary, fatal imbalance where the combined density becomes explosive rather than decaying at infinity. While this closed-form example is illustrative, most real-world models, especially those operating on complex data like molecules or images, involve non-Gaussian and compactly supported target distributions where such a direct variance calculation is impossible.

A natural and pressing question arises: can we find a general criterion that diagnoses the risk of collapse for these more complex cases, without needing a closed-form expression for the path?

## 2.3 Path Existence Criterion for Compactly Supported Densities

Compactly supported distributions are common in scientific applications, where data are bounded by physical constraints. In this setting, we can derive a clean criterion for path existence.

---

**Theorem 2.1** (Path Existence Test for Compactly Supported Densities). *For each $i = 1, \ldots, n$, let $\gamma_i(t), \alpha_t^{(i)}, \{q_t^{(i)}\}_{t \in [0,1]}, h_t(x)$ be as defined in Eq. 1 and 2. We only assume additionally that $q_1^{(i)}$ has compact support for all $i = 1, \ldots, n$.*

*If $h_1(x)$ is integrable and for every coordinate $k \in \{1, \ldots, d\}$ and all $t \in [0, 1)$,*

$$C_k(t) := \sum_{i: \, k \in I_i} \frac{\gamma_i(t)}{(\alpha_t^{(i)})^2} > 0, \tag{4}$$

*then $\{h_t\}_{t \in [0,1]}$ has the path existence property. Conversely, if there exists a cooridnate $k^* \in \{1, \ldots, d\}$ and $t^* \in [0, 1)$ such that $C_{k^*}(t^*) < 0$, then $\{h_t\}_{t \in [0,1]}$ is not integrable at $t^*$ (Marginal Path Collapse).*

---

We provide a proof in Appendix B.1. This theorem provides a tractable checklist: to certify path existence, one only needs to verify endpoint integrability and the positivity of the coefficients $C_k(t)$, which in practice can be checked on the discrete timesteps used by the sampler.

---

[1]A Gaussian path from $q_0 = \mathcal{N}(0, \sigma_1 I)$ to $q_1 = \mathcal{N}(0, \sigma_2 I)$ has the closed form expression $q_t = \mathcal{N}\big(0, (\sigma_1(1-t)^2 + \sigma_2 t^2)I\big)$

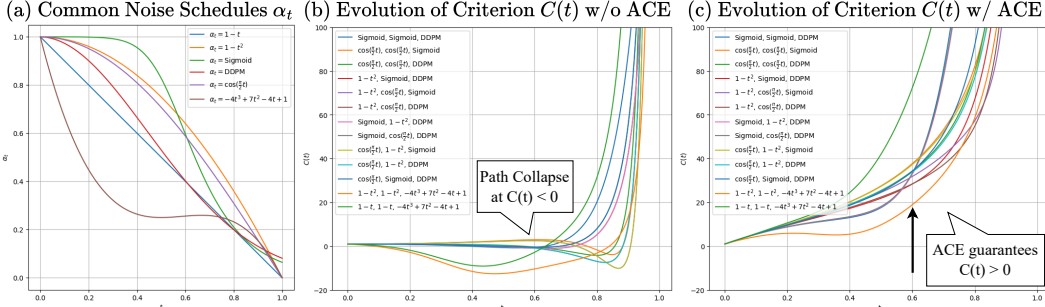

Figure 3: **Common noise schedules and Marginal Path Collapse.** (a) Representative noise schedules $\alpha_t$ used in modern diffusion and flow-matching models. (b) Path-existence criterion $C(t)$ (Eq. 4) for several heterogeneous three-expert compositions $h_t = q_t^{(1)} q_t^{(2)} / q_t^{(3)}$, formed from these schedules. Many combinations enter a region where $C(t) < 0$, implying non-normalizable intermediate densities. (c) Under ACE with a bump function (here $B = 50$), the corrected exponents ensure $C(t) > 0$ for all $t$, guaranteeing path existence.

Figure 3 illustrates the effect of this test on standard noise schedules. Panel (b) plots $C(t)$ for several heterogeneous compositions $h_t = q_t^{(1)} q_t^{(2)} / q_t^{(3)}$ built from linear, cosine, DDPM-style, and related schedules, revealing that many realistic combinations enter a region with $C(t) < 0$. This shows that widely used heuristic schedules can induce Marginal Path Collapse even when each individual expert is well behaved. Panel (c) shows that applying ACE with a bump function restores $C(t) > 0$ throughout the trajectory, converting these invalid heuristic paths into valid probability paths.

This discovery motivates the central goal of our work: to develop a method that can correct any given set of schedules to ensure the path existence criterion is always satisfied, thereby transforming unstable heuristics into a robust, guaranteed methodology. We introduce this method next.

### 2.4 Adaptive path Correction with Exponents: Bump Function Protocol

We develop our correction protocol, ACE, by first constructing a valid exponent schedule $\tilde{\gamma}_i(t)$ and then deriving the sampling dynamics that follow this corrected path.

In application scenarios, we need control over the initial distribution $p_0^* = \Pi_{i=1}^n (p_0^{(i)})^{\gamma_i(0)} / Z_0$ (usually fixed to $\mathcal{N}(0, I)$) and the target distribution $p_1^* = \Pi_{i=1}^n (p_1^{(i)})^{\gamma_i(1)} / Z_1$. However, we do not need to fix the intermediate marginals $p_t^*$, $t \in (0, 1)$. The idea is to choose an appropriate $\tilde{\gamma}_i(t)$ that preserves the original exponent values at the beginning and end, $\tilde{\gamma}_i(0) = \gamma_i(0), \tilde{\gamma}_i(1) = \gamma_i(1)$, while ensuring the intermediate densities are all normalizable.

---

**Theorem 2.2** (Adaptive Exponents with Bump Functions). *Let a set of noise schedules $\{\alpha_t^{(i)}\}$ and exponent boundary values $\{\gamma_i(0), \gamma_i(1)\}$ be given. Assume that the criterion $C_k(t)$ (Eq. 4) is positive at the boundaries. That is, $C_k(0) > 0$ and $\lim_{t \to 1-} C_k(t) > 0$ for all coordinates $k$. Then, there exists a set of differentiable functions $\{\tilde{\gamma}_i(t)\}$ such that $\tilde{\gamma}_i(0) = \gamma_i(0), \tilde{\gamma}_i(1) = \gamma_i(1)$ for all $i$, and the criterion $C_k(t) > 0$ is satisfied for all $k$ and $t \in [0, 1)$.*

---

We provide a constructive proof in Theorem B.2 showing that there always exists a positive constant $B > 0$ such that choosing one index $j$ and changing $\tilde{\gamma}_j(t) = \gamma_j(t) + Bt(1 - t)$ (adding a bump function) now satisfies the path existence criterion for all $t$. We call this the bump function protocol.

Now we have established "what" we are going to sample from. The next question is "how" to sample from the corrected probability path. The weighted SDE below is derived by extending the Feynman-Kac Correctors (FKC) (Skreta et al. (2025a), Alg. 2) to time-dependent exponents $\gamma_i(t)$.

Reviewer naNy,jeuE

For each $i = 1, \ldots, n$, let $\gamma_i(t) : \mathbb{R} \to \mathbb{R}$ be differentiable with respect to $t$ and $\{q_t^{(i)}\}_{t \in [0,1]}$ denote probability paths (defined in Eq. 1–2) associated with the stochastic differential equation (SDE):

$$X_0^{(i)} \sim q_0^{(i)}, \qquad dX_t^{(i)} = \left( v_t^{(i)}(X_t^{(i)}) + \frac{(\sigma_t^{(i)})^2}{2} \nabla \log q_t^{(i)}(X_t^{(i)}) \right) dt + \sigma_t^{(i)} dW_t^{(i)} \qquad (5)$$

where $W_t^{(i)}$ is a Wiener process in $\mathbb{R}^{d_i}$, $s_t^{(i)}(x^{(i)}) := \nabla_{x^{(i)}} \log q_t^{(i)}(x^{(i)})$, $x^{(i)} \in \mathbb{R}^{d_i}$ is the score function, and the fields $v_t^{(i)}, s_t^{(i)} \in \mathcal{C}^1$ are measurable. We embed vector fields $\tilde{v}_t^{(i)} : \mathbb{R}^{d_i} \to \mathbb{R}^{d_i}$ in $\mathbb{R}^d$ via $\tilde{v}_t^{(i)} = \iota_i \circ v_t^{(i)} \circ \pi_i : \mathbb{R}^d \to \mathbb{R}^d$. The following theorem establishes our sampling algorithm:

Reviewer jeuE

> **Theorem 2.3** (ACE: Adaptive path Correction with time-dependent Exponents). *If the path existence criterion (Eq. 4) holds, then for any differentiable vector field $v_t^* : \mathbb{R}^d \to \mathbb{R}^d$, the stochastic process given by the following weighted SDE/ODE with $s_t^*(X_t) := \sum_{i=1}^{n} \gamma_i(t)\tilde{s}_t^{(i)}(X_t), D_t^{(i)}(X_t) := -\nabla \cdot \tilde{v}_t^{(i)}(X_t) + (v_t^*(X_t) - \tilde{v}_t^{(i)}(X_t)) \cdot \tilde{s}_t^{(i)}(X_t)$:*
>
> $$dX_t = \left(v_t^*(X_t) + \frac{\sigma_t^2}{2} s_t^*(X_t)\right)dt + \sigma_t dW_t \tag{6}$$
>
> $$d\log\tilde{q}_t^{(i)}(X_t) = \left(D_t^{(i)} + \frac{\sigma_t^2}{2}\left(s_t^* \cdot \tilde{s}_t^{(i)} + \nabla \cdot \tilde{s}_t^{(i)}\right)\right)dt + \sigma_t\tilde{s}_t^{(i)} \cdot dW_t \tag{7}$$
>
> $$d\log w_t(X_t) = \left[\nabla \cdot v_t^*(X_t) + \sum_{i=1}^{n} \dot{\gamma}_i(t)\log\tilde{q}_t^{(i)}(X_t) + \sum_{i=1}^{n} \gamma_i(t)D_t^{(i)}(X_t)\right]dt \tag{8}$$
>
> *with initial values $X_0 \sim p_0^*, w_0 = \mathbf{1}, \log q_0^{(i)}(X_0)$ follows the probability path*
>
> $$p_t^*(x) = \frac{1}{Z_t}\prod_{i=1}^{n}\left(\tilde{q}_t^{(i)}(x)\right)^{\gamma_i(t)}, \ x \in \Omega_t$$
>
> *where $t \in [0,1]$, $\Omega_t := \mathrm{supp}(p_t^*)$ and $Z_t$ is the normalizing constant only dependent on $t$.*

We provide the full derivation and proof in Theorem A.1. Note we can minimize costly divergence computations when simulating the SDE in Theorem 2.3 by choosing $v_t^* = \sum_{i=1}^{n} \gamma_i(t)\tilde{v}_t^{(i)}$. We use this choice in our scaffold decoration experiment for better numeric stability and faster inference.

**Practical Remark.** Theorem 2.3 characterizes the weighted SDE whose marginal $p_t^*$ follows the corrected ratio-of-densities path. In practice, we implement this dynamics using a particle system with importance weights: at each step we propagate particles under the drift and diffusion in Eqs. 6–7, update their log-weights via Eq. 8, and trigger a resampling step whenever the effective sample size (ESS) falls below a threshold. During resampling, high-weight samples are duplicated and low-weight ones are removed in proportion to their weights. As illustrated in Figure 4, this procedure eliminates out-of-distribution trajectories in ACE, whereas the no-resampling heuristic (NR) leaves invalid samples in the batch. The complete algorithm for Theorem 2.3 is provided in Algorithm 1.

Reviewer jeuE

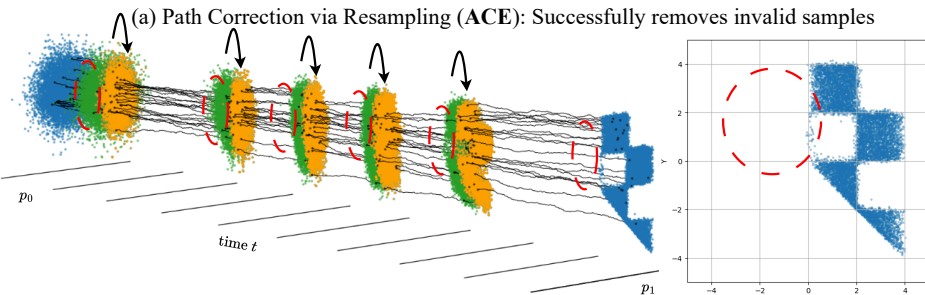

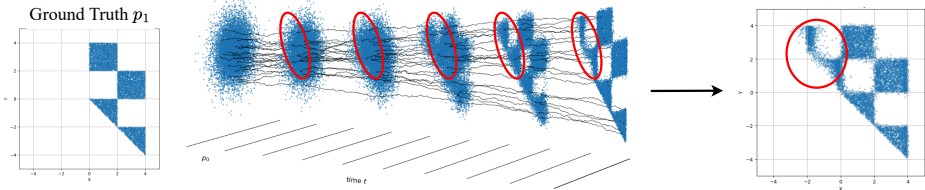

Figure 4: **Visualization of the sampling trajectories.** (a) ACE appropriately assigns weights to valid samples such that at each resampling step (green-to-orange), invalid samples are discarded. (b) No resampling (NR), a common heuristic (e.g., CFG), has no corrective mechanism that removes out-of-distribution samples.

## 2.5 APPLICATION: FLEXIBLE-POSE SCAFFOLD DECORATION

We now demonstrate the practical necessity of our framework on *flexible-pose scaffold decoration*, a task that illustrates all components of the heterogeneous ratio-of-densities setting developed so far. Here the goal is to generate molecules $\mathcal{M} = (\mathcal{M}^{\text{sc}}, \mathcal{M}^{\text{R}})$ that preserve a given scaffold bond topology $\mathcal{T}^{\text{sc}}$ and stably bind to a protein pocket $\mathcal{P}$, while allowing the scaffold's 3D pose to adapt within the pocket (Figure 5). This naturally yields the heterogeneous ratio-of-densities target

$$p(\mathcal{M} \mid \mathcal{T}^{\text{sc}}, \mathcal{P}) \underset{\text{Bayes}}{\propto} \frac{p(\mathcal{M}^{\text{sc}} \mid \mathcal{T}(\mathcal{M}^{\text{sc}}) = \mathcal{T}^{\text{sc}}) \, p(\mathcal{M} \mid \mathcal{M} \leftrightarrow \mathcal{P})}{p(\mathcal{M}^{\text{sc}})}, \tag{9}$$

where $\mathcal{M} \leftrightarrow \mathcal{P}$ denotes stable binding [2] and $\mathcal{T}(\cdot)$ extracts molecular topology (Appendix E.1).

Following classifier-free guidance, we introduce a guidance scale $\omega \geq 1$,

$$p_\omega(\mathcal{M}) \propto p(\mathcal{M}) \left( \frac{p(\mathcal{M} \mid \mathcal{T}^{\text{sc}}, \mathcal{P})}{p(\mathcal{M})} \right)^\omega, \tag{10}$$

so that larger $\omega$ enforces stronger scaffold and pocket conditioning. Crucially, this formulation (Eq. 9–10) decomposes into three pretrained diffusion experts:

- ($q^{(1)}$ and $q^{(2)}$) Unconditional de-novo model (DN) for $p(\mathcal{M}^{\text{sc}})$ and $p(\mathcal{M})$
- ($q^{(3)}$) Topology-conditioned conformer model (CONF) for $p(\mathcal{M}^{\text{sc}} \mid \mathcal{T}^{\text{sc}})$
- ($q^{(4)}$) Pocket-conditioned SBDD model for $p(\mathcal{M} \mid \mathcal{P})$

Writing the four corresponding factors with exponents $\gamma_1(t) = -\omega$, $\gamma_2(t) = -(\omega - 1)$, $\gamma_3(t) = \omega$, $\gamma_4(t) = \omega$ shows that Eq. 10 is exactly a *heterogeneous* ratio-of-densities composition of the form $p_t^* \propto \prod_i (q_t^{(i)})^{\gamma_i(t)}$ studied in Section 2.1.

Because existing DN, CONF, and SBDD experts are trained with *heterogeneous noise schedules* (see survey Tables E.2, E.3, and E.4), the constant-exponent path obtained from Eq. 10 violates the path-existence criterion (Theorem 2.1) for guidance scales $\omega > 1.1$, resulting in Marginal Path Collapse. Therefore, to raise $\omega$ while preserving path existence, we construct an *ACE-corrected* exponent schedule using Theorem 2.2. In practice, we apply a bump function of the form $B(t) = 30\,t(1-t)$ to one of the positive-exponent terms (here $\gamma_4$), which suffices to ensure $C_k(t) > 0$ for all $t$ and thereby guarantees a valid probability path. Finally, by simulating the importance-weighted SDE of Theorem 2.3, we obtain unbiased samples from the corrected path, and hence from the desired target distribution $p_\omega(\mathcal{M})$ in Eq. 10.

Reviewer gBQy, TtEF

Reviewer TtEF, naNy

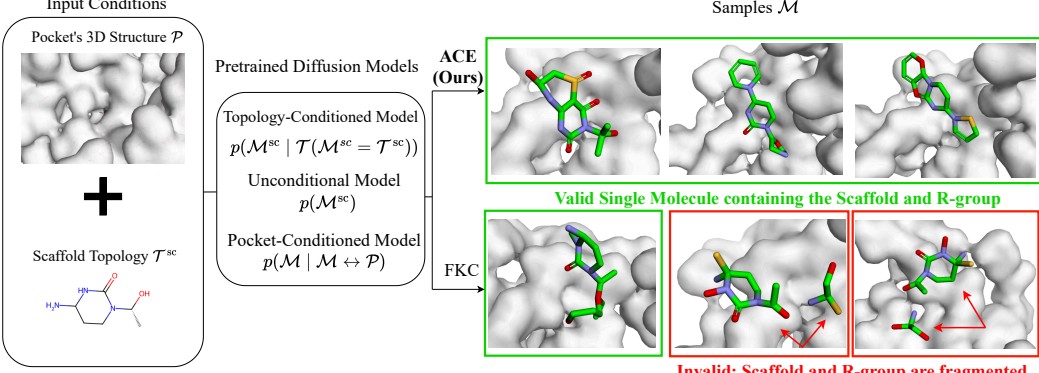

Figure 5: **Diffusion Steering Framework for Flexible-Pose Scaffold Decoration**. Qualitative results reveal that ACE, successfully modeling the ratio-of-density path, generates valid molecules containing the scaffold topology, while FKC (constant exponent baseline) generates invalid, fragmented molecules as a result of following an ill-defined probability path.

---

[2] Defined via docking energy $U^{\text{Dock}}(\mathcal{M} \leftrightarrow \mathcal{P}) < \tau^{\text{Dock}}$.

# 3 EXPERIMENTS

**Synthetic Checker Dataset.** We construct a synthetic benchmark that mirrors the heterogeneous conditioning structure encountered in our molecular task (Eq. 9). In both settings, one condition acts *locally* on a subset of variables (e.g., scaffold topology $A$ affecting only $\mathcal{M}^{\text{sc}}$), while another acts *globally* on the full configuration (e.g., pocket binding $B$ affecting $(\mathcal{M}^{\text{sc}}, \mathcal{M}^{\text{R}})$). As discussed in Section E.1 and illustrated in Figure E.2, this form of heterogeneous conditioning is pervasive in scientific problems such as scaffold decoration, linker generation, and protein–protein glue design.

To create the simplest possible analog of Eq. 9, we let $X$ and $Y$ be 1D variables having joint prior $(X, Y) \sim p_{\text{Checker}}$ and impose two constraints: $A = \{X \geq 0\}$ (local constraint on $X$) and $B = \{X + Y \geq 0\}$ (global constraint coupling $(X, Y)$). This induces the heterogeneous factorization

$$p(X, Y \mid A, B) \underset{\text{Bayes}}{\propto} p(X, Y, B \mid A) \underset{\text{Bayes}}{\propto} p(X \mid A)p(Y, B \mid A, X) \underset{(*)}{\propto} p(X \mid A)\frac{p(X, Y \mid B)}{p(X)} \tag{11}$$

For $(*)$, we use the fact that once $X$ is given, $A$ has no effect on $Y, B$, which can expressed by $A \perp Y, B \mid X$, implying $p(Y, B \mid A, X) = p(Y, B \mid X)$. Thus, exactly as in Eq. 9, the target distribution naturally decomposes into *three heterogeneous experts*: a 1D expert for $p(X \mid A)$, a 2D expert for $p(X, Y \mid B)$, and a 1D expert for $p(X)$.

To test the theoretical phenomenon identified by our path-existence criterion, we deliberately assign these experts a set of noise schedules drawn from Figure 3, many of which are known to induce *Marginal Path Collapse*. In Table 2, we use $\alpha_t^{(1)} = \alpha_t^{(2)} = 1 - t$, $\alpha_t^{(3)} = -4t^3 + 7t^2 - 4t + 1$. Importantly, the collapse behavior we observe here is *not* specific to this choice: Figures E.6–E.11 evaluate *realistic and widely used* schedules (DDPM, sigmoid, linear, cosine, polynomial) and show that the same failure patterns arise, confirming that the phenomenon is not an adversarial artifact. See implementation details and hyperparameter study in Appendix C.

**Flexible-Pose Scaffold Decoration.** We evaluate ACE on a realistic scientific task requiring heterogeneous DN/CONF/SBDD composition. Dataset details, implementation, and metrics appear in Appendix C.2. Briefly, we combine pretrained models EDM (Hoogeboom et al., 2022b) as DN, GeoDiff (Xu et al., 2022) as CONF, and DiffSBDD (Schneuing et al., 2024) as SBDD as defined in Section 2.5. We evaluate on subsets of CrossDock (Francoeur et al., 2020): CrossDock-Weak and CrossDock-SBDD. As diffusion-steering baselines we adopt NR and FKC (Skreta et al., 2025a), and as task-specific scaffold-decoration models we include Delete (Chen et al., 2025), DiffDec (Xie et al., 2024), and AutoFragDiff (Ghorbani et al., 2024).

## 3.1 RESULTS

We present the main quantitative results in Tables 2, 4, and 3. Across both synthetic and molecular settings, ACE achieves substantial improvements over NR and FKC, reflecting the benefits of enforcing the path-existence criterion and correcting heterogeneous ratio-of-density paths. We clarify that ACE differs from NR and FKC only by its adaptive exponent correction (with $B=30$ for all experiments); all other components and hyperparameters are shared across methods.

Table 2: Distributional similarity metrics (lower is better). For each metric, we report the minimum and mean $\pm$ standard deviation across 5 seeds. Best values are in **bold**. NR denotes no resampling, the common heuristic using only mixed scores. FKC refers to Feynman–Kac correctors, which fail when path-existence conditions are not satisfied. Path existence (O/X) is shown under Path Validity.

| Method | Path Validity | $W_1$ Min | $W_1$ Mean $\pm$ Std | $W_2$ Min | $W_2$ Mean $\pm$ Std | MMD (RBF) Min | MMD (RBF) Mean $\pm$ Std | Exponent Schedule |
|---|---|---|---|---|---|---|---|---|
| NR | X | 0.77 | $0.78 \pm 0.02$ | 1.06 | $1.07 \pm 0.02$ | 0.066 | $0.068 \pm 0.001$ | Constant |
| FKC | X | 2.09 | $2.13 \pm 0.04$ | 2.39 | $2.44 \pm 0.05$ | 1.07 | $1.43 \pm 0.31$ | |
| ACE ($B = 10$) | X | 1.47 | $2.02 \pm 0.65$ | 1.77 | $2.32 \pm 0.67$ | 0.44 | $0.78 \pm 0.23$ | Bump Function $Bt(1-t)$ |
| ACE ($B = 20$) | O | 0.13 | $0.52 \pm 0.77$ | 0.18 | $0.64 \pm 0.90$ | 0.009 | $0.12 \pm 0.24$ | |
| ACE ($B = 30$) | O | 0.24 | $\mathbf{0.28} \pm 0.036$ | 0.35 | $\mathbf{0.40} \pm 0.51$ | 0.019 | $\mathbf{0.027} \pm 0.0064$ | |
| ACE ($B = 40$) | O | 0.32 | $0.36 \pm 0.031$ | 0.44 | $0.475 \pm 0.025$ | 0.034 | $0.043 \pm 0.0099$ | |
| ACE ($B = 50$) | O | 0.30 | $0.40 \pm 0.07$ | 0.38 | $0.53 \pm 0.10$ | 0.034 | $0.052 \pm 0.01$ | |
| ACE ($B = 100$) | O | 0.43 | $0.52 \pm 0.07$ | 0.56 | $0.69 \pm 0.12$ | 0.070 | $0.080 \pm 0.011$ | |

Reviewer naNy

Reviewer naNy

Reviewer TtEF, naNy, gBQy

Reviewer naNy

Reviewer TtEF

Table 3: Comparison of NR, FKC, and ACE for CrossDock-Weak and CrossDock-SBDD. Higher is better for Validity and OSR; lower (more negative) is better for Vina scores. Best values are **bold**; second-best are underlined. FKC and ACE share all hyperparameters with the only difference being the addition of the bump function in the exponents of ACE. NR is FKC without resampling.

| Method | Path Validity | CrossDock-Weak Validity (%) | Vina Score (kcal/mol) Mean | Median | Top3 | Best | Worst | OSR (%) | CrossDock-SBDD Validity (%) | Vina Score (kcal/mol) Mean | Median | Top3 | Best | Worst | OSR (%) |
|---|---|---|---|---|---|---|---|---|---|---|---|---|---|---|---|
| NR ($w=1.0$) | O | 93.30 | -4.12 | -4.33 | -4.47 | -4.60 | -3.48 | 62.20 | 100.0 | -3.61 | -1.85 | -2.60 | -4.10 | -1.80 | 50.0 |
| NR ($w=1.1$) | X | 88.58 | -3.34 | -3.30 | -4.22 | -5.37 | -1.43 | 40.00 | 99.10 | -3.65 | -4.05 | -4.33 | -4.60 | -1.50 | 40.0 |
| NR ($w=1.2$) | X | 90.05 | -3.58 | -4.04 | -4.53 | -5.38 | -1.62 | 48.88 | 100.00 | -4.01 | -4.00 | -4.21 | -4.40 | -3.55 | 40.0 |
| NR ($w=1.3$) | X | 84.77 | -2.93 | -2.95 | -3.66 | -4.99 | -0.92 | 44.44 | 97.15 | -2.68 | -3.69 | -3.94 | -4.15 | 0.00 | 30.0 |
| NR ($w=1.4$) | X | 81.91 | -2.58 | -2.24 | -3.54 | -5.10 | -0.44 | 33.33 | 96.20 | -2.36 | -1.70 | -3.38 | -4.30 | -0.00 | 20.0 |
| FKC ($w=1.0$) | O | 93.30 | -4.12 | -4.33 | -4.47 | -4.60 | -3.48 | 62.20 | 100.0 | -2.28 | -1.85 | -2.60 | -4.10 | -1.80 | 50.0 |
| FKC ($w=1.1$) | X | 94.30 | -4.50 | -4.57 | -4.70 | -4.84 | -3.93 | 64.40 | 98.10 | -3.40 | -4.15 | -4.28 | -4.40 | -2.05 | 30.0 |
| FKC ($w=1.2$) | X | 86.70 | -3.46 | -3.42 | -3.81 | -4.50 | -2.51 | 57.80 | 98.10 | -3.04 | -3.70 | -3.85 | -4.05 | 0.00 | 40.0 |
| FKC ($w=1.3$) | X | 83.80 | -2.67 | -2.83 | -2.93 | -3.04 | -1.80 | 40.00 | 95.20 | -2.39 | -2.40 | -2.45 | -2.55 | -2.25 | 50.0 |
| FKC ($w=1.4$) | X | 78.10 | -2.20 | -2.48 | -2.59 | -2.74 | -1.33 | 28.90 | 100.0 | -4.16 | -4.10 | -4.25 | -4.45 | -4.00 | 50.0 |
| ACE ($w=1.0$) | O | 93.30 | -4.12 | -4.33 | -4.47 | -4.60 | -3.48 | 62.20 | 100.0 | -2.28 | -1.85 | -2.60 | -4.10 | -1.80 | 50.0 |
| ACE ($w=1.1$) | O | 96.70 | -5.12 | -4.92 | -5.33 | **-5.88** | -4.74 | 91.10 | 100.0 | _-4.88_ | _-4.90_ | _-4.93_ | _-5.00_ | _-4.70_ | **100.0** |
| ACE ($w=1.2$) | O | _98.60_ | _-5.44_ | _-5.67_ | _-5.75_ | -5.84 | _-4.59_ | _94.40_ | **100.0** | **-5.04** | **-5.05** | **-5.08** | **-5.15** | **-4.95** | _100.0_ |
| ACE ($w=1.3$) | O | **100.0** | **-5.72** | **-5.71** | **-5.78** | _-5.86_ | **-5.56** | _93.30_ | **100.0** | -4.68 | -4.75 | -4.78 | -4.85 | -4.35 | 90.0 |
| ACE ($w=1.4$) | O | 97.10 | -5.37 | -5.43 | -5.48 | -5.56 | -5.05 | 80.00 | 96.20 | -2.90 | -2.50 | -3.20 | -4.60 | -2.45 | 60.0 |

Table 4: Comparison on CrossDock-Weak and CrossDock-SBDD. Higher is better for Validity and OSR; lower (more negative) is better for Vina scores. Best values are **bold**; second-best are underlined. O indicates the method requires a reference scaffold pose; X indicates it does not.

| Method | Req. Ref. Pose | CrossDock-Weak Validity (%) | Vina Score (kcal/mol) Mean | Median | Top3 | Best | Worst | OSR (%) | CrossDock-SBDD Validity (%) | Vina Score (kcal/mol) Mean | Median | Top3 | Best | Worst | OSR (%) |
|---|---|---|---|---|---|---|---|---|---|---|---|---|---|---|---|
| Delete | O | 69.52 | -1.83 | -2.21 | -2.27 | -2.37 | -1.12 | 28.89 | 93.30 | _-4.98_ | _-4.95_ | **-5.23** | **-5.65** | _-4.55_ | **100.0** |
| DiffDec | O | 82.86 | -2.77 | -2.60 | -3.03 | -3.36 | -2.28 | 35.56 | 95.20 | -1.76 | -2.55 | -2.93 | -3.35 | 0.00 | 30.0 |
| AutoFragDiff | O | 98.10 | -4.82 | -5.02 | -5.43 | -5.83 | -3.38 | 73.33 | **100.0** | -4.61 | -4.60 | -4.95 | _-5.40_ | -3.95 | 60.0 |
| ACE ($w=1.2$) | X | _98.60_ | _-5.44_ | _-5.67_ | _-5.75_ | _-5.84_ | _-4.59_ | **94.40** | _100.0_ | **-5.04** | **-5.05** | _-5.08_ | -5.15 | **-4.95** | _100.0_ |
| ACE ($w=1.3$) | X | **100.0** | **-5.72** | **-5.71** | **-5.78** | **-5.86** | **-5.56** | _93.30_ | 100.0 | -4.68 | -4.75 | -4.78 | -4.85 | -4.35 | 90.0 |

# 4 DISCUSSION

**ACE Prevents Collapse at High Guidance Scales.** High guidance scales ($\omega > 1$) are crucial for generating high-quality molecules but are also where the risk of path collapse is highest. Our results make this trade-off explicit. As shown in Tables 3 and 4, ACE maintains near-perfect validity (96.7–100%) and improves docking scores from $-5.12$ ($\omega=1.1$) to $-5.72$ ($\omega=1.3$) as $\omega$ increases. In contrast, the FKC baseline, which uses a constant exponent, suffers catastrophic path collapse: its validity plummets to 78.1% and its docking score to $-2.20$ at $\omega=1.4$; qualitatively, in Fig. 5, the generated molecules are fragmented and chemically invalid. On CROSSDOCK-SBDD, ACE attains its best mean at $\omega=1.2$ ($-5.04$) with 100% validity and 100% OSR, whereas FKC exhibits weaker means and lower OSR (30–50%).

**Prevalence of Marginal Path Collapse.** In Appendix E.2, we evaluated all $5^3 = 125$ three-expert annealed compositions $h_t = q_t^{(1)} \big( q_t^{(2)}/q_t^{(3)} \big)^w$ formed from five standard noise schedules (DDPM (Ho et al., 2020), cosine (Nichol & Dhariwal, 2021), sigmoid (Xu et al., 2022), linear (Lipman et al., 2023), and polynomial (Hoogeboom et al., 2022a)). Excluding trivial homogeneous cases, among 100 heterogeneous schedule compositions the collapse rate increases sharply with guidance scale: 41% ($w=1.0$), 66% ($w=2.0$), 77% ($w=7.5$), and 80% ($w=15$). See Table E.5 and Figure E.5. Thus, collapse is a systematic and prevalent failure mode whenever experts differ in schedules. *(Reviewer TtEF, naNy, gBQy)*

**Impact of Path Collapse Duration.** Our empirical analysis suggests that the *presence* of Marginal Path Collapse is a more critical determinant of failure than its total duration: even short collapse intervals are sufficient to induce significant performance degradation. Figures E.6– E.11 show evaluation results and visualizations for schedule combinations exhibiting a wide range of collapse durations (from 7.5%, 9.0%, 10.9%, 11.4%, up to 48.8% of the full trajectory). While the precise error level varies due to secondary factors (e.g., magnitude of the criterion violation), the failure trend is consistent: NR suffers from inherent approximation errors regardless of path existence, whereas FKC performance degrades specifically because criterion violations destabilize the importance weights. In contrast, ACE reliably restores valid paths and consistently recovers the target distribution across all tested durations. *(Reviewer naNy)*

**Selection of the Bump Parameter $B$.** Since the path-existence criterion $C(t)$ depends solely on the analytical schedules, candidate values of $B$ can be pre-screened efficiently without expensive model evaluation. Empirically, we find that $B = 30$ consistently yields strong performance across *(Reviewer jeuE)*

all schedule combinations in the synthetic experiments (Fig. E.6) and in the scaffold-decoration task (Tables 4–3). The method also exhibits high stability: increasing $B$ further (e.g., $B = 100$) maintains performance far superior to NR and FKC, with only marginal degradation relative to the optimum (see Figure. 6). This reflects a trade-off: while increasing $B$ guarantees removal of path collapse, it also acts as a scalar multiplier on the guiding vector field, linearly amplifying inherent network approximation errors. Since larger $B$ incurs no additional computational cost, we recommend initializing with $B = 30$, which is generally sufficient to satisfy the criterion, and increasing it only if performance issues persist.

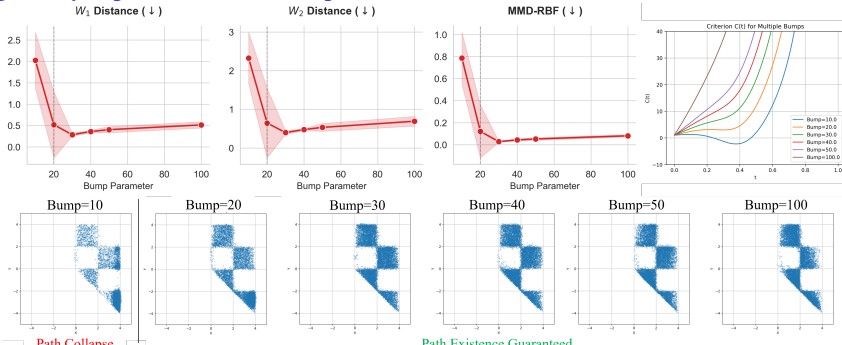

Figure 6: **Sensitivity to the bump parameter $B$.** Performance peaks near $B = 30$, but remains robust even at high values ($B = 100$), whereas criterion violations at $B = 10$ cause significant degradation. Shaded regions denote variance across 5 seeds.

**Steering Experts Outperforms Specialized Models.** Our scaffold-decoration experiments highlight the key advantage of our compositional framework: rather than training a monolithic, task-specific model, ACE achieves superior performance by flexibly combining existing DN/CONF/SBDD experts. Unlike specialized baselines that require a reference scaffold pose (Delete, DiffDec, AutoFragDiff), ACE requires no such input and still achieves stronger docking and robustness. On CROSSDOCK-WEAK, ACE attains a mean docking score of $-5.72$ ($\omega{=}1.3$), outperforming AutoFragDiff ($-4.82$) with higher validity (100% vs. 98.1%) and OSR (93.3% vs. 73.3%). On CROSSDOCK-SBDD, ACE reaches a mean of $-5.04$ ($\omega{=}1.2$) and 100% OSR, surpassing Delete ($-4.98$, 100% OSR) and AutoFragDiff ($-4.61$, 60% OSR).

**Related Work.** Diffusion models can be steered at inference time using ratio-of-densities methods such as classifier-free guidance (Ho & Salimans, 2021; Chung et al., 2025), but these lack guarantees that the sampling path remains valid. Feynman–Kac correctors (FKC) (Stoltz et al., 2010; Skreta et al., 2025a; Mark et al., 2025) provide principled foundations under restrictive assumptions (e.g., homogeneous models and constant exponents), leaving real-world scenarios vulnerable to Marginal Path Collapse. Our work introduces a path-existence criterion and the Adaptive path Correction with Exponents (ACE) framework, which extend FKC to heterogeneous pretrained models and time-varying exponents, enabling robust multi-expert composition for applications in molecular drug design (Guan et al., 2023; Schneuing et al., 2024; Gao et al., 2024). We provide a comprehensive review of related work in Appendix D.

**Broader Applications.** We further demonstrate on a compositional image generation benchmark in Appendix E.4 that even in homogeneous settings, where path existence already holds, ACE can yield additional gains over NR and FKC. We discuss limitations and further directions in Appendix F.

## 5 CONCLUSION

We identified *Marginal Path Collapse* as a fundamental failure mode in ratio-of-densities diffusion steering and provided ACE, a framework that provably prevents it. Our path existence criterion serves as a diagnostic tool to predict collapse, while our ACE framework first utilizing dynamic exponent scheduling provably prevents it by ensuring a valid probability path from noise to data. Empirically, ACE eliminates collapse, reduces distributional error by more than $5\times$ on benchmarks, and enables stable scaffold-based molecular design where existing methods fail. By transforming ratio-of-densities steering from an unstable heuristic into a theoretically grounded methodology, our work establishes both the foundations and the practical tools needed for robust inference-time control. We hope these contributions pave the way for reliable composition of heterogeneous generative models in both creative applications and high-stakes scientific domains.

**Use of Large Language Models.** Large language models (LLMs) were used to assist with writing, including grammar, style, and clarity improvements, and for organizational feedback on early drafts. LLMs were not used for generating original scientific content, designing experiments, or analyzing results. All technical contributions, experiments, and conclusions are the work of the authors.

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
