| | | Validity (%) | Vina Score (kcal/mol) | | | | | OSR (%) | Validity (%) | Vina Score (kcal/mol) | | | | | OSR (%) |
| | | | Mean | Median | Top3 | Best | Worst | | | Mean | Median | Top3 | Best | Worst | |
| NR ($w = 1.0$) | O | 93.30 | -4.12 | -4.33 | -4.47 | -4.60 | -3.48 | 62.20 | 100.0 | -3.61 | -1.85 | -2.60 | -4.10 | -1.80 | 50.0 |
| NR ($w = 1.1$) | X | 88.58 | -3.34 | -3.30 | -4.22 | -5.37 | -1.43 | 40.00 | 99.10 | -3.65 | -4.05 | -4.33 | -4.60 | -1.50 | 40.0 |
| NR ($w = 1.2$) | X | 90.05 | -3.58 | -4.04 | -4.53 | -5.38 | -1.62 | 48.88 | 100.00 | -4.01 | -4.00 | -4.21 | -4.40 | -3.55 | 40.0 |
| NR ($w = 1.3$) | X | 84.77 | -2.93 | -2.95 | -3.66 | -4.99 | -0.92 | 44.44 | 97.15 | -2.68 | -3.69 | -3.94 | -4.15 | 0.00 | 30.0 |
| NR ($w = 1.4$) | X | 81.91 | -2.58 | -2.24 | -3.54 | -5.10 | -0.44 | 33.33 | 96.20 | -2.36 | -1.70 | -3.38 | -4.30 | -0.00 | 20.0 |
| FKC ($w = 1.0$) | O | 93.30 | -4.12 | -4.33 | -4.47 | -4.60 | -3.48 | 62.20 | 100.0 | -2.28 | -1.85 | -2.60 | -4.10 | -1.80 | 50.0 |
| FKC ($w = 1.1$) | X | 94.30 | -4.50 | -4.57 | -4.70 | -4.84 | -3.93 | 64.40 | 98.10 | -3.40 | -4.15 | -4.28 | -4.40 | -2.05 | 30.0 |
| FKC ($w = 1.2$) | X | 86.70 | -3.46 | -3.42 | -3.81 | -4.50 | -2.51 | 57.80 | 98.10 | -3.04 | -3.70 | -3.85 | -4.05 | 0.00 | 40.0 |
| FKC ($w = 1.3$) | X | 83.80 | -2.67 | -2.83 | -2.93 | -3.04 | -1.80 | 40.00 | 95.20 | -2.39 | -2.40 | -2.45 | -2.55 | -2.25 | 50.0 |
| FKC ($w = 1.4$) | X | 78.10 | -2.20 | -2.48 | -2.59 | -2.74 | -1.33 | 28.90 | 100.0 | -4.16 | -4.10 | -4.25 | -4.45 | -4.00 | 50.0 |
| ACE ($w = 1.0$) | O | 93.30 | -4.12 | -4.33 | -4.47 | -4.60 | -3.48 | 62.20 | 100.0 | -2.28 | -1.85 | -2.60 | -4.10 | -1.80 | 50.0 |
| ACE ($w = 1.1$) | O | 96.70 | -5.12 | -4.92 | -5.33 | **-5.88** | -4.74 | 91.10 | 100.0 | _-4.88_ | _-4.90_ | _-4.93_ | _-5.00_ | _-4.70_ | **100.0** |
| ACE ($w = 1.2$) | O | _98.60_ | _-5.44_ | _-5.67_ | _-5.75_ | -5.84 | _-4.59_ | _94.40_ | **100.0** | **-5.04** | **-5.05** | **-5.08** | **-5.15** | **-4.95** | _100.0_ |
| ACE ($w = 1.3$) | O | **100.0** | **-5.72** | **-5.71** | **-5.78** | _-5.86_ | **-5.56** | _93.30_ | **100.0** | -4.68 | -4.75 | -4.78 | -4.85 | -4.35 | 90.0 |
| ACE ($w = 1.4$) | O | 97.10 | -5.37 | -5.43 | -5.48 | -5.56 | -5.05 | 80.00 | 96.20 | -2.90 | -2.50 | -3.20 | -4.60 | -2.45 | 60.0 |

Table 4: Comparison on CrossDock-Weak and CrossDock-SBDD. Higher is better for Validity and OSR; lower (more negative) is better for Vina scores. Best values are **bold**; second-best are underlined. O indicates the method requires a reference scaffold pose; X indicates it does not.

| Method | Req. Ref. Pose | CrossDock-Weak | | | | | | | CrossDock-SBDD | | | | | | |
| | | Validity (%) | Vina Score (kcal/mol) | | | | | OSR (%) | Validity (%) | Vina Score (kcal/mol) | | | | | OSR (%) |
| | | | Mean | Median | Top3 | Best | Worst | | | Mean | Median | Top3 | Best | Worst | |

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

# Appendix

The source code is available at `To_Be_Updated`.

# A THEORETICAL FOUNDATIONS OF ACE FOR HETEROGENEOUS RATIO-OF-DENSITIES

## A.1 MAIN THEOREMS AND PROOFS

**Definition A.1** (Ratio-of-Density Probability Path). Fix $t \in [0, 1]$ and let $\mu$ be a $\sigma$-finite reference measure on $\mathbb{R}^d$. For $i = 1, \ldots, n$, let $q_t^{(i)} : \mathbb{R}^{d_i} \to [0, \infty)$ be measurable densities and $\tilde{q}_t^{(i)} = q_t^{(i)} \circ \pi_i$ be their canonical lifts to $\mathbb{R}^d$. We define the **ratio-of-densities path** as the family of functions $\{p_t^*\}_{t \in [0,1]}$ given by:

$$p_t^*(x) := \frac{h_t(x)}{Z_t}, \quad \text{where} \quad h_t(x) = \prod_{i=1}^{n} \left( \tilde{q}_t^{(i)}(x) \right)^{\gamma_i}, \tag{A.1}$$

provided the following existence conditions are satisfied:

- **Support Inclusion:** Let $I_+ = \{i : \gamma_i > 0\}$ and $I_- = \{i : \gamma_i < 0\}$. Then $\text{supp}(\prod_{i \in I_+} \tilde{q}_t^{(i)}) \subseteq \text{supp}(\prod_{i \in I_-} \tilde{q}_t^{(i)}) =: \Omega_t$, preventing singularities on the boundary $\partial \Omega_t$.

- **Integrability:** The unnormalized product $h_t$ belongs to $L^1(\mu)$ with $0 < Z_t < \infty$.

*Remark.* Measurability of $h_t$ is guaranteed as the product of compositions of measurable functions with continuous projections. The integrability condition ensures $p_t^*$ is a valid Radon-Nikodym derivative $d\mathbb{P}_t^*/d\mu$.

**Proposition A.1** (Expectation identity under Feynman–Kac dynamics). *Let* $(p_t)_{t \in [0,T]}$ *be a family of probability densities on* $\mathbb{R}^d$ *evolving according to the weighted Feynman–Kac PDE*

$$\frac{\partial}{\partial t} p_t(x) = -\nabla \cdot \left( p_t(x) \, v_t(x) \right) + \frac{\sigma_t^2}{2} \, \Delta p_t(x) + p_t(x) \left( g_t(x) - \int g_t(y) \, p_t(y) \, dy \right), \quad \text{(A.2)}$$

*where* $v_t : \mathbb{R}^d \to \mathbb{R}^d$ *is a drift field,* $\sigma_t > 0$ *a scalar diffusion coefficient, and* $g_t : \mathbb{R}^d \to \mathbb{R}$ *a measurable weight function.*

*Consider the diffusion process*

$$dx_t = v_t(x_t) \, dt + \sigma_t \, dW_t, \qquad x_0 \sim p_0,$$

*driven by a standard Brownian motion* $(W_t)_{t \geq 0}$. *Define the weight process*

$$w_T = \exp\left( \int_0^T g_s(x_s) \, ds \right).$$

*Then for any bounded test function* $\phi : \mathbb{R}^d \to \mathbb{R}$,

$$\mathbb{E}_{p_T}[\phi(x_T)] = \frac{1}{Z_T} \mathbb{E}[w_T \, \phi(x_T)], \tag{A.3}$$

*where the expectation on the right is with respect to the law of the process* $(x_t)_{t \in [0,T]}$, *and* $Z_T > 0$ *is a normalizing constant for* $w_T$ *independent of* $x_T$.

*Proof.* The proof can be found in **Proposition A.1.** of Skreta et al. (2025a) or Section 4 of Stoltz et al. (2010). □

**Theorem A.1** (Adaptive path Correction with time-dependent Exponents (ACE)). *For each $i \in \{1, \ldots, n\}$, let*

$$\gamma_i(t) : \mathbb{R} \to \mathbb{R}$$

*be differentiable with respect to $t$ and $\{q_t^{(i)}\}_{t \in [0,1]}$ denote probability densities in $\mathbb{R}^{d_i}$ with respect to the Lebesgue measure. Suppose each probability path $\{q_t^{(i)}\}_{t \in [0,1]}$ is associated with the stochastic differential equation (SDE):*

$$X_0^{(i)} \sim q_0^{(i)}, \qquad dX_t^{(i)} = \left(v_t^{(i)}(X_t^{(i)}) + \frac{\sigma_t^{(i)2}}{2} \nabla \log q_t^{(i)}(X_t^{(i)})\right) dt + \sigma_t^{(i)} dW_t^{(i)} \qquad \text{(A.4)}$$

*or, equivalently, the ordinary differential equation (ODE):*

$$X_0^{(i)} \sim q_0^{(i)}, \qquad dX_t^{(i)} = v_t^{(i)}(X_t^{(i)}) dt \qquad \text{(A.5)}$$

*where $W_t^{(i)}$ is a Wiener process in $\mathbb{R}^{d_i}$, $s_t^{(i)}(x^{(i)}) := \nabla_{x^{(i)}} \log q_t^{(i)}(x^{(i)})$, $x^{(i)} \in \mathbb{R}^{d_i}$ is the score function, and the fields $v_t^{(i)}$, $s_t^{(i)} \in \mathcal{C}^1$ are measurable.*

*Let $d = \max_i d_i$, $\pi_i : \mathbb{R}^d \to \mathbb{R}^{d_i}$ be a linear projection map, and $\iota_i : \mathbb{R}^{d_i} \to \mathbb{R}^d$ be a linear embedding map such that $\pi_i \circ \iota_i = \mathrm{Id}_i$. For a vector field $f^{(i)} : \mathbb{R}^{d_i} \to \mathbb{R}^{d_i}$, define its canonical extension $\tilde{f}^{(i)} : \mathbb{R}^d \to \mathbb{R}^d$ by $\tilde{f}^{(i)} = \iota_i \circ f^{(i)} \circ \pi_i$. For $q_t^{(i)} : \mathbb{R}^{d_i} \to [0, \infty)$, define the canonical lift $\tilde{q}_t^{(i)} : \mathbb{R}^d \to [0, \infty)$ by $\tilde{q}_t^{(i)} = q^{(i)} \circ \pi_i$.*

*If the assumptions of definition A.1 hold, then, for any differentiable vector field $v_t^* : \mathbb{R}^d \to \mathbb{R}^d$, the stochastic process given by the following weighted SDE/ODE with $s_t^*(X_t) := \sum_{i=1}^n \gamma_i(t) \tilde{s}_t^{(i)}(X_t)$,*

$$X_0 \sim p_0^*, w_0 = \mathbf{1}, \log q_0^{(i)}(X_0) : \text{ standard gaussian density}$$

$$dX_t = \left(v_t^*(X_t) + \frac{\sigma_t^2}{2} s_t^*(X_t)\right) dt + \sigma_t dW_t \quad or \quad dX_t = v_t^*(X_t) dt$$

$$d \log \tilde{q}_t^{(i)}(X_t) = \left(-\nabla \cdot \tilde{v}_t^{(i)} + (v_t^* - \tilde{v}_t^{(i)}) \cdot \tilde{s}_t^{(i)} + \frac{\sigma_t^2}{2}\left(s_t^* \cdot \tilde{s}_t^{(i)} + \nabla \cdot \tilde{s}_t^{(i)}\right)\right) dt + \sigma_t \tilde{s}_t^{(i)} \cdot dW_t$$

$$or$$

$$d \log \tilde{q}_t^{(i)}(X_t) = \left[-\nabla \cdot \tilde{v}_t^{(i)}(X_t) + (v_t^*(X_t) - \tilde{v}_t^{(i)}(X_t)) \cdot \tilde{s}_t^{(i)}(X_t)\right] dt$$

$$d \log w_t(X_t) = \left[\nabla \cdot v_t^*(X_t) + \sum_{i=1}^n \dot{\gamma}_i(t) \log \tilde{q}_t^{(i)}(X_t)\right.$$

$$\left. + \sum_{i=1}^n \gamma_i(t)\left(-\nabla \cdot \tilde{v}_t^{(i)}(X_t) + \left(v_t^*(X_t) - \tilde{v}_t^{(i)}(X_t)\right) \cdot \tilde{s}_t^{(i)}(X_t)\right)\right] dt$$

*follows the probability path $p_t^*(x) = \frac{1}{Z_t} \prod_{i=1}^n \left(\tilde{q}_t^{(i)}(x)\right)^{\gamma_i(t)}$, $x \in \Omega_t$ where $t \in [0,1]$, $\Omega_t := \mathrm{supp}(p_t^*)$ and $Z_t = \int_{\Omega_t} \prod_{i=1}^n \left(\tilde{q}_t^{(i)}(x)\right)^{\gamma_i(t)} dx$ is the normalizing constant only dependent on $t$.*

*Proof.* From eq. (A.5) (or, equivalently, eq. (A.4)), each probability path $\{q_t^{(i)}\}_{t \in [0,1]}$ solves the Fokker-Planck PDE given by $\partial_t q_t^{(i)} = -\nabla \cdot (v_t^{(i)} q_t^{(i)})$. Dividing both sides by $q_t^{(i)}$, we get,

$$\partial_t \log q_t^{(i)} = -\nabla \cdot v_t^{(i)} - v_t^{(i)} \cdot s_t^{(i)} \qquad \text{(A.6)}$$

By the Leibniz product rule,

$$\frac{1}{Z_t} \partial_t \prod_{i=1}^n \left(\tilde{q}_t^{(i)}\right)^{\gamma_i(t)} = \frac{1}{Z_t} \partial_t \prod_{i=1}^n \exp\left(\gamma_i(t) \log \tilde{q}_t^{(i)}\right) = p_t^* \sum_{i=1}^n \left(\dot{\gamma}_i(t) \log \tilde{q}_t^{(i)} + \gamma_i(t) \partial_t \log \tilde{q}_t^{(i)}\right)$$

$$\text{(A.7)}$$

We derive the unified Feynman-Kac PDE for the heterogeneous product $\{p_t^*\}_{t\in[0,1]}$ by the following:

$$\partial_t \log p_t^* = \sum_{i=1}^{n} \left( \dot{\gamma}_i(t) \log \tilde{q}_t^{(i)} + \gamma_i \partial_t \log \tilde{q}_t^{(i)} \right) - \partial_t \log Z_t \tag{A.8}$$

$$\stackrel{(A.7)}{=} \sum_{i=1}^{n} \left( \dot{\gamma}_i(t) \log \tilde{q}_t^{(i)} + \gamma_i \partial_t \log \tilde{q}_t^{(i)} \right) - \int_{\Omega} p_t^* \sum_{i=1}^{n} \left( \dot{\gamma}_i(t) \log \tilde{q}_t^{(i)} + \gamma_i(t) \partial_t \log \tilde{q}_t^{(i)} \right) dx \tag{A.9}$$

$$= -\nabla \cdot v_t^* - v_t^* \cdot s_t^* \tag{A.10}$$

$$+ \underbrace{\nabla \cdot v_t^* + v_t^* \cdot s_t^* + \sum_{i=1}^{n} \left( \dot{\gamma}_i(t) \log \tilde{q}_t^{(i)} + \gamma_i \partial_t \log \tilde{q}_t^{(i)} \right)}_{=:g_t} \tag{A.11}$$

$$- \mathbb{E}_{p_t^*} \left[ \sum_{i=1}^{n} \left( \dot{\gamma}_i(t) \log \tilde{q}_t^{(i)} + \gamma_i(t) \partial_t \log \tilde{q}_t^{(i)} \right) \right] \tag{A.12}$$

$$= -\nabla \cdot v_t^* - v_t^* \cdot s_t^* + g_t - \mathbb{E}_{p_t^*}[g_t] \tag{A.13}$$

where we used the boundary assumption from definition A.1 and the divergence theorem:

$$\mathbb{E}_{p_t^*}[\nabla \cdot v_t^* + v_t^* \cdot s_t^*] = \int_{\Omega_t} (\nabla \cdot v_t^*(x) + v_t^*(x) \cdot s_t^*(x)) p_t^*(x) dx \tag{A.14}$$

$$= \int_{\Omega_t} \nabla \cdot (v_t^*(x) p_t^*(x)) dx = \int_{\partial\Omega_t} (v_t^*(x) \underbrace{p_t^*(x)}_{0 \text{ on } \partial\Omega_t}) \cdot d\mathbf{S} = 0 \tag{A.15}$$

Multiplying $p_t^*$ on both sides of eq. (A.13) yields:

$$\partial_t p_t^* = -\nabla \cdot (v_t^* p_t^*) + p_t^* \left( g_t - \mathbb{E}_{p_t^*}[g_t] \right) \tag{A.16}$$

where

$$g_t = \nabla \cdot v_t^* + v_t^* \cdot s_t^* + \sum_{i=1}^{n} \left( \dot{\gamma}_i(t) \log \tilde{q}_t^{(i)} + \gamma_i \partial_t \log \tilde{q}_t^{(i)} \right) \tag{A.17}$$

$$= \nabla \cdot v_t^* + v_t^* \cdot s_t^* + \sum_{i=1}^{n} \left( \dot{\gamma}_i(t) \log \tilde{q}_t^{(i)} + \gamma_i(-\nabla \cdot v_t^{(i)} - v_t^{(i)} \cdot s_t^{(i)}) \right) \tag{A.18}$$

$$= \nabla \cdot v_t^* + \sum_{i=1}^{n} \dot{\gamma}_i(t) \log \tilde{q}_t^{(i)} + \sum_{i=1}^{n} \gamma_i \left( -\nabla \cdot \tilde{v}_t^{(i)} + \left( v_t^* - \tilde{v}_t^{(i)} \right) \cdot \tilde{s}_t^{(i)} \right) \tag{A.19}$$

We can obtain the value of $\log \tilde{q}_t^{(i)}(X_t)$ by simulating an ODE (if $X_t$ follows an ODE) or an SDE (if $X_t$ follows an SDE).

$$\partial_t \log \tilde{q}_t^{(i)} = -\nabla \cdot \tilde{v}_t^{(i)} - \tilde{v}_t^{(i)} \cdot \tilde{s}_t^{(i)} \tag{A.20}$$

**ODE case:** $dX_t = v_t^*(X_t) dt$

$$\frac{d}{dt} \log \tilde{q}_t^{(i)}(X_t) = \partial_t \log \tilde{q}_t^{(i)}(x = X_t) + \nabla \log \tilde{q}_t^{(i)}(x = X_t) \cdot \frac{dX_t}{dt} \tag{A.21}$$

$$= -\nabla \cdot \tilde{v}_t^{(i)}(X_t) + (v_t^*(X_t) - \tilde{v}_t^{(i)}(X_t)) \cdot \tilde{s}_t^{(i)}(X_t) \tag{A.22}$$

**SDE case:** $dX_t = \left( v_t^*(X_t) + \frac{\sigma_t^2}{2} s_t^*(X_t) \right) dt + \sigma_t dW_t$

For the SDE case, we must use Itó's formula to find the differential $d \log \tilde{q}_t^{(i)}(X_t)$. Given the function $f(t, x)$ and the SDE for $X_t$ above, Itó's formula states that

$$df(t, X_t) = \left( \partial_t f + \mu_t \cdot \nabla_X f + \frac{1}{2} \text{Tr}(\Sigma_t^\mathsf{T} \nabla_X^2 f \Sigma_t) \right) dt + (\nabla_X f)^\mathsf{T} \Sigma_t dW_t,$$

where $f(t, X_t) = \log \tilde{q}_t^{(i)}(X_t)$, $\mu_t = v_t^* + \frac{\sigma_t^2}{2} s_t^*$, and $\Sigma_t = \sigma_t I$.

Applying that $\nabla_X f = \nabla_X \log \tilde{q}_t^{(i)} = \tilde{s}_t^{(i)}$, $\frac{1}{2} \operatorname{Tr}(\Sigma_t^\intercal \nabla_X^2 f \Sigma_t) = \frac{\sigma_t^2}{2} \Delta_X f = \frac{\sigma_t^2}{2} \nabla \cdot \tilde{s}_t^{(i)}$:

$$d \log \tilde{q}_t^{(i)}(X_t) = \left( -\nabla \cdot \tilde{v}_t^{(i)} + (v_t^* - \tilde{v}_t^{(i)}) \cdot \tilde{s}_t^{(i)} + \frac{\sigma_t^2}{2} \left( s_t^* \cdot \tilde{s}_t^{(i)} + \nabla \cdot \tilde{s}_t^{(i)} \right) \right) dt + \sigma_t \tilde{s}_t^{(i)} \cdot dW_t$$

$$(A.23)$$

By Proposition A.1, we can sample from $p_t^*$ by simulating the following weighted SDE or ODE starting from $X_0 \sim p_0^*$, $w_0 = \mathbf{1}$:

$$\text{SDE:} \quad dX_t = \left( v_t^*(X_t) + \frac{\sigma_t^2}{2} s_t^*(X_t) \right) dt + \sigma_t dW_t, \qquad d \log w_t = g_t(X_t) dt \qquad (A.24)$$

$$\text{ODE:} \quad dX_t = v_t^*(X_t) dt, \qquad\qquad\qquad\qquad d \log w_t = g_t(X_t) dt \qquad (A.25)$$

Our result extends, subsumes, and unifies prior formulations (Skreta et al. (2025a); Mark et al. (2025)) in the literature. $\qquad\square$

*Remark.* **Interpretation.** The auxiliary weight process $(w_t)$ plays the role of a *likelihood ratio corrector*: it accounts for the discrepancy between the law induced by the forward dynamics $(X_t)$ and the target density $p_t^*$. In other words, although the marginal of $X_t$ alone may not coincide with $p_t^*$, the pair $(X_t, w_t)$ ensures unbiased recovery of expectations under $p_t^*$ via Proposition A.1. This extends and unifies earlier formulations of weighted Feynman–Kac dynamics in the literature (Skreta et al., 2025a; Mark et al., 2025).

*Remark.* **Practical simulation.** From an algorithmic perspective, the weighted SDE/ODE requires no additional training or architecture-specific modifications. Practitioners only need to simulate sample paths $(X_t)$ according to the chosen dynamics and accumulate weights via the exponential update $d \log w_t = g_t(X_t) \, dt$. In practice, this can be carried out efficiently with Sequential Monte Carlo (SMC) or particle filtering methods, where the weights $w_t$ play the usual role of importance weights.

Although the heterogeneous Feynman–Kac framework (Theorem A.1) may appear abstract, its input–output structure is remarkably simple: given the forward dynamics and score functions $\{v_t^{(i)}, s_t^{(i)}\}_i$, one can simulate particles $(X_t, w_t)$ and obtain unbiased estimators for expectations under $p_t^*$. This makes the method broadly applicable without architectural constraints such as attention-map control or model-specific fine-tuning.

## A.2 ALGORITHMIC FORMULATIONS AND COMPARISON: FKC VS. ACE

Here, we present the algorithmic tables for FKC and ACE. As the tables indicate, FKC arises as a special case of ACE when the gamma schedule is constant. This results in an extra update for $\log q$ components, which is needed because $\dot{\gamma}_i(t) \neq 0$.

Reviewer jeuE, naNy

---

**Algorithm 1** Adaptive Correction with Exponents (ACE, Ours)

---

**Require:**
- Batch size $N$, initial particles $X_0^j \sim p_0^*$, weights $w_0^j = 1/N$
- Networks: scores $s_{\theta_i}^{(i)}$ and velocities $v_{\theta_i}^{(i)}$ for the paths $q_t^{(i)}$
- Projection $\pi_i$, embedding $\iota_i$, **time-varying exponents** $\gamma_i(t)$
- Base drift $v_\phi^*$, noise schedule $\sigma_t$, steps $T$, resampling threshold $\tau$

1: $\Delta t \leftarrow 1/T$
2: **for** $t = 0, \Delta t, \ldots, 1 - \Delta t$ **do**
3:     **Mixture score:** $s_t^*(x) = \sum_{i=1}^n \gamma_i(t)\, \tilde{s}_t^{(i)}(x)$ with $\tilde{s}_t^{(i)}(x) = \iota_i(s_{\theta_i}^{(i)}(\pi_i(x), t))$.
4:     **Drift:** $\mu_t(x) = v_\phi^*(x, t) + \frac{\sigma_t^2}{2} s_t^*(x)$
5:     **Component drift correction** with $\tilde{v}_t^{(i)}(x) = \iota_i(v_{\theta_i}^{(i)}(\pi_i(x), t))$:

$$D_t^{(i)}(x) = -\nabla \cdot \tilde{v}_t^{(i)}(x) + (v_\phi^*(x, t) - \tilde{v}_t^{(i)}(x)) \cdot \tilde{s}_t^{(i)}(x),$$

6:     **for** $j = 1, \ldots, N$ **do**
7:         **Propagate particle:** $X_{t+\Delta t}^j = X_t^j + \mu_t(X_t^j)\Delta t + \sigma_t \sqrt{\Delta t}\, \xi_t^j, \qquad \xi_t^j \sim \mathcal{N}(0, I)$
8:         **Update log-components:** $\log q_{t+\Delta t}^{(i),j} = \log q_t^{(i),j} + \Delta \log q_t^{(i),j}\, \Delta t$ with

$$\Delta \log q_t^{(i),j} = D_t^{(i)}(X_t^j) + \frac{\sigma_t^2}{2}\left(s_t^* \cdot \tilde{s}_t^{(i)} + \nabla \cdot \tilde{s}_t^{(i)}\right) + \sigma_t\, \tilde{s}_t^{(i)} \cdot \xi_t^j \sqrt{\Delta t}$$

9:         **Update weight:** $\log w_{t+\Delta t}^j = \log w_t^j + \Delta \log w_t^j\, \Delta t$ with

$$\Delta \log w_t^j = \nabla \cdot v_\phi^*(X_t^j, t) + \sum_{i=1}^n \dot{\gamma}_i(t)\, \log q_t^{(i),j} + \sum_{i=1}^n \gamma_i(t)\, D_t^{(i)}(X_t^j)$$

10:     **end for**
11:     Compute Effective Sample Size (ESS) $= \dfrac{(\sum_j w_{t+\Delta t}^j)^2}{\sum_j (w_{t+\Delta t}^j)^2}$
12:     **if** ESS $< \tau N$ **then**
13:         Resample particles according to $\{\frac{w_{t+\Delta t}^j}{\sum_j w_{t+\Delta t}^j}\}$
14:         Reset $w_{t+\Delta t}^j = 1/N$
15:     **end if**
16: **end for**

---

---

**Algorithm 2** Feynman–Kac Corrector (FKC, (Skreta et al., 2025a))

---

**Require:**

- Batch size $N$, initial particles $X_0^j \sim p_0^*$, weights $w_0^j = 1/N$
- Pretrained component networks: scores $s_{\theta_i}^{(i)}$ and velocities $v_{\theta_i}^{(i)}$ for $q_t^{(i)}$
- Projections $\pi_i$, embeddings $\iota_i$, constant exponents $\gamma_i$
- Base drift $v_\phi^*(x, t)$, noise schedule $\sigma_t$, total steps $T$, resampling threshold $\tau$

1: $\Delta t \leftarrow 1/T$
2: **for** $t = 0, \Delta t, \ldots, 1 - \Delta t$ **do**
3:     **Mixture score:** $s_t^*(x) = \sum_{i=1}^n \gamma_i \, \tilde{s}_t^{(i)}(x)$ with $\tilde{s}_t^{(i)}(x) = \iota_i(s_{\theta_i}^{(i)}(\pi_i(x), t))$.
4:     **Drift:** $\mu_t(x) = v_\phi^*(x, t) + \frac{\sigma_t^2}{2} s_t^*(x)$
5:     **Component drift correction** with $\tilde{v}_t^{(i)}(x) = \iota_i(v_{\theta_i}^{(i)}(\pi_i(x), t))$:

$$D_t^{(i)}(x) = -\nabla \cdot \tilde{v}_t^{(i)}(x) + (v_\phi^*(x, t) - \tilde{v}_t^{(i)}(x)) \cdot \tilde{s}_t^{(i)}(x),$$

6:     **for** $j = 1, \ldots, N$ **do**
7:         **Propagate particle:** $X_{t+\Delta t}^j = X_t^j + \mu_t(X_t^j)\Delta t + \sigma_t \sqrt{\Delta t}\, \xi_t^j, \qquad \xi_t^j \sim \mathcal{N}(0, I)$
8:         **Update weight:** $\log w_{t+\Delta t}^j = \log w_t^j + \Delta \log w_t^j \, \Delta t$ with

$$\Delta \log w_t^j = \nabla \cdot v_\phi^*(X_t^j, t) + \sum_{i=1}^n \gamma_i D_t^{(i)}(X_t^j)$$

9:     **end for**
10:    Compute Effective Sample Size (ESS) $= \dfrac{(\sum_j w_{t+\Delta t}^j)^2}{\sum_j (w_{t+\Delta t}^j)^2}$
11:     **if** ESS $< \tau N$ **then**
12:         Resample particles according to $\{ \frac{w_{t+\Delta t}^j}{\sum_j w_{t+\Delta t}^j} \}$
13:         Reset $w_{t+\Delta t}^j = 1/N$
14:     **end if**
15: **end for**

---

*Remark.* **Remark on computability vs. validity.** The FKC algorithm remains numerically computable even when Marginal Path Collapse occurs: the mixed score $s_t^*$ and the update rules in Algorithm 2 produce finite values at every step. However, this computability does *not* imply that the algorithm is sampling from a valid probability model. FKC is theoretically justified only when the target path $p_t^*(x) \propto h_t(x)$ exists as a family of *normalizable* densities for all $t$ on the discretization grid. If for some $t^*$ the ratio-of-densities integrand $h_{t^*}$ fails to lie in $L^1$ (i.e. $Z_{t^*} = \int h_{t^*} = \infty$), then $p_{t^*}^*$ does not exist, and the drift field $s_{t^*}^*$ used by FKC is no longer the score of any probability density. As a result, the reverse SDE/ODE and the weighted Feynman–Kac dynamics no longer transport the intended ratio-of-densities path $\{p_t^*\}$: instead they follow a different density path $\{p_t'\}$ determined by their coefficients. **Even though the sampler remains numerically well-defined, its terminal law $p_t'$ is no longer equal to the desired target $p_1^*$.**

Under standard regularity assumptions on the drift and diffusion, the Fokker–Planck equation associated with a given SDE has a unique weak solution for each initial probability density. Therefore, if there exists $t_c$ with $h_{t_c} \notin L^1$ (Marginal Path Collapse), there is no probability path $\{p_t^*\}$ that both (i) coincides with $h_t/Z_t$ whenever $h_t \in L^1$ and (ii) solves the same Fokker–Planck equation globally. The path produced by FKC, $\{p_t'\}$, is hence a different solution induced by its own initial condition and cannot coincide with the intended ratio-of-densities path.

Empirically, in every heterogeneous setting we tested, violation of the path-existence criterion (Theorem 2.1) leads FKC to complete failure (typically through unstable or degenerate importance weights) despite the algorithm itself producing finite updates. This behavior is exactly the Marginal Path Collapse phenomenon. Our path-existence criterion characterizes when $Z_t < \infty$ holds, and Theorem 2.2 shows how ACE constructs corrected exponent schedules ensuring that the entire path $\{p_t^*\}_{t \in [0,1]}$ is well-defined, allowing the weighted SDE established in Theorem 2.3 and Algorithm 1 to provide unbiased samples from the desired target.

# B  INTEGRABILITY PRESERVATION ALONG STOCHASTIC PATHS

The problem of preserving integrability for ratios of products of densities along stochastic paths is non-trivial. While general ratios can exhibit pathological behavior where integrability is lost, imposing a structural condition of component-wise dominance for GMMs, or positivity of a simple criterion for compactly supported densities, is sufficient to prevent such failures. This section demonstrates that under these conditions, integrability, once established at endpoints, is maintained throughout the evolution.

## B.1  MATHEMATICAL PRELIMINARIES

**Definition B.1** (Generative Stochastic Path for Stochastic Interpolants). Let $X_0 \sim \mathcal{N}(0, \mathbf{I})$ and $X_1 \sim p_1(x)$ be independent random variables. A **generative stochastic path** is a time-indexed random variable $X_t$ for $t \in [0, 1]$ defined by the sample-wise interpolation:

$$X_t = \alpha_t X_0 + \beta_t X_1 \tag{B.1}$$

where $\alpha_t, \beta_t$ are non-negative, differentiable functions of $t$ satisfying the boundary conditions $\alpha_0 = 1, \beta_0 = 0$ and $\alpha_1 = 0, \beta_1 = 1$.

*Remark* (Ornstein-Uhlenbeck Process and Flow Matching as Special Cases of Stochastic Interpolants). The generalized path encompasses the two most common paths in generative modeling.

- **Flow Matching (Linear Path):** Setting $\alpha_t = 1 - t$ and $\beta_t = t$ gives the linear interpolation path $X_t = (1 - t)X_0 + tX_1$.

- **Diffusion Models (OU-like Path):** Setting $\alpha_t = e^{-\int_0^t \gamma(s)ds}$ and $\beta_t = \sqrt{1 - e^{-2\int_0^t \gamma(s)ds}}$ corresponds to the path generated by an Ornstein-Uhlenbeck process, commonly used in diffusion models.

Our results hold for the general path, which includes these specific cases.

**Lemma B.1** (The relationship between the velocity and score). *Suppose $v_t(x)$ is a locally Lipschitz vector field which generates the probability path $p_t$ between $p_0 \sim \mathcal{N}(0, I)$ and $p_1$ with differentiable schedules $\alpha_t, \beta_t$ such that $X_t = \alpha_t X_0 + \beta_t X_1 \sim p_t$. Then the score can be expressed as a function of the velocity field by:*

$$\nabla \log p_t(x) = \frac{1}{\alpha_t \left(\frac{\dot{\beta}_t}{\beta_t}\alpha_t - \dot{\alpha}_t\right)} \left(v_t(x) - \frac{\dot{\beta}_t}{\beta_t}x\right) \tag{B.2}$$

*Specifically, for $\alpha_t = 1 - t, \beta_t = t$ (Flow Matching), the score can be expressed as*

$$\nabla \log p_t(x) = \frac{t v_t(x) - x}{1 - t} \tag{B.3}$$

*Proof.* The proof can be found in **B.4** of Domingo-Enrich et al. (2024). $\square$

**Lemma B.2** (Sum of Independent Random Variables). *Let $A$ and $B$ be two independent random variables in $\mathbb{R}^d$ with probability density functions $p_A(a)$ and $p_B(b)$, respectively. The probability density function of their sum, $C = A + B$, is given by the convolution of their individual PDFs:*

$$p_C(c) = (p_A * p_B)(c) = \int_{\mathbb{R}^d} p_A(y)p_B(c - y)dy \tag{B.4}$$

*Proof.* The proof can be found in Chapter 6 of Ross et al. (1998). $\square$

**Proposition B.1** (Distribution Along the Path of the Stochastic Interpolant). *The probability density function $p_t(x)$ of the random variable $X_t = \alpha_t X_0 + \beta_t X_1$ is given by the convolution of the scaled final density with a Gaussian kernel:*

$$p_t(x) = \left( \frac{1}{\beta_t^d} p_1 \left( \frac{\cdot}{\beta_t} \right) \right) * \mathcal{N}(x | \mathbf{0}, \alpha_t^2 \mathbf{I}) \tag{B.5}$$

*Proof.* $X_t$ is the sum of two independent random variables: $A = \alpha_t X_0$ and $B = \beta_t X_1$. The density of $A$ is $\mathcal{N}(x | \mathbf{0}, \alpha_t^2 \mathbf{I})$. The density of $B$ is $\frac{1}{\beta_t^d} p_1 \left( \frac{x}{\beta_t} \right)$. By lemma B.2, the density of their sum $X_t$ is the convolution of their respective densities. $\square$

**Lemma B.3** (Integrability of Exponential Functions with Quadratic Exponents). *Let*

$$f(x) = \exp\left( -\tfrac{1}{2} x^\top A x + b^\top x + c \right)$$

*be an exponential function with a quadratic exponent, where $A \in \mathbb{R}^{d \times d}$ is symmetric. Then $f \in L^1(\mathbb{R}^d)$ if and only if $A \succ 0$.*

Reviewer gBQy

*Proof.* ($\Rightarrow$) If $A \succ 0$, then $-\tfrac{1}{2} x^\top A x$ dominates the linear term $b^\top x$, so $f(x)$ is bounded by a Gaussian density and is therefore integrable over $\mathbb{R}^d$. ($\Leftarrow$) Suppose $A \not\succ 0$. Then there exists a unit eigenvector $v$ of $A$ with eigenvalue $\lambda \le 0$. Decompose $x = tv + y$ with $y \perp v$; in this orthogonal basis the exponent becomes

$$-\tfrac{1}{2} \lambda t^2 + b_1 t \; - \; \tfrac{1}{2} y^\top A_\perp y + b_\perp^\top y.$$

Consider integrating $f$ over the unbounded tube

$$\mathcal{C}_r = \{ (t, y) : \|y\| \le r \}, \qquad r > 0.$$

If $\lambda < 0$, the term $-\tfrac{1}{2} \lambda t^2$ grows *positively* and the integral diverges. If $\lambda = 0$, then the exponent along the $t$-direction is at most linear: If $b_1 \ne 0$, the integrand grows (or decays too slowly) linearly; if $b_1 = 0$, the integrand has no decay along $t$. In either case the integral over $\mathcal{C}_r$ diverges. Hence $f \notin L^1(\mathbb{R}^d)$ whenever $A$ is not positive definite. $\square$

### B.2 COMPACTLY SUPPORTED DISTRIBUTIONS

We consider compactly supported densities, a condition satisfied by the vast majority of real-world data. Examples include:

- **Images/Videos**: Pixel values are bounded, typically in a hypercube like $[0, 1]^{H \times W \times C}$. Videos are a sequence of images spread across the time dimension $T$, bounded in a hypercube of even higher dimension, like $[0, 1]^{H \times W \times C \times T}$.

- **3D Molecules/Shapes**: Atomic coordinates are constrained within a finite volume centered at the center of mass.

**Proposition B.2** (Isotropic Gaussian to Compactly Supported Target Density). *Let $p_0(x) = \mathcal{N}(x; 0, \sigma_0^2 I)$ and let $p_1$ be a probability density with compact support in $\mathbb{R}^d$. Let $\{p_t\}_{t \in [0,1]}$ be the probability path generated by stochastic interpolant $X_t = \alpha_t X_0 + \beta_t X_1$. Then, for any $t \in [0, 1)$, there exist finite positive constants $0 < C_\pm, \mu_t, V_t, R_t < \infty$ such that $p_t(x)$ is bounded by Gaussian envelopes:*

$$C_- \exp\left( \frac{-\|x - \mu_t\|^2 + \|\mu_t\|^2 - V_t}{2\alpha_t^2 \sigma_0^2} \right) \le p_t(x) \le C_+ \exp\left( \frac{-(\|x\| - R_t)^2 + R_t^2}{2\alpha_t^2 \sigma_0^2} \right)$$

*Proof.* Let $\mathcal{N}_t(x) = \mathcal{N}(x; 0, \sigma_t^2 I)$ be the density of $\alpha_t X_0$, where $\sigma_t^2 = \alpha_t^2 \sigma_0^2$. Let $\rho_t(y)$ be the density of $Y_t = \beta_t X_1$. Since $p_1$ is compactly supported, $\rho_t$ is supported on a compact set $\Omega_t$. The path density is the convolution $p_t = \mathcal{N}_t * \rho_t$. We analyze the ratio $p_t(x)/\mathcal{N}_t(x)$:

$$\frac{p_t(x)}{\mathcal{N}_t(x)} = \int_{\Omega_t} \frac{\mathcal{N}_t(x-y)}{\mathcal{N}_t(x)} \rho_t(y) dy = \int_{\Omega_t} \exp\left(\frac{2x \cdot y - \|y\|^2}{2\sigma_t^2}\right) \rho_t(y) dy = \mathbb{E}_{Y \sim \rho_t}\left[\exp\left(\frac{2x \cdot Y - \|Y\|^2}{2\sigma_t^2}\right)\right].$$

**Upper Bound:** Let $R_t = \sup_{y \in \Omega_t} \|y\| < \infty$. Using Cauchy-Schwarz, $2x \cdot y - \|y\|^2 \le 2\|x\|R_t$. Thus,

$$p_t(x) \le \mathcal{N}_t(x) \exp\left(\frac{2\|x\|R_t}{2\sigma_t^2}\right) = \frac{1}{(2\pi\sigma_t^2)^{d/2}} \exp\left(\frac{-\|x\|^2 + 2\|x\|R_t}{2\sigma_t^2}\right).$$

Completing the square in the exponent yields the form in the proposition statement.

**Lower Bound:** We apply Jensen's inequality ($\mathbb{E}[e^Z] \ge e^{\mathbb{E}[Z]}$). Let $\mu_t = \mathbb{E}[Y]$ and $V_t = \mathbb{E}[\|Y\|^2]$.

$$\frac{p_t(x)}{\mathcal{N}_t(x)} \ge \exp\left(\mathbb{E}_{Y \sim \rho_t}\left[\frac{2x \cdot Y - \|Y\|^2}{2\sigma_t^2}\right]\right) = \exp\left(\frac{2x \cdot \mu_t - V_t}{2\sigma_t^2}\right).$$

Multiplying by $\mathcal{N}_t(x)$ and rearranging terms yields the lower bound. $\square$

*Remark.* The compact support assumption is crucial. It ensures $R_t < \infty$ (valid upper bound) and the existence of all moments $\mu_t, V_t$ (valid lower bound). For heavy-tailed target distributions (e.g., Cauchy), these moments may not exist, invalidating the lower bound.

---

**Theorem B.1** (Integrability Preservation Condition for Compactly Supported Targets)**.** *For each $i \in \{1, \ldots, n\}$, let $\{q_t^{(i)}\}_{t \in [0,1]}$ be probability paths in $\mathbb{R}^{d_i}$ generated by $X_t^{(i)} = \alpha_t^{(i)} X_0^{(i)} + \beta_t^{(i)} X_1^{(i)}$ where $X_0^{(i)} \sim q_0^{(i)} = \mathcal{N}(0, I)$ and $X_1^{(i)} \sim q_1^{(i)}$ has compact support. Let $d := \max_i d_i$ and $\gamma_i(t) \in \mathbb{R}$ for $t \in [0,1]$.*

*Let $\pi_i : \mathbb{R}^d \to \mathbb{R}^{d_i}$ be projections onto coordinate sets $I_i$[a]. Define the lifted product*

$$h_t(x) := \prod_{i=1}^n (\tilde{q}_t^{(i)}(\pi_i(x)))^{\gamma_i(t)}$$

*If $h_1(x)$ is integrable and for every coordinate $k \in \{1, \ldots, d\}$ and all $t \in [0, 1)$,*

$$C_k(t) := \sum_{i:\, k \in I_i} \frac{\gamma_i(t)}{(\alpha_t^{(i)})^2} > 0, \tag{B.6}$$

*then $\{h_t\}_{t \in [0,1]}$ has the path existence property (i.e., $h_t \in L^1(\mathbb{R}^d)$ for all $t \in [0,1]$).*

*Conversely, if there exists a coordinate $k^* \in \{1, \ldots, d\}$ and $t^* \in [0, 1)$ such that $C_{k^*}(t^*) < 0$, then $\{h_t\}_{t \in [0,1]}$ is not integrable at $t^*$ (Marginal Path Collapse).*

---

[a]We can write $\pi_i(x_1, \ldots, x_d) = (x_{k_1}, \ldots, x_{k_{d_i}})$. Let $I_i := \{k_1, \ldots, k_{d_i}\}$ be the set of coordinate indices that $\pi_i$ projects onto. Lifted densities are $\tilde{q}_t^{(i)} = q_t^{(i)} \circ \pi_i$.

*Proof.* **1. Sufficiency for path existence.**

**Endpoint** ($t = 1$)**:** Integrability of $h_1$ holds by assumption.

**Interval** $t \in [0, 1)$**:** We construct an integrable upper bound. Applying Proposition B.2 to the lifted densities $\tilde{q}_t^{(i)}(x) = q_t^{(i)}(\pi_i(x))$, we use the upper bound formula for $\gamma_i(t) > 0$ and the lower bound formula for $\gamma_i(t) < 0$ (since negative exponents reverse the inequality).

$$h_t(x) \le \prod_{\gamma_i > 0} \left(C_{+,i} e^{\frac{-\|\pi_i(x)\|^2 + 2\|\pi_i(x)\|R_t^{(i)}}{2(\alpha_t^{(i)})^2}}\right)^{\gamma_i} \prod_{\gamma_i < 0} \left(C_{-,i} e^{\frac{-\|\pi_i(x) - \mu_t^{(i)}\|^2 + \|\mu_t^{(i)}\|^2 - V_t^{(i)}}{2(\alpha_t^{(i)})^2}}\right)^{\gamma_i}$$

The integrability is determined by the coefficient of the quadratic term $\|x\|^2$ in the exponent. Expanding $\|\pi_i(x)\|^2 = \sum_{k \in I_i} x_k^2$, the aggregate quadratic term in the exponent is:

$$\sum_{i=1}^n \gamma_i(t) \left( -\frac{\|\pi_i(x)\|^2}{2(\alpha_t^{(i)})^2} \right) = -\frac{1}{2} \sum_{i=1}^n \sum_{k \in I_i} \frac{\gamma_i(t)}{(\alpha_t^{(i)})^2} x_k^2 = -\frac{1}{2} \sum_{k=1}^d \left( \sum_{i:k \in I_i} \frac{\gamma_i(t)}{(\alpha_t^{(i)})^2} \right) x_k^2. \quad \text{(B.7)}$$

By Condition B.6, the coefficient for every $x_k^2$ is strictly negative. Thus, the upper bound behaves as $\exp(-\frac{1}{2} x^\top \Lambda_t x + O(\|x\|))$ with positive definite $\Lambda_t$, ensuring integrability (Lemma B.3).

**2. Sufficiency for Marginal Path Collapse at $t^*$.**

We construct a diverging lower bound. Applying Proposition B.2 using the lower bound formula for $\gamma_i(t) > 0$ and the upper bound formula for $\gamma_i(t) < 0$, we derive:

$$h_t(x) \geq \prod_{\gamma_i > 0} \left( C_{-,i} e^{\frac{-\|\pi_i(x) - \mu_t^{(i)}\|^2 + \|\mu_t^{(i)}\|^2 - V_t^{(i)}}{2(\alpha_t^{(i)})^2}} \right)^{\gamma_i} \prod_{\gamma_i < 0} \left( C_{+,i} e^{\frac{-\|\pi_i(x)\|^2 + 2\|\pi_i(x)\| R_t^{(i)}}{2(\alpha_t^{(i)})^2}} \right)^{\gamma_i}$$

The quadratic term in exponent is precisely equation B.7 since the quadratic terms are the same for both upper and lower bounds in Proposition B.2. Thus, the exponent can be written as $-\frac{1}{2} x^\top A x + D(x)$, where $A$ is a diagonal matrix with entries $A_{jj} = \sum_{i:k \in I_i} \frac{\gamma_i(t)}{(\alpha_t^{(i)})^2}$, and $D(x)$ collects the linear and constant terms. At time $t^*$, since $A_{k^* k^*} < 0$, the matrix $A$ is not positive definite ($A \not\succ 0$), and by Lemma B.3, the lower bound diverges, leading to Marginal Path Collapse. $\quad \square$

*Remark.* As a consequence of Theorem B.1 (sufficiency), the path $p_t^* = h_t / Z_t$, $t \in [0, 1]$ with $Z_t = \int_{\mathbb{R}^d} h_t(x) dx$ is well defined on $\mathbb{R}^d$, establishing the conditions for ACE (Theorem A.1).

### B.3 ADAPTIVE EXPONENTS AND BUMP FUNCTION CORRECTIONS

If the criterion in Theorem B.1 is violated, we can adapt the exponents $\gamma_i(t)$ to restore integrability. We present two cases: the general continuous case and the practical discretized case.

---

**Theorem B.2** (Adaptive Exponents). *Let $\alpha_t^{(i)}$ be differentiable on $[0, 1]$ with $\alpha_0^{(i)} = 1, \alpha_1^{(i)} = 0$, and $\alpha_t^{(i)} > 0$ for $t \in (0, 1)$. Let $\gamma_i(t)$ be the linear interpolation of fixed boundary values $\gamma_i(0), \gamma_i(1)$ such that*

$$S(t, \{\gamma_i\}_i) := \sum_{i=1}^n \frac{\gamma_i(t)}{(\alpha_t^{(i)})^2}$$

*satisfies $S(0, \{\gamma_i\}_i) > 0$ and $\lim_{t \to 1^-} S(t, \{\gamma_i\}_i) > 0$.*

*Then, there exist differentiable functions $\tilde{\gamma}_i(t)$ satisfying the boundary conditions such that $S(t, \{\tilde{\gamma}_i\}_i) > 0$ for all $t \in [0, 1)$.*

---

*Proof.* We prove by constructing a working solution. Note that the solution is not unique.

**Step 1.** First check if the default solution (linear interpolation of boundary values), $\gamma_i(t) = t\gamma_i(0) + (1 - t)\gamma_i(1)$ for all $i$ satisfies $S(t, \{\gamma_i\}_i) > 0 \ \forall t$. If so, we are done. If not, we may construct a valid solution in a single corrective step.

**Step 2.** If $\gamma_i$ fails, it is because $\exists t \in (0, 1)$ such that $S(t, \{\gamma_i\}_i) \leq 0$.

1. Find the time $t_{\min}$ where the sum $S(t, \{\gamma_i\}_i)$ is at its minimum. We can ensure that this exists, since there exists $\delta_1, \delta_2 > 0$ such that $\forall t \in [0, \delta_1) \cup (\delta_2, 1), S(t, \{\gamma_i\}_i) > 0$. This comes from our assumption $S(0, \{\gamma_i\}_i) > 0, \lim_{t \to 1^-} S(t, \{\gamma_i\}_i) > 0$ and the continuity of $S(t, \{\gamma_i\}_i)$ with respect to $t$. Then, $S(t, \{\gamma_i\}_i)$ is a continuous function on a compact set $[\delta_1, \delta_2]$ which admits a maximum and minimum value. Since the negative values of $S(t, \{\gamma_i\}_i)$ are all contained in the compact set $[\delta_1, \delta_2], \exists t_{\min} \in [\delta_1, \delta_2]$ such that

$$S_{\min} := \min_{t \in [0,1]} S(t, \{\gamma_i\}_i) = \min_{t \in [\delta_1, \delta_2]} S(t, \{\gamma_i\}_i) = S(t_{\min}, \{\gamma_i\}_i)$$

2. Define the **bump function** by $b(t) = t(1 - t)$ such that $b(0) = b(1) = 0$ and $b(t)$ is always nonnegative on $[0, 1]$. We denote the minimum value of $b(t)$ on $t \in [\delta_1, \delta_2]$ by $b_{\min} > 0$.

3. We will choose the index $j$ such that $\frac{\gamma_j(t_{\min})}{(\alpha_{t_{\min}}^{(j)})^2}$ is the largest. This will have a greater positive impact when the denominator is small.

4. Let $c_{\min} := \min_{t \in [\delta_1, \delta_2]} b_{\min}/(\alpha_t^{(j)})^2$. This exists and is positive since $\alpha_t^{(j)}$ is positive and continuous on $[\delta_1, \delta_2]$. We also have the inequality,

$$c_{\min} \leq \frac{b(t)}{(\alpha_t^{(j)})^2} \quad \forall t \in [\delta_1, \delta_2]$$

5. Finally, define the **adaptive exponent** via $\tilde{\gamma}_j(t) := \gamma_j(t) + B \cdot b(t)$ where

$$B = \left( \frac{|S_{\min}|}{c_{\min}} + 1 \right)$$

All other functions are preserved: $\tilde{\gamma}_i = \gamma_i \quad \forall i \neq j$.

Now we verify that $S(t, \{\tilde{\gamma}_i\}_i)$ with our adaptive exponent $\tilde{\gamma}_j(t)$ is strictly positive for all $t \in [0, 1]$. For $t \in [0, \delta_1) \cup (\delta_2, 1]$, $S(t, \{\tilde{\gamma}_i\}_i) \geq S(t, \{\gamma_i\}_i) > 0$ is already positive. For $t \in [\delta_1, \delta_2]$,

$$S(t, \{\tilde{\gamma}_i\}_i) = S(t, \{\gamma_i\}_i) + \left( \frac{|S_{\min}|}{c_{\min}} + 1 \right) \frac{b(t)}{(\alpha_t^{(j)})^2} \geq S_{\min} + |S(t_{\min})| + c_{\min} > 0$$

from the positivity of $c_{\min}$. $\qquad \square$

**Remark.** For numeric stability, we want each $\gamma_i(t)$ to stay as constant as possible. In other words, the functions that satisfy the boundary conditions while minimizing change,

$$\int_0^1 \sum_{i=1}^n (\dot{\gamma}_i(t))^2 dt$$

are the functions we want. The best a simplest candidate solution is therefore the constant function, $\gamma_i(t) = \gamma_i(0) = \gamma_i(1)$. Our solution guarantees

$$\int_0^1 \sum_{i=1}^n (\dot{\gamma}_i(t))^2 dt \leq \left( \frac{|S_{\min}|}{c_{\min}} + 1 \right)$$

---

**Theorem B.3** (Adaptive Exponents for Discretized Sampling). *Consider a discretization of the time interval $0 = t_0 < \cdots < t_M = t_{\text{end}} < 1$ and let $\alpha_t^{(i)}, \gamma_i(t), S(t, \{\gamma_i\}_i)$ be defined as in Theorem B.2. Assume $S(0, \{\gamma_i\}_i) > 0$. Then there exist differentiable functions $\tilde{\gamma}_i(t)$ preserving boundary values such that the criterion sum*

$$S(t, \{\tilde{\gamma}_i\}_i) = \sum_{i=1}^n \frac{\tilde{\gamma}_i(t)}{(\alpha_t^{(i)})^2}$$

*is strictly positive for all $t \in [0, t_{end}]$.*

---

*Proof.* Since $t_{\text{end}} < 1$ and $\alpha_t^{(i)} > 0$ on $[0, 1)$, the term $1/(\alpha_t^{(i)})^2$ is bounded on the compact interval $[0, t_{\text{end}}]$. If $S(t, \{\gamma_i\}_i) \leq 0$ at some point, let $t_{\min} \in (0, t_{\text{end}}]$ be the time minimizing $S(t, \{\gamma_i\}_i)$ and $S_{\min}$ be the minimum value. Since $S(0, \{\gamma_i\}_i) > 0$, there exists $\delta > 0$ such that $S(t, \{\gamma_i\}_i)$ is positive on $[0, \delta]$. We choose a smooth bump function $b(t)$ such that $b(0) = 0$, $b(1) = 0$, and $b(t) > 0$ on $[\delta, t_{\text{end}}]$. Pick index $j$. Since $\alpha_t^{(j)}$ is bounded and non-zero on $[\delta, t_{\text{end}}]$, the quantity $b(t)/(\alpha_t^{(j)})^2$ is bounded and non-negative. Let $C_{\min} > 0$ be its minimum on $[\delta, t_{\text{end}}]$.

We update $\tilde{\gamma}_j(t) = \gamma_j(t) + B \cdot b(t)$. For $t < \delta$, the sum is always positive. For $t \in [\delta, t_{\text{end}}]$, the new sum is

$$S(t, \{\tilde{\gamma}_i\}_i) = S(t, \{\gamma_i\}_i) + B \frac{b(t)}{(\alpha_t^{(j)})^2} \geq S_{\min} + B \cdot C_{\min}$$

By selecting $B$ sufficiently large, we can force $S(t, \{\tilde{\gamma}_i\}_i) > 0$ over the entire sampling interval while maintaining the fixed boundary values for the exponents. $\square$

## B.4 GAUSSIAN MIXTURE MODELS

We investigate integrability preservation for Gaussian Mixture Models (GMMs). Consider probability paths $\{p_t\}_{t \in [0,1]}$ generated by stochastic interpolants (Definition B.1), where the initial distribution $p_0$ is standard normal $\mathcal{N}(0, I)$ and the final distribution $p_1$ is a GMM.

---

**Definition B.2** (Gaussian Mixture Model). A **Gaussian Mixture Model (GMM)** $p(x)$ in $\mathbb{R}^d$ is a probability density of the form:

$$p(x) = \sum_{j=1}^{J} w_j \mathcal{N}(x; \mu_j, \Sigma_j) \tag{B.8}$$

where weights $w_j > 0$ sum to 1, and each covariance matrix $\Sigma_j$ is positive definite.

---

**Product of GMMs.** Similar to the compactly supported target case, integrability at the boundaries does not automatically imply integrability at intermediate times. A counterexample is provided in the main text (Eq. 3 and Fig. 2), demonstrating that intermediate paths can diverge even when endpoints are valid.

To guarantee integrability for ratios of product-of-GMMs, we must enforce a stronger structural condition. We first derive precise exponential bounds.

---

**Lemma B.4** (Exponential Bounds for GMMs). *Let* $q(x) = \sum_{j=1}^{J} c_j \mathcal{N}(x; \mu_j, \Sigma_j)$. *Define the extremal eigenvalues* $\lambda_{\max}(q) := \max_j \lambda_{\max}(\Sigma_j)$ *and* $\lambda_{\min}(q) := \min_j \lambda_{\min}(\Sigma_j)$. *Then there exist finite constants* $K_{\pm}, L_{\pm} > 0$ *such that for all* $x \in \mathbb{R}^d$:

$$q(x) \leq K_+ \exp\left(-\frac{\|x\|^2}{2\lambda_{\max}(q)} + L_+ \|x\|\right), \tag{B.9}$$

$$q(x) \geq K_- \exp\left(-\frac{\|x\|^2}{2\lambda_{\min}(q)} - L_- \|x\|\right). \tag{B.10}$$

---

*Proof.* **Upper Bound:** $q(x) \leq \sum_j \mathcal{N}(x; \mu_j, \Sigma_j)$. For each component $j$, we bound the quadratic form in the exponent using Rayleigh quotients:

$$(x - \mu_j)^\top \Sigma_j^{-1}(x - \mu_j) \geq \lambda_{\max}(\Sigma_j)^{-1}\|x - \mu_j\|^2 \geq \frac{1}{\lambda_{\max}(q)}(\|x\|^2 - 2\|x\|\|\mu_j\|).$$

Substituting this into the Gaussian PDF, we define $L_+ = \max_j \frac{\|\mu_j\|}{\lambda_{\max}(q)}$ and collect constants into $K_+$.

**Lower Bound:** $q(x) \geq c_{j^*} \mathcal{N}(x; \mu_{j^*}, \Sigma_{j^*})$ for any index $j^*$. We upper bound the quadratic form:

$$(x - \mu_{j^*})^\top \Sigma_{j^*}^{-1}(x - \mu_{j^*}) \leq \lambda_{\min}(\Sigma_{j^*})^{-1}\|x - \mu_{j^*}\|^2 \leq \frac{1}{\lambda_{\min}(q)}(\|x\|^2 + 2\|x\|\|\mu_{j^*}\| + \|\mu_{j^*}\|^2).$$

Substituting this yields the form in eq. (B.10) with $L_- = \frac{\|\mu_{j^*}\|}{\lambda_{\min}(q)}$. $\square$

We now state the sufficient and necessary conditions for integrability of heterogeneous products.

**Theorem B.4** (Integrability Preservation for Heterogeneous Products). *Let $\{q_t^{(i)}\}$ be GMM probability paths generated by $X_t^{(i)} = \alpha_t^{(i)} X_0^{(i)} + \beta_t^{(i)} X_1^{(i)}$ with $X_0^{(i)} \sim \mathcal{N}(0, I)$. Let $\pi_i : \mathbb{R}^d \to \mathbb{R}^{d_i}$ be projections onto coordinate index sets $I_i$, and define the composite function $g_t(x) := \prod_{i=1}^n (\tilde{q}_t^{(i)}(x))^{\gamma_i}$, where $\tilde{q}_t^{(i)} = q_t^{(i)} \circ \pi_i$.*

*For each coordinate $k \in \{1, \ldots, d\}$ and time $t \in [0, 1)$, we define two stability coefficients. The **sufficiency criterion** $C_k^{suff}(t)$ is defined as:*

$$C_k^{suff}(t) := \sum_{i: k \in I_i, \gamma_i > 0} \frac{\gamma_i}{\lambda_{\max}(q_t^{(i)})} + \sum_{i: k \in I_i, \gamma_i < 0} \frac{\gamma_i}{\lambda_{\min}(q_t^{(i)})}. \tag{B.11}$$

*The **necessity criterion** $C_k^{nec}(t)$ is defined as:*

$$C_k^{nec}(t) := \sum_{i: k \in I_i, \gamma_i > 0} \frac{\gamma_i}{\lambda_{\min}(q_t^{(i)})} + \sum_{i: k \in I_i, \gamma_i < 0} \frac{\gamma_i}{\lambda_{\max}(q_t^{(i)})}. \tag{B.12}$$

*If $g_1(x)$ is integrable, then:*

1. *(**Sufficiency**) If $C_k^{suff}(t) > 0$ for all $k$ and all $t \in [0, 1)$, then $g_t(x)$ is integrable for all $t \in [0, 1]$.*

2. *(**Necessity**) Conversely, if there exists a coordinate $k$ and time $t$ such that $C_k^{nec}(t) < 0$, then $g_t(x)$ is not integrable.*

*Proof.* **Sufficiency ($C_k^{\mathbf{suff}}(t) > 0$):** We construct an integrable upper bound for $g_t(x)$. To upper bound a product with positive and negative exponents, we require an *upper* bound for terms with $\gamma_i > 0$ and a *lower* bound for terms with $\gamma_i < 0$ (since negative exponents invert the inequality). Applying Lemma B.4:

$$g_t(x) \leq \prod_{\gamma_i > 0} \left( K_{+,i} e^{-\frac{\|\pi_i(x)\|^2}{2\lambda_{\max}(q_t^{(i)})} + L_{+,i}\|\pi_i(x)\|} \right)^{\gamma_i} \prod_{\gamma_i < 0} \left( K_{-,i} e^{-\frac{\|\pi_i(x)\|^2}{2\lambda_{\min}(q_t^{(i)})} - L_{-,i}\|\pi_i(x)\|} \right)^{\gamma_i}$$

$$= K \exp\left( -\frac{1}{2} \sum_{k=1}^d C_k^{\text{suff}}(t) x_k^2 + \sum_{i=1}^n \delta_i \|\pi_i(x)\| \right).$$

The coefficient of the quadratic term $x_k^2$ in the exponent corresponds exactly to $C_k^{\text{suff}}(t)$ defined in Eq. B.11. The linear terms are bounded by $D\|x\|$ for some finite $D > 0$ (the constants $\delta_i$ are also finite). Thus, the exponent behaves as $-\frac{1}{2}x^\top \text{diag}(C^{\text{suff}}(t))x + O(\|x\|)$. Since $C_k^{\text{suff}}(t) > 0$ for all $k$, the quadratic decay dominates the linear growth, ensuring integrability on $\mathbb{R}^d$.

**Necessity ($C_k^{\mathbf{nec}}(t) < 0$):** We prove this by constructing a lower bound that diverges. To lower bound the product, we require a *lower* bound for terms with $\gamma_i > 0$ and an *upper* bound for terms with $\gamma_i < 0$. Applying the reverse inequalities from Lemma B.4:

$$g_t(x) \geq \prod_{\gamma_i > 0} \left( K_{-,i} e^{-\frac{\|\pi_i(x)\|^2}{2\lambda_{\min}(q_t^{(i)})} - L_{-,i}\|\pi_i(x)\|} \right)^{\gamma_i} \prod_{\gamma_i < 0} \left( K_{+,i} e^{-\frac{\|\pi_i(x)\|^2}{2\lambda_{\max}(q_t^{(i)})} + L_{+,i}\|\pi_i(x)\|} \right)^{\gamma_i}$$

$$= \tilde{K} \exp\left( -\frac{1}{2} \sum_{k=1}^d C_k^{\text{nec}}(t) x_k^2 - \tilde{D}(x) \right)$$

where the quadratic coefficient is exactly $C_k^{\text{nec}}(t)$ defined in Eq. B.12, and $\tilde{D}(x)$ captures linear terms. The exponent is quadratic with coefficient matrix $A$, a diagonal matrix whose entries are $A_{jj} = C_k^{\text{nec}}(t)$. Let $k^*$ be a coordinate such that $C_{k^*}^{\text{nec}}(t) < 0$. Then $A_{k^* k^*} < 0$, so $A$ is not positive definite ($A \nsucc 0$), and by Lemma B.3 the integral over $\mathbb{R}^d$ diverges. $\square$

**Revisiting the counterexample in the main text.** We apply Theorem B.4 to analyze the pathological Gaussian path example presented in Eq. 3 and Figure 2 of the main text. Since the components

are single Gaussians (trivial GMMs with $J = 1$), the spectral bounds collapse to the scalar variance: $\lambda_{\max}(q_t^{(i)}) = \lambda_{\min}(q_t^{(i)}) = \sigma_i^2(t)$. Consequently, the sufficiency and necessity criteria coincide: $C_k^{\text{suff}}(t) = C_k^{\text{nec}}(t) =: C(t)$. The dimension is $d = 1$. The exponents are $\gamma_1 = 1, \gamma_2 = 1$ (numerator) and $\gamma_3 = -1, \gamma_4 = -1$ (denominator). The variance paths are given by $\sigma_i^2(t) = (1-t)^2\sigma_{i,0}^2 + t^2\sigma_{i,1}^2$. We evaluate the criterion at the critical time $t = 0.5$:

$$\sigma_1^2(0.5) = (0.5)^2(1) + (0.5)^2(0.5) = 0.375 \quad \sigma_2^2(0.5) = (0.5)^2(1) + (0.5)^2(7) = 2.0$$

$$\sigma_3^2(0.5) = (0.5)^2(1.5) + (0.5)^2(1) = 0.625 \quad \sigma_4^2(0.5) = 0.625 \quad \text{(identical to } q^{(3)})$$

Substituting these into Eq. B.12, $C(0.5) \approx -0.03 < 0$. Since $C(0.5) < 0$, the necessity condition of Theorem B.4 implies that $g_{0.5}(x)$ is **not integrable**. This theoretical prediction aligns perfectly with the empirical divergence and "Marginal Path Collapse" illustrated in Figure 2.

## C  EXPERIMENTAL SETUP

For full reproducibility, we provide the exact analytical expressions for the noise schedules $\alpha_t$ and $\beta_t$ used in our experiments in Table C.1. The stochastic interpolant is defined as $X_t = \alpha_t X_0 + \beta_t X_1$.

Table C.1: Exact formulations of noise schedules used in experiments.

| Schedule Name | Signal Scale $\alpha_t$ | Noise Scale $\beta_t$ |
|---|---|---|
| $1 - t$ | $1 - t$ | $t$ |
| $1 - t^2$ | $1 - t^2$ | $t$ |
| $\cos(\frac{\pi}{2}t)$ | $\cos\left(\frac{\pi}{2}t\right)$ | $\sin\left(\frac{\pi}{2}t\right)$ |
| DDPM | $\exp\left(-\frac{1}{4}(19.9t^2 + 0.1t)\right)$ | $\sqrt{1 - \alpha_t^2}$ |
| Sigmoid† | $\sqrt{1 - \exp\left(-\eta(1-t)\right)}$ | $t$ |
| Custom | $1 - 4t + 7t^2 - 4t^3$ | $t$ |

**Custom**: This polynomial schedule was used specifically for the synthetic quantitative benchmark (Table 2) to test robustness against non-monotonic effective variances. **DDPM** corresponds to the variance preserving (VP) schedule with linear $\beta$-scheduling from $\beta_{\min} = 0.1$ to $\beta_{\max} = 20$ (scaled to $t \in [0, 1]$). The expression is derived from $\alpha_t = \sqrt{\exp(-\int_0^t \beta(s)ds)}$. For **Sigmoid** we utilize a numerically stable formulation involving the softplus function. The term $\eta(x)$ is defined as:

$$\eta(x) = \frac{20}{12}\text{softplus}\left(12(x - 0.5)\right) + 0.001x, \quad \text{where softplus}(z) = \log(1 + e^z).$$

### C.1  SYNTHETIC DATASET: 2D CHECKER

**Dataset.** We evaluate the fidelity of samples generated by ACE, NR, and FKC by computing distributional discrepancies against the ground truth, reporting Wasserstein distances ($W_1$, $W_2$) and Maximum Mean Discrepancy with an RBF kernel (MMD). The ground truth 2D Checkerboard distribution on the domain $\mathcal{D} = [-4, 4]^2$ is formally defined by the density:

$$p_{\text{Checker}}(x, y) = \frac{1}{32} \cdot \mathbf{1}\left(\left\lfloor\frac{x}{2}\right\rfloor + \left\lfloor\frac{y}{2}\right\rfloor \text{ is odd}\right) \cdot \mathbf{1}\left((x, y) \in \mathcal{D}\right), \tag{C.1}$$

where $\lfloor \cdot \rfloor$ denotes the floor function and $\mathbf{1}(\cdot)$ is the indicator function. The support consists of alternating $2 \times 2$ squares.

**Implementation.** For all experiments, we sampled 10,000 samples over 1,000 SDE steps with noise level $\sigma_t = 0.5$. For the 2D benchmark experiments, we chose to resample based on Effective Sample Size (ESS), a commonly used technique in sequential Monte Carlo (Naesseth et al., 2019). We used a threshold of 0.7 (Hyperparameter study in Figure. C.1). For both 1D and 2D experiments, we used a multilayer perceptron (MLP) with a time embedding module concatenated to the input. The main network consisted of four hidden layers of 256 units with SiLU activations. Depending on the task

Reviewer naNy

dimensionality, the input layer size was $d + 256$ (where $d = 1, 2$ for 1D/2D inputs, respectively) and the output dimension matched the data dimension. Models were trained with the interpolant loss (Albergo et al., 2023) using the Adam optimizer at learning rate $2 \times 10^{-3}$, for 2000 epochs on 1D tasks and 10,000 epochs on 2D tasks. We use (Hutchinson, 1989) to compute divergences.

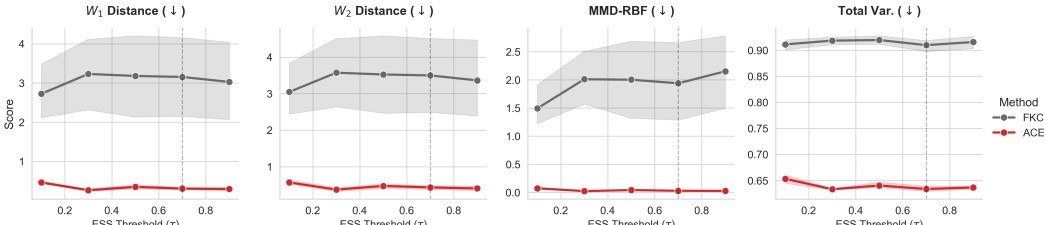

Figure C.1: **Performance comparison across varying ESS Thresholds.** Across multiple ESS thresholds, $\tau \in \{0.1, 0.3, 0.5, 0.7, 0.9\}$, ACE displays stable performance (with best values at $\tau = 0.7$) across different hyperparameter settings, outperforming FKC by a large margin. FKC consistently performs worse, implying its performance degradation is due to path collapse and not due to suboptimal parameter tuning. We used the same evaluation setup as in our main experiments. Shaded regions denote variance across 5 seeds.

**Path collapse frequencies.** For annealed three-expert composition $q_t^{(1)}(q_t^{(2)}/q_t^{(3)})^w$, there are $5^3 = 125$ total triplets, 120 heterogeneous triplets (excluding $\alpha_t^{(1)} = \alpha_t^{(2)} = \alpha_t^{(3)}$), and 100 likelihood-nonhomogeneous ($\alpha_t^{(2)} \neq \alpha_t^{(3)}$) triplets, the heterogeneous steering case. Among the 100 heterogeneous steering cases, path collapse frequencies are given in Table. E.5 with the full list of schedule compositions given in Figure. E.5.

**Visualization Details.** All sample/trajectory visualizations use seed 0. In criterion plots Figures 3 and E.5, we visualize the criterion $C(t)$ to demonstrate the prevalence of path collapse. To maintain a consistent and readable scale across all 100 combinations, we fix the y-axis range to $[-20, 100]$. For the majority of schedules, we plot the full trajectory $t \in [0, 1]$. However, in some cases (e.g., high-guidance regimes $w = 15$), the criterion may exhibit rapid asymptotic growth (exceeding 300) followed by a sharp singularity as $t \to 1$. In these cases, to prevent visual clutter, we cease plotting the line segment once it exits the visible y-axis frame. This visualization strategy focuses on the relevant operational range for discretized sampling ($t \leq t_{\text{end}}$, Theorem B.3), where the sign of $C(t)$ determines numerical stability.

## C.2 FLEXIBLE-POSE SCAFFOLD DECORATION

**Dataset.** We evaluate on a subset of CrossDocked2020 (Francoeur et al., 2020). To stress the benefit of flexible scaffold decoration, we select 9 ligand–pocket pairs (CrossDock-Weak) with weak binding (Vina score $> -5$ kcal/mol). To test generalization, we build an auxiliary dataset (CrossDock-SBDD) by sampling ligands from DiffSBDD and extracting scaffolds as a ring plus atoms within two bonds. The task is to generate ligands that (i) preserve the scaffold topology and (ii) achieve favorable binding to the pocket.

**Our Implementation** As described in Sec. 2.5, we combine pretrained density models: GeoDiff (Xu et al., 2022), DiffSBDD (Schneuing et al., 2024), and EDM (Hoogeboom et al., 2022b) as the conformer generation model, structure-based drug design model, and de novo generation model, respectively, enabling generation that preserves scaffold topology while allowing scaffold poses to adapt within the pocket. Unless stated otherwise, we use 500 denoising steps with resampling every 10 steps. For weighting the ratio-of-density, we use the weight values 1.1, 1.2, 1.3, and 1.4. For each pocket–scaffold pair, we sample five candidates per seed, repeat over two seeds, and report mean. We run our experiment on Python 3.8.9 and NVIDIA-A100 GPUs. We also note that we run ACE with batch size 10 and it takes 2 GB GPU VRAM.

**Baselines** We compare ACE with the FKC and NR methodologies in flexible scaffold decoration by constructing FKC, which applies FKC to the target density (Section 2.5) while sharing the

same pretrained topology-conditioned, unconditional, and pocket-conditioned models. NR is implemented as FKC without resampling. To further illustrate the benefit of flexibility, we adapt strong pocket-aware generative models to a fixed-pose regime by clamping scaffold atom coordinates to the reference pose and permitting generation only for non-scaffold atoms: Delete (Chen et al., 2025), DiffDec (Xie et al., 2024), and AutoFragDiff (Ghorbani et al., 2024).

**Metrics.**  For comparison between ACE and baseline diffusion steering methods NR, FKC, we evaluate the sampling path using three metrics. (i) Validity measures whether generated molecules are chemically consistent without fragmentation or the docking process of QVina2 (Alhossary et al., 2015) success. (ii) Docking Score, computed with QVina2 (Alhossary et al., 2015), assesses pocket compatibility. For each protein pocket, we report the Mean, Median, Best, Top-3, and Worst scores, as well as the Optimization Success Rate (OSR), defined as the proportion of generated molecules with docking scores better than the reference ligand. We further compare our flexible scaffold decoration pipeline against fixed-scaffold decoration baselines with the same metrics.

## D  BACKGROUND AND RELATED WORK

Our work sits at the intersection of three key areas: inference-time control for generative models, the theory of path sampling correctors, and the practical challenge of composing pretrained models for scientific discovery. We situate our contributions in relation to each (Table 1).

**Inference-Time Control and Guidance.** A major advantage of diffusion models is their steerability at inference time. Classifier-free guidance (CFG) is a widely used heuristic that mixes conditional and unconditional scores to enhance sample fidelity (Ho & Salimans, 2021; Chung et al., 2025). More recent work has proposed training-free or post hoc steering frameworks, including universal training-free guidance (Ye et al., 2024; Bansal et al., 2024), scalable steering pipelines (Singhal et al., 2025), and energy- or self-guided sampling approaches (Epstein et al., 2023; Yu et al., 2023; Song et al., 2023). These methods often adopt ratio-of-densities formulations (e.g., $p(x \mid c)^w/p(x)^{w-1}$), a powerful primitive that also underlies contrastive decoding in language models (Li et al., 2023a) and controlled generation in discrete diffusion models (Schiff et al., 2025; Hasan et al., 2025). While practically effective, these approaches focus on how to steer (e.g., by mixing scores) but provide no guarantees that the resulting sampling path remains valid when models are combined multiplicatively. Our work is the first to formalize and solve this underlying path-existence problem.

**Theoretical Foundations and Path Sampling Correctors.** The steering of stochastic processes has been studied using the Feynman–Kac (FK) formula (Stoltz et al., 2010), which connects pathwise sampling weights to principled unbiased estimators. This provides the basis for modern corrector methods in diffusion models (Skreta et al., 2025a;b; Mark et al., 2025; Hasan et al., 2025). Feynman–Kac Correctors (FKC) can in principle yield unbiased samples, but their guarantees require restrictive assumptions: (*i*) homogeneity of models with compatible noise schedules and dimensions, and (*ii*) constant exponents in the ratio-of-densities. These assumptions exclude important real-world cases, such as heterogeneous pretrained models or time-dependent negative exponents for denominator annealing. When violated, the FK framework provides no guidance, and *Marginal Path Collapse* occurs. Our work extends the FK formulation by introducing a path-existence criterion for heterogeneous products and incorporating time-varying exponents into an FK-style partial differential equation, enabling robust correction beyond prior limits.

**Composing Pretrained Models for Scientific Discovery.** The ability to combine specialized pretrained models is central in domains like structure-based drug design (SBDD). A researcher may wish to compose a protein-pocket conditioned ligand generator (Guan et al., 2023; Schneuing et al., 2024), a molecule conformer generator (Xu et al., 2022), and a denovo generation model (Hoogeboom et al., 2022b) into a single product distribution. Similar needs arise in scaffold decoration and fragment-based pipelines (Tan et al., 2022; Hu et al., 2023; Liu et al., 2025). However, naive multiplicative composition can trigger path collapse, making the generative process invalid. Our ACE framework directly enables robust multi-expert composition by guaranteeing path existence and providing adaptive exponent scheduling. This transforms composition from an unreliable heuristic into a principled, verifiable procedure. Applications to SBDD and scaffold decoration are especially compelling, as contemporary diffusion methods (Peng et al., 2022; Gao et al., 2024; Huang et al., 2024d;c) often assume fixed poses or homogeneous conditions—assumptions ACE can relax.

### D.1 COMMON STEERING OBJECTIVES AS RATIO-OF-DENSITIES PATHS.

Many standard inference-time control techniques can be unified under the ratio-of-densities framework, where the target density takes the form $p^\star(x) \propto \prod_i q^{(i)}(x)^{\gamma_i}$. Along the generative trajectory, this induces a time-indexed family $p_t^\star(x) \propto \prod_i q_t^{(i)}(x)^{\gamma_i}$, whose intermediate densities must remain normalizable for valid sampling. **Marginal Path Collapse** is the violation of this assumption.

**Classifier-free guidance (CFG):** To enhance samples consistent with condition $y$, CFG reweights the conditional model relative to the unconditional one, $p^\star(x) \propto p_\theta(x \mid y)^\gamma \, p_\theta(x)^{1-\gamma}$, implying the path $p_t^\star(x) \propto q_t(x \mid y)^\gamma \, q_t(x)^{1-\gamma}$ with score $\gamma \nabla \log q_t(x \mid y) + (1 - \gamma) \nabla \log q_t(x)$.

**Product-of-experts:** To enforce multiple constraints $c_1, \ldots, c_K$, one multiplies separate experts $q^{(k)}(x) = p_\theta(x \mid c_k)$, yielding $p^\star(x) \propto \prod_{k=1}^{K} q^{(k)}(x)$ and the path $p_t^\star(x) \propto \prod_{k=1}^{K} q_t^{(k)}(x)$.

**Reward-tilted / RL-style guidance:** Combining a base model $p_\theta(x)$ with a reward $r(x)$ creates a target $p^\star(x) \propto p_\theta(x) \exp(\beta r(x))$. This can be viewed as a product of base density $q_{\text{base}}(x)$ with a reward-density $q_{\text{rew}}(x) \propto \exp(\beta r(x))$, leading to the path $p_t^\star(x) \propto q_t^{\text{base}}(x) \, q_t^{\text{rew}}(x)$.

**Contrastive / reference model decoding:** To suppress generic patterns from a reference model $p_{\text{ref}}$, the target is defined as $p^\star(x) \propto p_{\text{LM}}(x)^\gamma \, p_{\text{ref}}(x)^{-\gamma}$, inducing the path $p_t^\star(x) \propto q_t^{\text{LM}}(x)^\gamma \, q_t^{\text{ref}}(x)^{-\gamma}$.

**Bayesian composition:** When only conditionals $p_\theta(x \mid c_k)$ are available, Bayes yields

$$p(x \mid c_{1:K}) \propto p(x) \prod_k p(c_k \mid x) \propto \underbrace{p(x)}_{\text{prior}} \prod_k p_\theta(x \mid c_k) / \underbrace{p_\theta(x)}_{\text{marginal}}$$

The assumptions and derivation under the heterogeneous conditioning structure is given in Eq. 11. This induces the path $p_t^\star(x) \propto q_t^{\text{prior}}(x) \prod_k q_t^{(k)}(x) \, q_t^{\text{marg}}(x)^{-1}$ which fits our general ratio-of-densities template.

**Scientific applications.** Our molecular DN/CONF/SBDD compositions for scaffold decoration, protein glue generation, fragment linking are the examples of Bayesian composition; the full derivations are given in Section E.1.

Reviewer gBQy, naNy

# E ADDITIONAL ANALYSES: HETEROGENEOUS COMPOSITION AND PATH COLLAPSE

## E.1 HETEROGENEOUS CONDITIONING STRUCTURES IN SCIENTIFIC TASKS

Reviewer gBQy, naNy

> **Definition E.1** (Heterogeneous Conditioning Structure). We define a heterogeneous conditioning structure as a property of the **target distribution** $p(X, Y \mid A, B)$ where conditioning variables exert partial or differing constraints on the system components. For instance, condition $A$ may constrain only the subset $X$, while condition $B$ constrains the joint system $(X, Y)$. This structure is distinct from global conditioning (where a condition affects all variables uniformly) and is pervasive in scientific modeling (see Figure E.2).

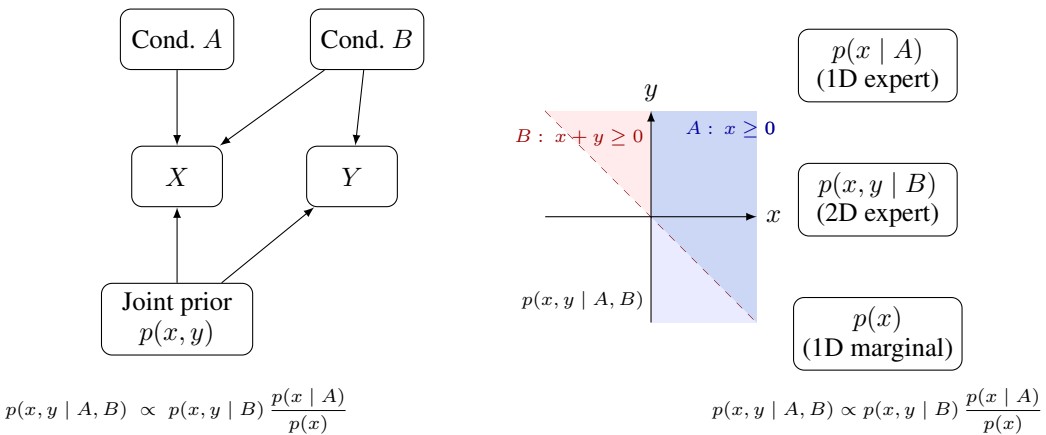

(a) Heterogeneous conditioning structure.  (b) Synthetic checker: 1D+2D experts.

Figure E.2: **Illustration of heterogeneous conditioning structure and its decomposition.** (a) Heterogeneous conditioning structure: condition $A$ acts only on $X$, condition $B$ acts on $(X, Y)$, leading to the Bayes factorization $p(x, y \mid A, B) \propto p(x \mid A) \, p(x, y \mid B)/p(x)$. (b) Synthetic checker: $A = \{x \geq 0\}$ constrains only the $x$-axis, while $B = \{x + y \geq 0\}$ constrains the joint plane, motivating a heterogeneous composition of two 1D experts and one 2D expert.

**Decomposition and Diffusion Steering.** Crucially, these heterogeneous target structures can be decomposed via Bayes' rule (Eq. 11) into products of simpler marginal or conditional distributions. This decomposition is powerful because it allows complex targets to be modeled by composing separate pretrained experts. We visualize this abstractly in Figure E.2a, which highlights how the joint prior governs the fusion of heterogeneous signals. For a concrete example, consider the constrained distribution $p(x, y \mid x \geq 0, \, x + y \geq 0)$ in Figure E.2b. It decomposes into

$$p(x, y \mid A, B) \propto p(x, y \mid B)\frac{p(x \mid A)}{p(x)}$$

allowing a heterogeneous task to be solved by combining a joint expert $p(x, y|B)$ with a marginal expert $p(x|A)$. While this compositional approach aligns with the principles of diffusion steering (e.g., FKC (Skreta et al., 2025a)), previous frameworks overlook the implications of mixing experts with differing domains and schedules, a "blind spot" that leads to the instabilities discussed in Appendix E.2.

**Prevalence in Molecular Generation.** Heterogeneous conditioning is the norm in molecular discovery. Figures E.3a and E.3b illustrate three critical tasks—**scaffold decoration** (Chen et al., 2025; Ghorbani et al., 2024; Xie et al., 2024), **protein-glue generation** (Kawasaki et al., 2023; Slabicki et al., 2020), and **fragment linking** (Sunseri & Koes, 2020; Igashov et al., 2024)—that fundamentally rely on this structure. In Examples E.1–E.3, we demonstrate that each of these tasks decomposes into a combination of three generative primitives: **de novo generation (DN)** (Hoogeboom et al., 2022a), **conformer generation (CONF)** (Xu et al., 2022), and **structure-based drug design**

**(SBDD)** (Schneuing et al., 2024). Consequently, solving these high-value problems requires the ability to faithfully compose these specific model families.

**Common Notation and Primitives.** In the following examples, we represent a molecule with $N$ atoms as a 3D point cloud $\mathcal{M}$ with bond topology $\mathcal{T}$. We denote the target protein pocket by $\mathcal{P}$ (`.pdb`). The condition of **stable binding** is denoted as $\mathcal{M} \leftrightarrow \mathcal{P}$ (satisfied when docking energy $U^{\text{Dock}} < \tau^{\text{Dock}}$). We utilize three fundamental pretrained model families:

- **DN**: Unconditional *De Novo* generation ($p(\mathcal{M})$).
- **CONF**: Topology-conditioned *Conformer* generation ($p(\mathcal{M} \mid \mathcal{T})$).
- **SBDD**: Pocket-conditioned *Structure-Based Drug Design* ($p(\mathcal{M} \mid \mathcal{P})$).

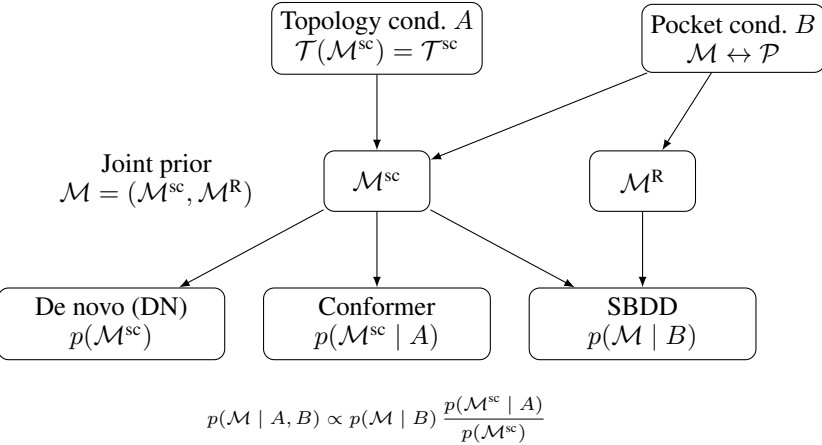

$$p(\mathcal{M} \mid A, B) \propto p(\mathcal{M} \mid B) \frac{p(\mathcal{M}^{\text{sc}} \mid A)}{p(\mathcal{M}^{\text{sc}})}$$

(a) Scaffold decoration: DN/CONF/SBDD composition.

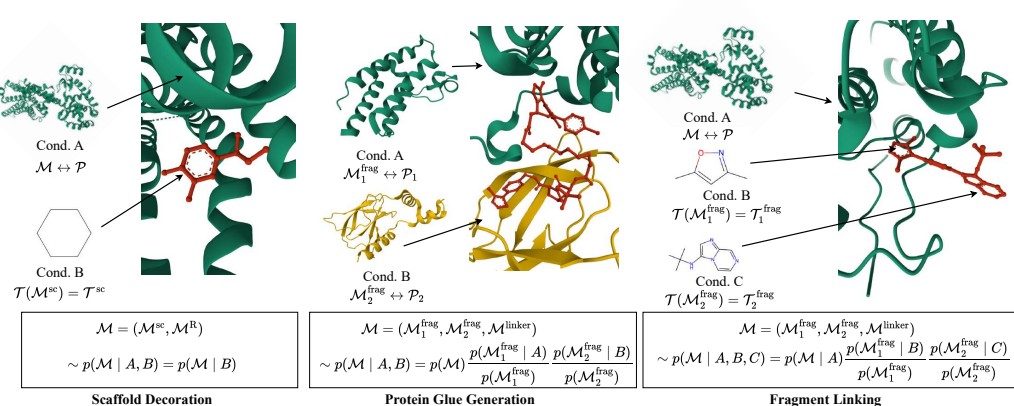

(b) Examples in molecular generation: Scaffold Decoration, Protein Glue Generation, Fragment Linking

Figure E.3: **Formulation of heterogeneous condition generation and its dominance in molecular generative tasks.** (a) Flexible-pose scaffold decoration mirrors the same structure: topology acts only on the scaffold coordinates, pocket binding acts on the full ligand, and the de-novo model provides the marginal over scaffolds; together they form a DN/CONF/SBDD ratio-of-densities target. (b) Illustration of three molecular generative tasks formulated under the heterogeneous-conditioning framework: scaffold decoration, protein-glue generation, and fragment linking. Each panel specifies the corresponding conditions, target distribution, and the decomposition into expert models for DN, CONF, and SBDD.

**Example E.1** (Scaffold Decoration). Scaffold decoration (Guan et al., 2009) is a pivotal strategy in lead optimization, enabling the refinement of physicochemical properties and potency while preserving the biological activity of a validated core structure. The goal is to generate

an R-group side chain $\mathcal{M}^{\mathrm{R}}$ optimized for binding to a pocket $\mathcal{P}$, while preserving a conserved scaffold backbone $\mathcal{M}^{\mathrm{sc}}$ with fixed topology $\mathcal{T}^{\mathrm{sc}}$ (SMILES).

**Formulation.** The task is to sample a molecule $\mathcal{M} = (\mathcal{M}^{\mathrm{sc}}, \mathcal{M}^{\mathrm{R}})$ from:

$$(\mathcal{M}^{\mathrm{sc}}, \mathcal{M}^{\mathrm{R}}) \sim p(\mathcal{M}^{\mathrm{sc}}, \mathcal{M}^{\mathrm{R}} \mid \mathcal{T}(\mathcal{M}^{\mathrm{sc}}) = \mathcal{T}^{\mathrm{sc}}, (\mathcal{M}^{\mathrm{sc}}, \mathcal{M}^{\mathrm{R}}) \leftrightarrow \mathcal{P}). \tag{E.1}$$

This exhibits the heterogeneous conditioning structure $p(X, Y \mid A, B)$ where $X = \mathcal{M}^{\mathrm{sc}}$ (constrained by topology $A$) and $Y = \mathcal{M}^{\mathrm{R}}$ (where the joint system is constrained by binding $B$).

**Decomposition.** Applying Bayes' rule, we decompose the target into a ratio of experts:

$$p^*(\mathcal{M}^{\mathrm{sc}}, \mathcal{M}^{\mathrm{R}}) \propto \frac{p(\mathcal{M}^{\mathrm{sc}} \mid \mathcal{T}(\mathcal{M}^{\mathrm{sc}}) = \mathcal{T}^{\mathrm{sc}}) \, p(\mathcal{M} \mid \mathcal{M} \leftrightarrow \mathcal{P})}{p(\mathcal{M}^{\mathrm{sc}})}. \tag{E.2}$$

ACE allows us to sample from this using off-the-shelf experts:

$$p(\mathcal{M}^{\mathrm{sc}} \mid \mathcal{T}^{\mathrm{sc}}) \rightarrow \textbf{CONF} \quad \text{(Fixes scaffold geometry)} \tag{E.3}$$

$$p(\mathcal{M}^{\mathrm{sc}}) \rightarrow \textbf{DN} \quad \text{(Prior correction)} \tag{E.4}$$

$$p(\mathcal{M} \mid \mathcal{P}) \rightarrow \textbf{SBDD} \quad \text{(Optimizes binding)} \tag{E.5}$$

---

**Example E.2** (Protein Glue Generation). Protein glues (Mogaki et al., 2019) represent a transformative therapeutic class capable of targeting "undruggable" proteins by inducing novel interactions, such as ubiquitin-mediated degradation. This task requires designing a linker $\mathcal{M}^{\mathrm{linker}}$ that connects two molecular fragments $\mathcal{M}^{\mathrm{frag}}_1, \mathcal{M}^{\mathrm{frag}}_2$ such that they simultaneously bind to two distinct protein surfaces $\mathcal{P}_1, \mathcal{P}_2$.

**Formulation.** The full molecule $\mathcal{M} = (\mathcal{M}^{\mathrm{frag}}_1, \mathcal{M}^{\mathrm{frag}}_2, \mathcal{M}^{\mathrm{linker}})$ is sampled from:

$$\mathcal{M} \sim p(\mathcal{M} \mid \mathcal{M}^{\mathrm{frag}}_1 \leftrightarrow \mathcal{P}_1, \mathcal{M}^{\mathrm{frag}}_2 \leftrightarrow \mathcal{P}_2). \tag{E.6}$$

This corresponds to $p(X, Y \mid A, B)$ where $A$ constrains fragment 1 and $B$ constrains fragment 2.

**Decomposition.** We decompose the posterior to isolate the binding constraints:

$$p^*(\mathcal{M}) \propto \frac{p(\mathcal{M}^{\mathrm{frag}}_1 \mid \mathcal{P}_1) \, p(\mathcal{M}^{\mathrm{frag}}_2 \mid \mathcal{P}_2) \, p(\mathcal{M})}{p(\mathcal{M}^{\mathrm{frag}}_1) \, p(\mathcal{M}^{\mathrm{frag}}_2)}. \tag{E.7}$$

This task effectively composes two local binding models with a global prior:

$$p(\mathcal{M}^{\mathrm{frag}}_i \mid \mathcal{P}_i) \rightarrow \textbf{SBDD} \quad \text{(Fragment-specific binding)} \tag{E.8}$$

$$p(\mathcal{M}^{\mathrm{frag}}_i), \, p(\mathcal{M}) \rightarrow \textbf{DN} \quad \text{(Global \& Marginal priors)} \tag{E.9}$$

---

**Example E.3** (Fragment Linking). Fragment linking (Bedwell et al., 2022) is the cornerstone of fragment-based drug discovery (FBDD), enabling the construction of high-affinity ligands by chemically bridging low-affinity fragments bound to adjacent sub-pockets. The goal is to assemble a ligand by connecting two pre-determined fragments $\mathcal{M}^{\mathrm{frag}}_1, \mathcal{M}^{\mathrm{frag}}_2$ (with fixed chemical topologies $\mathcal{T}_1, \mathcal{T}_2$) via a generated linker $\mathcal{M}^{\mathrm{linker}}$, ensuring the final structure binds to a target pocket $\mathcal{P}$.

**Formulation.** The molecule $\mathcal{M}$ is sampled from:

$$\mathcal{M} \sim p(\mathcal{M} \mid \mathcal{T}(\mathcal{M}^{\mathrm{frag}}_1) = \mathcal{T}_1, \, \mathcal{T}(\mathcal{M}^{\mathrm{frag}}_2) = \mathcal{T}_2, \, \mathcal{M} \leftrightarrow \mathcal{P}). \tag{E.10}$$

This represents a complex conditioning structure $p(X, Y \mid A, B, C)$ where $A, B$ enforce local topology and $C$ enforces global binding.

**Decomposition.** The target density factorizes as:

$$p^*(\mathcal{M}) \propto \frac{p(\mathcal{M}_1^{\mathrm{frag}} \mid \mathcal{T}_1) \, p(\mathcal{M}_2^{\mathrm{frag}} \mid \mathcal{T}_2) \, p(\mathcal{M} \mid \mathcal{P})}{p(\mathcal{M}_1^{\mathrm{frag}}) \, p(\mathcal{M}_2^{\mathrm{frag}})}. \tag{E.11}$$

ACE constructs this path by coordinating three distinct model families:

$$p(\mathcal{M}_i^{\mathrm{frag}} \mid \mathcal{T}_i) \to \textbf{CONF} \quad \text{(Fragment topology)} \tag{E.12}$$

$$p(\mathcal{M}_i^{\mathrm{frag}}) \to \textbf{DN} \quad \text{(Marginal correction)} \tag{E.13}$$

$$p(\mathcal{M} \mid \mathcal{P}) \to \textbf{SBDD} \quad \text{(Global binding)} \tag{E.14}$$

**From Heterogeneous Targets to Heterogeneous Paths.** While the heterogeneous *conditioning structure* dictates which expert models must be combined to define the target at $t = 1$, the resulting *probability path* for $t \in (0, 1)$ depends on the noise schedulers of those experts. This creates a systemic challenge: different model families (DN, CONF, SBDD) are historically trained with different noise schedules. We conducted a comprehensive survey of existing diffusion and flow-based models across these domains (summarized in Tables E.2, E.3, and E.4). The results show a lack of standardization, with the literature employing a fragmented mix of sigmoid, polynomial, and cosine schedules. Therefore, constructing generative paths for heterogeneous scientific tasks *inevitably* leads to heterogeneous schedules, where the mismatch in noise contraction rates induces path collapse. This confirms that the collapse phenomenon is not an edge case, but an inherent obstacle arising from the modular nature of scientific generative modeling. We provide a detailed quantitative analysis of this phenomenon, demonstrating its systematic prevalence across standard model combinations and its detrimental impact on sampling, in Appendix E.2.

**The Infeasibility of Homogeneous Composition.** Since Tables E.2–E.4 show that several schedulers appear across tasks, one might wonder whether composing models under a *shared* (homogeneous) scheduler could avoid path collapse. We clarify that using a common schedule across experts is generally infeasible in molecular generation due to two fundamental constraints.

**1. Heterogeneity in Representations.** Unlike images, molecules admit multiple incompatible representations. Some models encode atom types as continuous one-hot vectors $H^{\mathrm{one\text{-}hot}} \in \mathbb{R}^N$ evolving under Gaussian convolution; others use categorical variables with discrete states (Dunn & Koes, 2025; Lin et al., 2023) or operate in latent spaces (Ketata et al., 2024; Oestreich et al., 2024). Because these spaces differ in dimensionality, continuity, and semantic meaning, they cannot be embedded into a unified ambient space where a single diffusion path is consistently defined. This fundamentally constrains which experts can be mathematically composed.

**2. The "Empty Intersection" of Schedules.** Even if we restrict our scope to models with compatible continuous representations, a homogeneous triplet often does not exist. In our scaffold decoration experiment, all applicable DN models (EDM, GCDM) employ **quadratic** schedulers, while SBDD models (DiffSBDD, DualDiff) use either quadratic or sigmoid schedules. However, state-of-the-art CONF models not use the quadratic schedulers but mostly adopt the **sigmoid** schedules. Consequently, it is impossible to construct a homogeneous DN/CONF/SBDD triplet. This structural mismatch necessitates the use of heterogeneous schedules (quadratic for DN/SBDD, sigmoid for CONF), making the path-existence guarantees of ACE a prerequisite for reliable generation.

Table E.2: **(CONF)** Diffusion and flow-based models for molecular conformer generation. $N$ denotes the number of atoms.

| Model / Paper | Scheduler | Main Datasets | Domain |
|---|---|---|---|
| **GeoDiff** (Xu et al., 2022) | $\alpha_t = \exp\left(-\frac{5}{6}\left[\log(1+e^{12(t-\frac{1}{2})}) - \log(1+e^{-6})\right]\right)$ $\beta_t = \sqrt{1 - \exp\left(-\frac{5}{3}\left[\log(1+e^{12(t-\frac{1}{2})}) - \log(1+e^{-6})\right]\right)}$ | GEOM-QM9, GEOM-Drugs | $\mathbb{R}^{3N}$ |
| **TorsionDiff** (Jing et al., 2022) | $\alpha_t = (\frac{\pi}{100})^t \pi^{1-t}$ $\beta_t = 1$ | GEOM-Drugs | $SO(2)^M$ |
| **EC-Conf** (Fan et al., 2023) | $\alpha_t = T(1-t)$ $\beta_t = 1$ | GEOM-QM9, GEOM-Drugs | $\mathbb{R}^{3N}$ |
| **GADIFF** (Wang et al., 2024a) | $\alpha_t = \exp\left(-\frac{5}{6}\left[\log(1+e^{12(t-\frac{1}{2})}) - \log(1+e^{-6})\right]\right)$ $\beta_t = \sqrt{1 - \exp\left(-\frac{5}{3}\left[\log(1+e^{12(t-\frac{1}{2})}) - \log(1+e^{-6})\right]\right)}$ | GEOM-QM9, GEOM-Drugs | $\mathbb{R}^{3N}$ |
| **ET-Flow** (Hassan et al., 2024) | $\alpha_t = 1-t$ $\beta_t = t$ | GEOM-QM9, GEOM-Drugs | $\mathbb{R}^{3N}$ |
| **AGDIFF** (Vieira Wyzykowski et al., 2025) | $\alpha_t = \exp\left(-\frac{5}{6}\left[\log(1+e^{12(t-\frac{1}{2})}) - \log(1+e^{-6})\right]\right)$ $\beta_t = \sqrt{1 - \exp\left(-\frac{5}{3}\left[\log(1+e^{12(t-\frac{1}{2})}) - \log(1+e^{-6})\right]\right)}$ | GEOM-QM9, GEOM-Drugs | $\mathbb{R}^{3N}$ |
| **LoQI** (Nikitin et al., 2025) | $\alpha_t = \cos\left(\frac{\pi}{2}t\right)$ $\beta_t = \sin\left(\frac{\pi}{2}t\right)$ | ChEMBL3D | $\mathbb{R}^{3N}$ |
| **CoFM** (Xu et al., 2025) | $\alpha_t = 1-t$ $\beta_t = t$ | GEOM-QM9 | $\mathbb{R}^{3N}$ |

Table E.3: **(DN)** Diffusion and flow-based models for de novo molecular generation across coordinate-space, latent-space, and mixed graph–geometry domains. $N$ denotes the number of atoms and $A$, $B$, $C$ the number of atom types, bond types, and formal charge types. $\mathcal{A} = [A], \mathcal{B} = [B]$, and $\mathcal{C}$ are the categorical spaces of $A$ atom types, $B$ bond types, and $C$ formal charge types.

| Model / Paper | Scheduler | Main Datasets | Domain |
|---|---|---|---|
| **EDM** (Hoogeboom et al., 2022a) | $\alpha_t = \sqrt{1 - (1-(t-1)^2)^2}$ $\beta_t = 1 - (t-1)^2$ | GEOM-QM9, GEOM-Drugs | $\mathbb{R}^{3N+AN}$ |
| **MiDi** (Vignac et al., 2023) | $\alpha_t = \cos\left(\frac{\pi}{2}t\right)$ $\beta_t = \sin\left(\frac{\pi}{2}t\right)$ | QM9, GEOM-Drugs | $\mathbb{R}^{3N} \times \mathcal{A}^N \times \mathcal{B}^{\frac{N(N-1)}{2}}$ |
| **GCDM** (Morehead & Cheng, 2023) | $\alpha_t = \sqrt{1 - (1-(t-1)^2)^2}$ $\beta_t = 1 - (t-1)^2$ | GEOM-QM9, GEOM-Drugs | $\mathbb{R}^{3N+AN}$ |
| **EDM-SyCo** (Ketata et al., 2024) | $\alpha_t = \sqrt{1 - (1-(t-1)^2)^2}$ $\beta_t = 1 - (t-1)^2$ | ZINC250K, GuacaMol | Latent Euclidean Space |
| **DrugDiff** (Oestreich et al., 2024) | $\alpha_t = \sqrt{1 - (1-(t-1)^2)^2}$ $\beta_t = 1 - (t-1)^2$ | ChEMBL, GuacaMol | Latent Euclidean Space |
| **VEDA** (Zhang et al., 2025) | $\alpha_t = T_{\min}\left(\frac{T_{\max}}{T_{\min}}\right)^{(1-\rho)(1-t)+\rho\frac{2}{\pi}\arcsin\sqrt{1-t}}$ $\beta_t = 1$ | GEOM-QM9, GEOM-Drugs | $\mathbb{R}^{3N} \times \mathcal{A}^N$ |
| **FlowMol3** (Dunn & Koes, 2025) | $\alpha_t = 1-t$ $\beta_t = t$ | GEOM-Drugs | $\mathbb{R}^{3N} \times \mathcal{A}^N \times \mathcal{B}^{\frac{N(N-1)}{2}} \times \mathcal{C}^N$ |

Table E.4: **(SBDD)** Pocket-conditioned (structure-based drug design) diffusion and flow-based models, including sigmoid, cosine, and quadratic noise schedules across 3D and mixed 3D–categorical domains. $N$ denotes the number of atoms and $A$ the number of atom types. $\mathcal{A} = [A]$ is the categorical space of $A$ atom types. For D3FG, $N_{\text{fg}}$ and $N_{\text{at}}$ denote the numbers of functional-group atoms and remaining atoms, respectively, and $\mathcal{A}_{\text{fg}}$ and $\mathcal{A}_{\text{at}}$ denote the categorical spaces for functional-group types and atom types.

| Model / Paper | Scheduler | Main Datasets | Domain |
|---|---|---|---|
| **TargetDiff** (Guan et al., 2023) | $\alpha_t = \exp\left(-\frac{5}{6}[\log(1 + e^{12(t-\frac{1}{2})}) - \log(1 + e^{-6})]\right)$ 
 $\beta_t = \sqrt{1 - \exp\left(-\frac{5}{3}[\log(1 + e^{12(t-\frac{1}{2})}) - \log(1 + e^{-6})]\right)}$ | CrossDocked2020 | $\mathbb{R}^{3N} \times \mathcal{A}^N$ |
| **D3FG** (Lin et al., 2023) | $\alpha_t = \cos\left(\frac{\pi}{2}t\right)$ 
 $\beta_t = \sin\left(\frac{\pi}{2}t\right)$ | CrossDocked2020 | $\mathbb{R}^{3(N_{\text{fg}}+N_{\text{at}})} \times \mathcal{A}_{\text{fg}}^N \times \mathcal{A}_{\text{at}}^N$ |
| **DiffSBDD** (Schneuing et al., 2024) | $\alpha_t = \sqrt{1 - (1 - (t-1)^2)^2}$ 
 $\beta_t = 1 - (t-1)^2$ | CrossDocked2020 | $\mathbb{R}^{3N+AN}$ |
| **BindDM** (Huang et al., 2024b) | $\alpha_t = \exp\left(-\frac{5}{6}[\log(1 + e^{12(t-\frac{1}{2})}) - \log(1 + e^{-6})]\right)$ 
 $\beta_t = \sqrt{1 - \exp\left(-\frac{5}{3}[\log(1 + e^{12(t-\frac{1}{2})}) - \log(1 + e^{-6})]\right)}$ | CrossDocked2020 | $\mathbb{R}^{3N} \times \mathcal{A}^N$ |
| **DualDiff** (Huang et al., 2024a) | $\alpha_t = \exp\left(-\frac{5}{6}[\log(1 + e^{12(t-\frac{1}{2})}) - \log(1 + e^{-6})]\right)$ 
 $\beta_t = \sqrt{1 - \exp\left(-\frac{5}{3}[\log(1 + e^{12(t-\frac{1}{2})}) - \log(1 + e^{-6})]\right)}$ | CrossDocked2020 | $\mathbb{R}^{3N+AN}$ |
| **MolSnapper** (Ziv et al., 2025) | $\alpha_t = \cos\left(\frac{\pi}{2}t\right)$ 
 $\beta_t = \sin\left(\frac{\pi}{2}t\right)$ | CrossDocked2020 | $\mathbb{R}^{3N} \times \mathcal{A}^N$ |
| **PAFlow** (Zhou et al., 2025) | $\alpha_t = \exp\left(-\frac{5}{6}[\log(1 + e^{12(t-\frac{1}{2})}) - \log(1 + e^{-6})]\right)$ 
 $\beta_t = \sqrt{1 - \exp\left(-\frac{5}{3}[\log(1 + e^{12(t-\frac{1}{2})}) - \log(1 + e^{-6})]\right)}$ | CrossDocked2020 | $\mathbb{R}^{3N} \times \mathcal{A}^N$ |
| **MolFORM** (Huang & Zhang, 2025) | $\alpha_t = 1 - t$ 
 $\beta_t = t$ | CrossDocked2020 | $\mathbb{R}^{3N} \times \mathcal{A}^N$ |

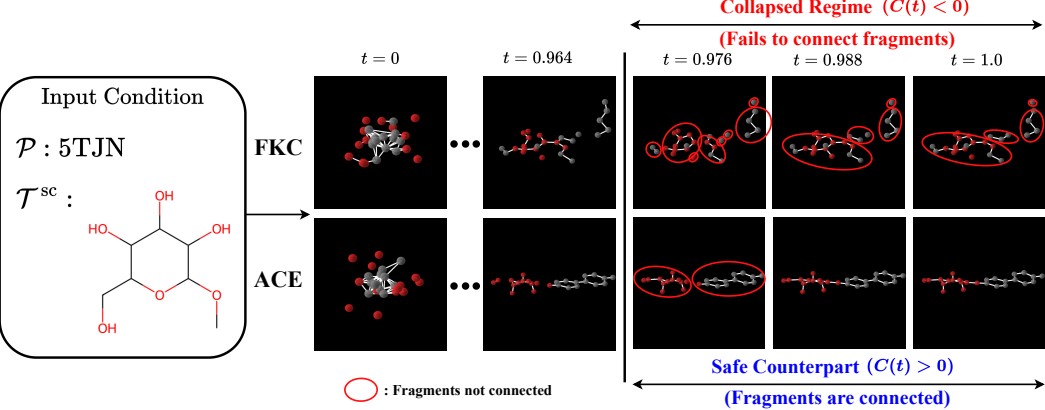

Figure E.4: **Impact of Marginal Path Collapse on molecular generation.** We compare sampling trajectories for FKC and ACE on the scaffold-decoration task ($\omega = 1.3$, 500 steps). Both methods target the same distribution $p_\omega(\mathcal{M}) \propto p(\mathcal{M})(p(\mathcal{M} \mid \mathcal{T}^{\text{sc}}, \mathcal{P})/p(\mathcal{M}))^\omega$. Crucially, FKC enters a collapsed regime for $t \in (0.974, 1)$ where the path existence criterion is violated ($C(t) < 0$). This theoretical singularity manifests empirically as a failure to assemble, resulting in disjoint fragments (top row). In contrast, ACE employs the bump correction $B(t) = 30t(1 - t)$ to guarantee $C(t) > 0$ throughout the trajectory, reliably ensuring the successful formation of a connected, chemically valid molecule (bottom row).

## E.2 PREVALENCE AND CONSEQUENCES OF MARGINAL PATH COLLAPSE

Reviewer
TtEF,
naNy

To confirm that Marginal Path Collapse is a systematic vulnerability rather than an edge case, we present a quantitative analysis of its frequency across standard model combinations and examine its downstream impact on generation quality.

**Systematic Prevalence of Collapse.** We evaluated all $5^3 = 125$ annealed compositions of three experts $h_t = q_t^{(1)}(q_t^{(2)}/q_t^{(3)})^w$ formed from five standard noise schedules: DDPM (Ho et al., 2020), cosine (Nichol & Dhariwal, 2021), sigmoid (Xu et al., 2022), linear (Lipman et al., 2023), and polynomial (Hoogeboom et al., 2022a). Excluding trivial homogeneous combinations, the results confirm that path collapse is a widespread failure mode. As shown in Table E.5, the collapse rate for heterogeneous compositions starts at **41%** even at a unit guidance scale ($w = 1.0$). Crucially, as the guidance scale increases, as in standard practice to improve sample quality, the collapse rate rises sharply, reaching **66%** at moderate guidance ($w = 2.0$) and **80%** at high guidance ($w = 15$). This indicates that in heterogeneous compositions, operating on invalid paths is the statistical norm, not the exception.

Table E.5: Frequency of Marginal Path Collapse across 100 heterogeneous schedule compositions. Higher guidance scales $w$ drastically increase the likelihood of encountering invalid paths.

| Guidance Scale ($w$) | # Collapses | % Collapses |
|---|---|---|
| 1.0 | 41 | 41% |
| 1.1 | 47 | 47% |
| 1.5 | 52 | 52% |
| 2.0 | 66 | 66% |
| 7.5 | 77 | 77% |
| 15 | 80 | 80% |

**Empirical Consequences of Collapse.** The violation of the path existence criterion has tangible negative impacts on generation performance, as illustrated in Figures E.2 and E.6–E.11. We observe distinct failure modes across domains:

- **Functional Failure (Molecular Domain):** In scaffold decoration, path collapse correlates directly with the generation of chemically invalid structures. As quantified in Tables 3–4, baselines (NR, FKC) suffer from significantly higher rates of invalid samples when the criterion is violated. Figure E.4 visualizes this stark contrast: ACE produces chemically valid molecules that respect valency constraints, whereas FKC generates fragmented, disconnected structures. While inherent network approximation errors can occasionally yield invalid samples in any method, ACE's failure rate is negligible (maintaining near-perfect validity) compared to the systemic collapse observed in baselines.

- **Distributional Failure (Synthetic Domain):** On synthetic benchmarks, the consequences manifest as a loss of distributional fidelity (Figure E.6). NR fails to effectively remove out-of-distribution samples due to its heuristic nature, regardless of path collapse (Table E.6 shows non-collapse results). More critically, FKC performance degrades because its importance weights rely on the assumption that the drift term corresponds to the score of a valid probability distribution. When path collapse occurs, the simple mixture of scores, while computationally possible, does not correspond to a score of normalizable distribution. This theoretical violation destabilizes the importance weights, leading to unpredictable behavior such as severe *mode collapse* (see trajectory plots in Figures E.7–E.11).

Marginal Path Collapse is a pervasive practical problem in heterogeneous model composition. Ignoring it leads to measurable degradation in both sample validity and distributional diversity. ACE provides the necessary theoretical and practical fix to guarantee stable, high-quality generation in these heterogeneous regimes.

Table E.6: **Metrics under Homogeneous Composition (collapse duration 0%).** Across 5 seeds, We evaluate the scenario where all expert schedules are identical ($\alpha_t^{(i)}$ = DDPM for all $i$), resulting in a collapse duration of 0.0%. ACE criterion is naturally satisfied without correction ($B = 0$), making ACE mathematically equivalent to FKC. Both methods significantly outperform the heuristic NR (CFG) baseline, which suffers from approximation errors even when the path is valid.

| Method | $W_1(\downarrow)$ | $W_2(\downarrow)$ | MMD (RBF) $(\downarrow)$ |
|---|---|---|---|
| NR (CFG) | $0.859 \pm 0.023$ | $1.113 \pm 0.023$ | $0.077 \pm 0.003$ |
| FKC | $\mathbf{0.162 \pm 0.019}$ | $\mathbf{0.282 \pm 0.017}$ | $\mathbf{0.012 \pm 0.001}$ |
| ACE | $\mathbf{0.162 \pm 0.019}$ | $\mathbf{0.282 \pm 0.017}$ | $\mathbf{0.012 \pm 0.001}$ |

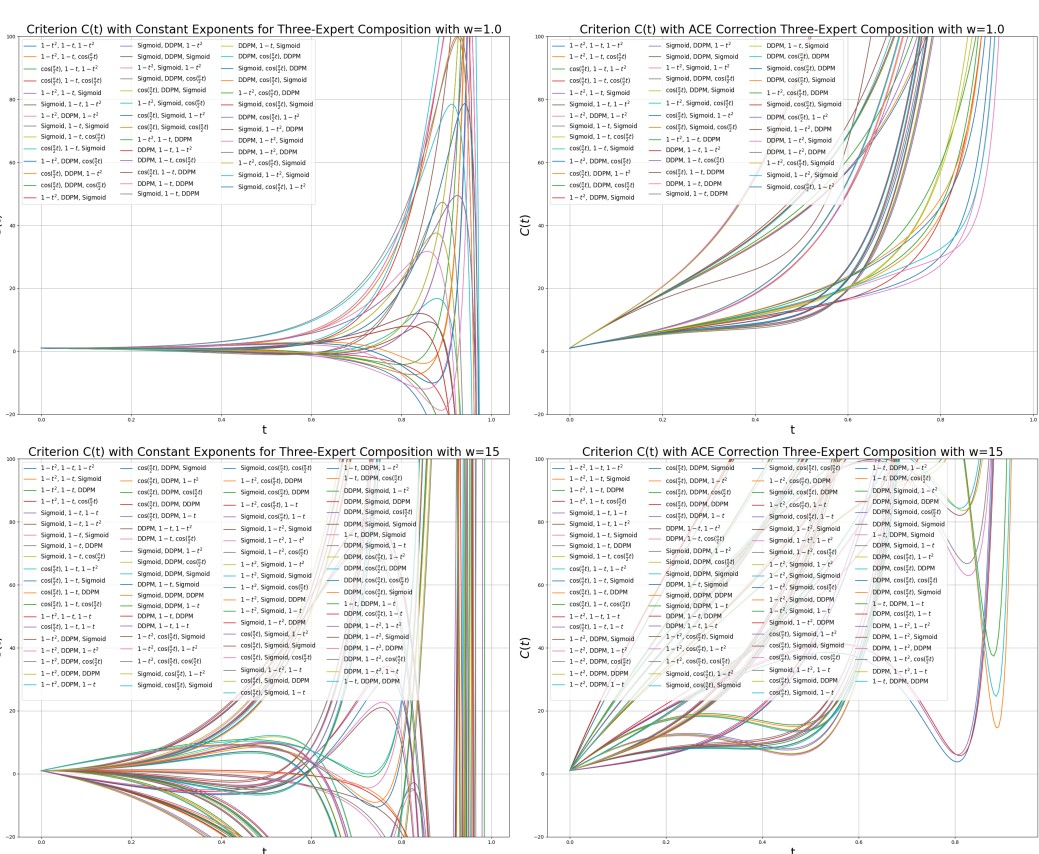

Figure E.5: **Path existence criterion $C(t)$ across heterogeneous schedule combinations.** We visualize the evolution of $C(t)$ for the 100 heterogeneous triplets from Table E.5 under low ($w = 1.0$, top) and high ($w = 15.0$, bottom) guidance scales. **Left (Constant Exponents):** A significant fraction of combinations violate the existence condition ($C(t) < 0$), with the frequency and magnitude of collapse increasing sharply as $w$ increases. **Right (ACE):** By applying the adaptive bump correction, ACE guarantees $C(t) > 0$ for all trajectories, restoring valid probability paths even in high-guidance regimes where baselines fail. Note that the y-axis is clipped to $[-20, 100]$ for readability; trajectories that diverge to extreme values or singularities near $t \to 1$ are cut off once they exit the visual frame, as practical sampling terminates at $t_{\text{end}} < 1$ (see Theorem B.3).

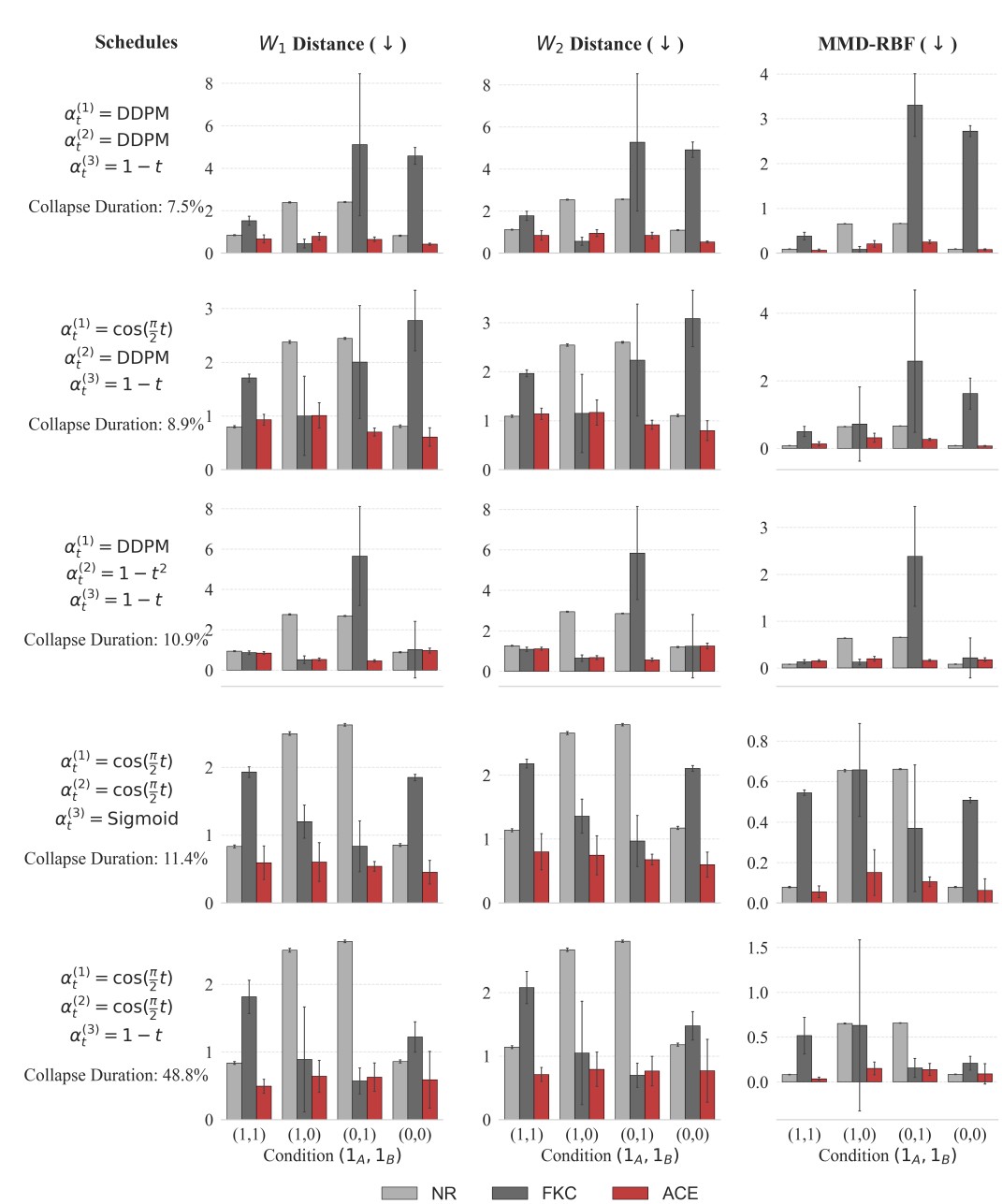

Figure E.6: **Quantitative evaluation of Heterogeneous Ratio-of-Densities Sampling.** We evaluate on a checkerboard distribtuion $(x, y) \sim p_{\texttt{Checker}}[-4, 4]$ subject to constraints $A = \{x \geq 0\}, B = \{x + y \geq 0\}$. The $x$-axis denotes the conditioning configuration $(\mathbf{1}_A, \mathbf{1}_B)$. We consider the case of combining three expert models $q_t^{(1)}, q_t^{(2)}, q_t^{(3)}$ to form the heterogeneous ratio-of-densities $h_t = q_t^{(1)} q_t^{(2)} / q_t^{(3)}$. The rows correspond to configurations of common noise schedules which induce path collapse, where the labels denote the noise schedule $\alpha_t^{(i)}$ assigned to expert $i$. Across varying conditions and metrics $(W_1, W_2, \text{MMD})$, **ACE (red)** consistently outperforms the baselines **NR** and **FKC** (gray). This performance gap reflects the practical benefits of ACE in generating samples across a guaranteed generative path, whereas baselines suffer from path collapse in heterogeneous schedule scenarios. All results are evaluated across 5 seeds with 10k samples, 1k diffusion steps, ESS threshold 0.7, and Bump 30 for ACE. See Figure. E.5 for the corresponding criterion plot.

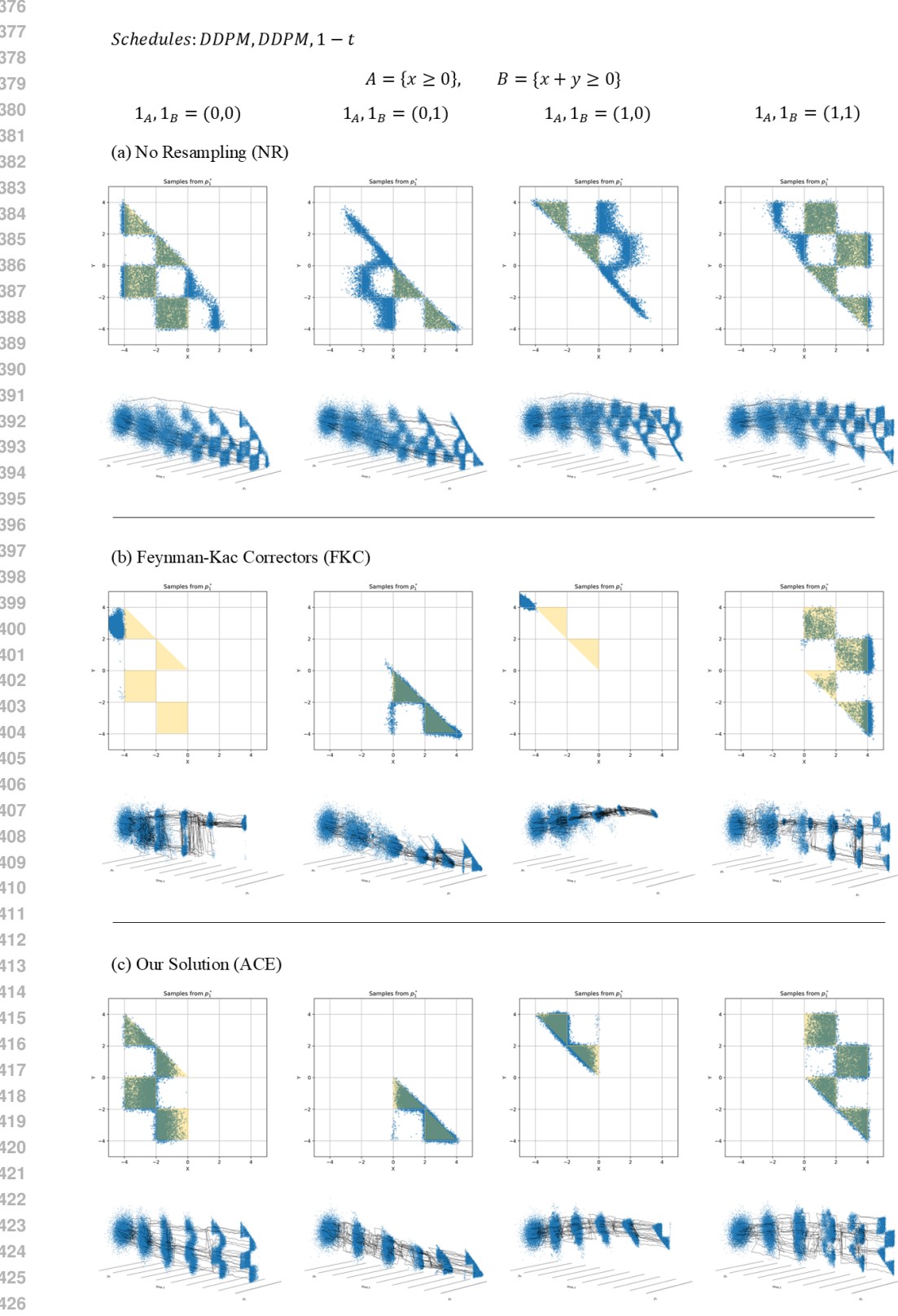

Figure E.7: **Visualization of generative trajectories and final samples.** We target the composite density $p^*(x, y) \propto p^{(1)}(x, y \mid B)p^{(2)}(x \mid A)/p^{(3)}(x)$ using the heterogeneous schedule configuration $(\alpha_t^{(1)}, \alpha_t^{(2)}, \alpha_t^{(3)}) = (\texttt{DDPM}, \texttt{DDPM}, 1 - t)$ and sampling with NR, FKC, and ACE.

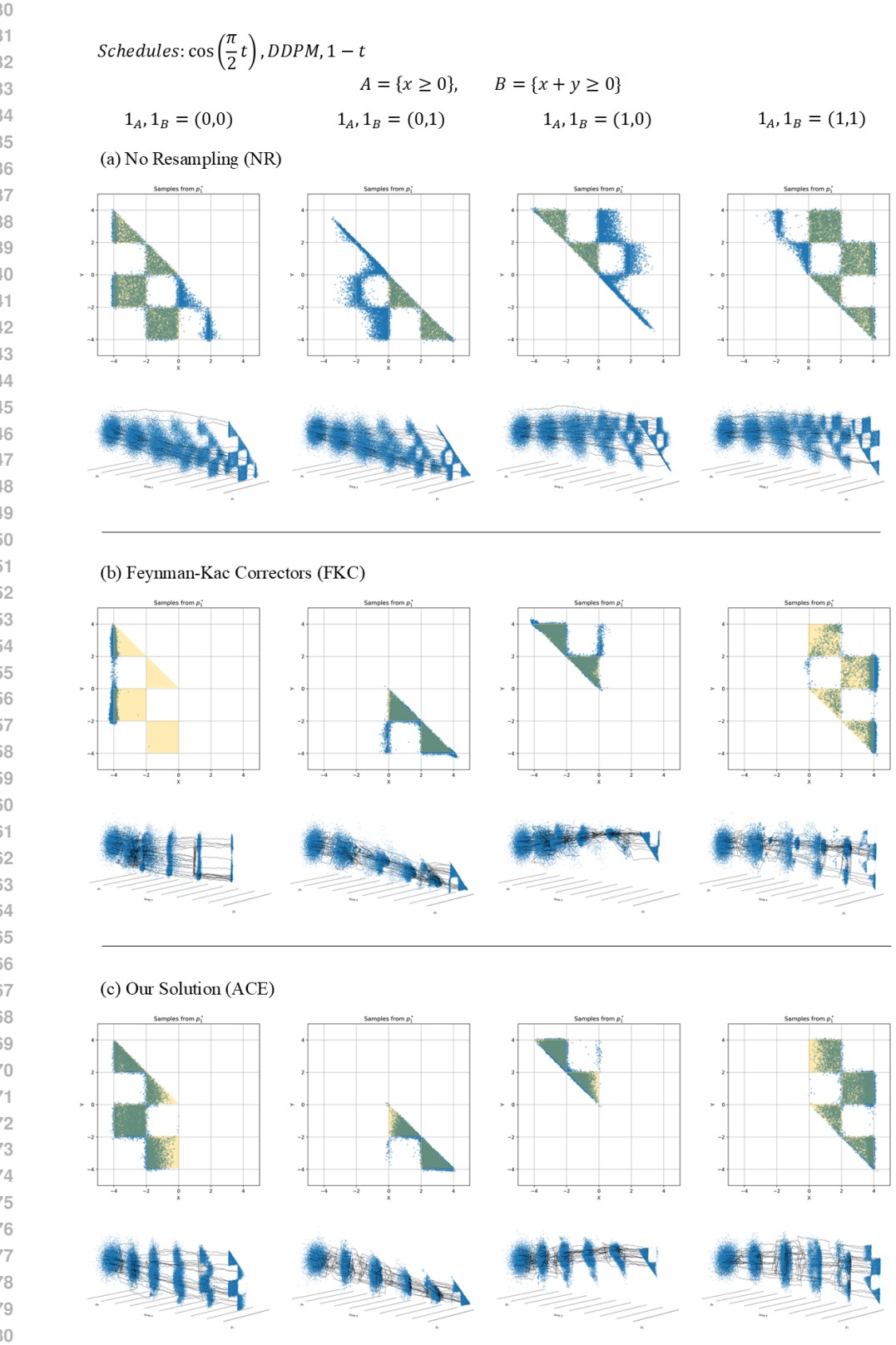

Figure E.8: **Visualization of generative trajectories and final samples.** We target the composite density $p^*(x, y) \propto p^{(1)}(x, y \mid B)p^{(2)}(x \mid A)/p^{(3)}(x)$ using the heterogeneous schedule configuration $(\alpha_t^{(1)}, \alpha_t^{(2)}, \alpha_t^{(3)}) = (\cos(\frac{\pi}{2}t), \text{DDPM}, 1 - t)$ and sampling with NR, FKC, and ACE.

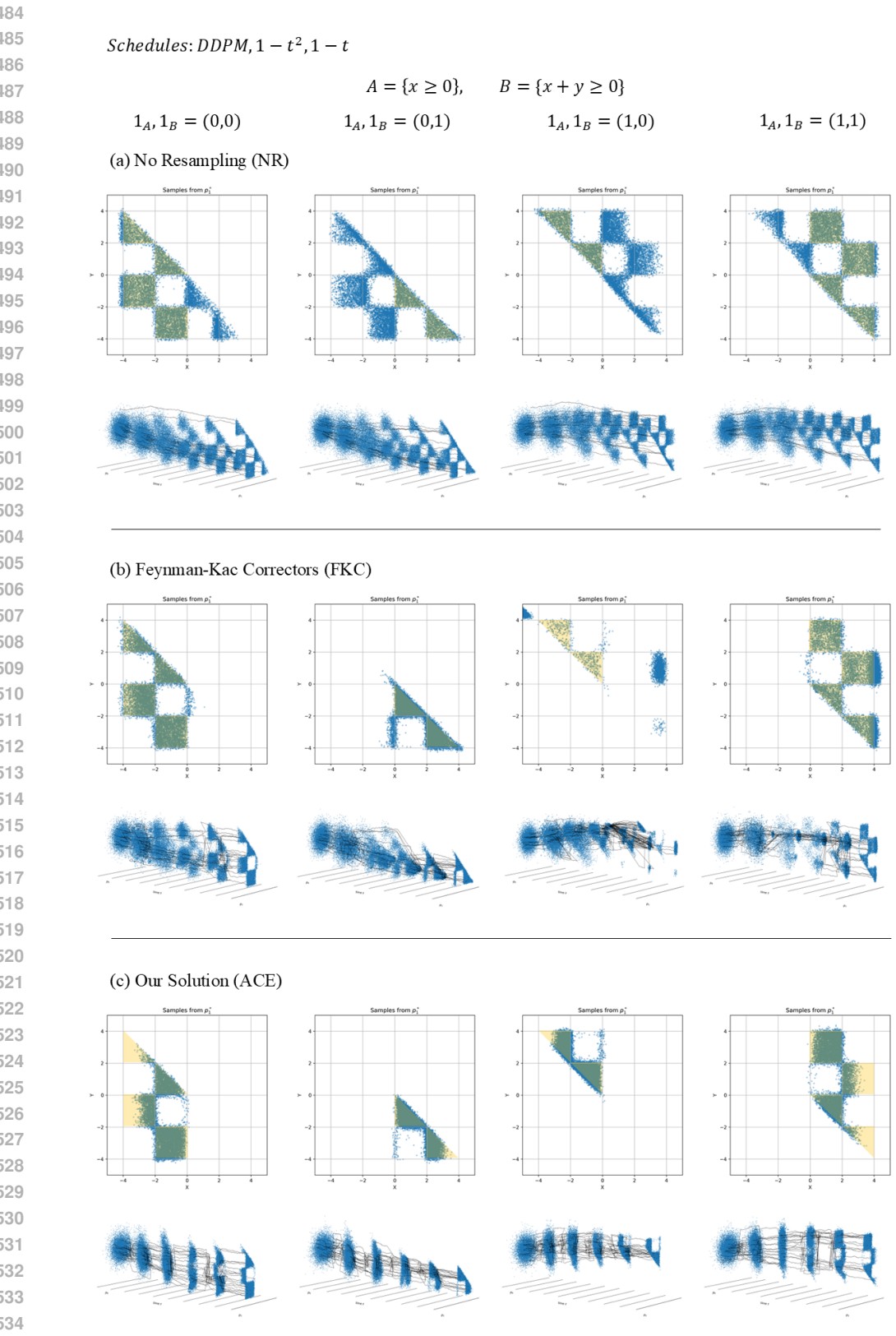

Figure E.9: **Visualization of generative trajectories and final samples.** We target the composite density $p^*(x, y) \propto p^{(1)}(x, y \mid B)p^{(2)}(x \mid A)/p^{(3)}(x)$ using the heterogeneous schedule configuration $(\alpha_t^{(1)}, \alpha_t^{(2)}, \alpha_t^{(3)}) = (\text{DDPM}, 1 - t^2, 1 - t)$ and sampling with NR, FKC, and ACE.

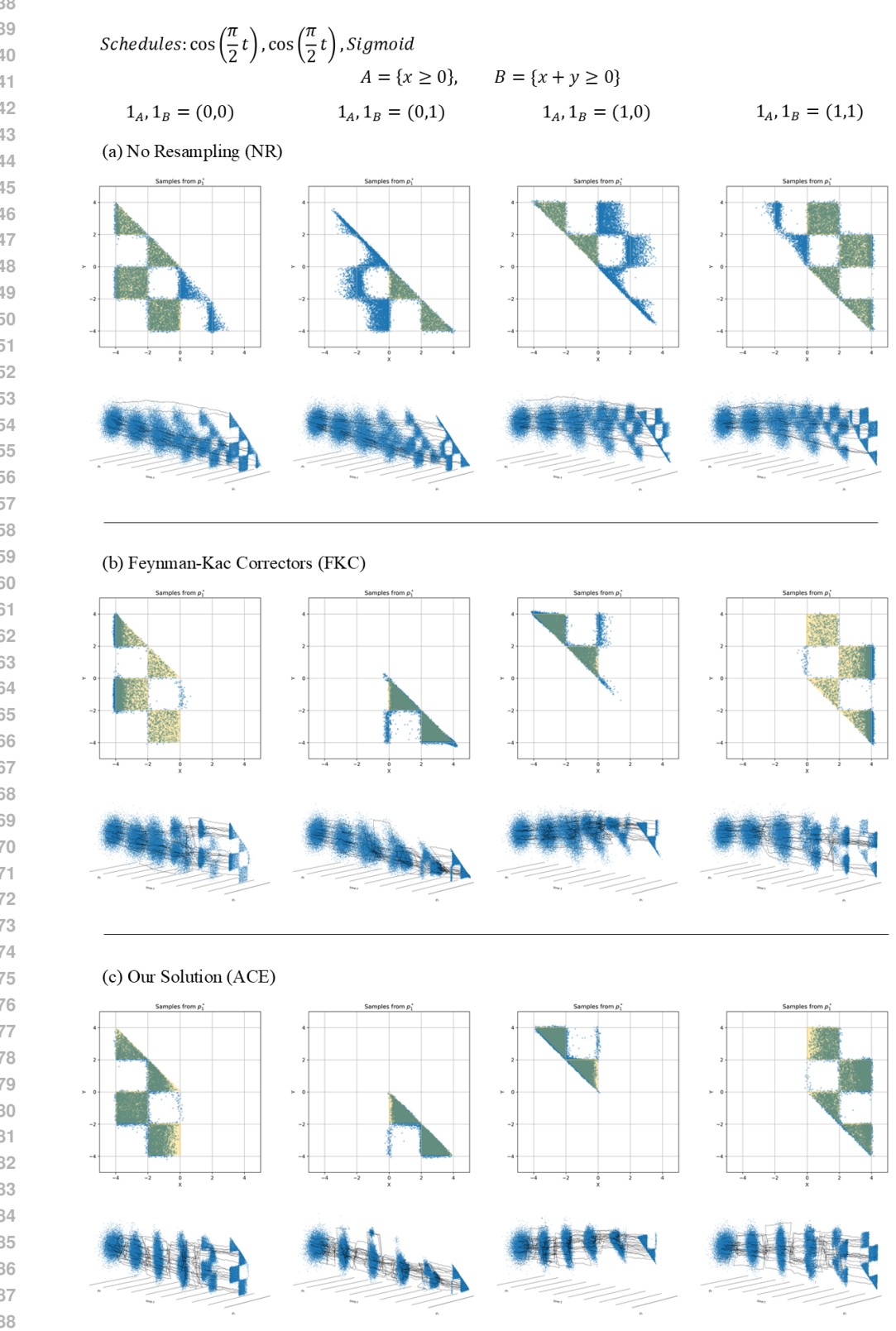

Figure E.10: **Visualization of generative trajectories and final samples.** We target the composite density $p^*(x, y) \propto p^{(1)}(x, y \mid B)p^{(2)}(x \mid A)/p^{(3)}(x)$ using the heterogeneous schedule configuration $(\alpha_t^{(1)}, \alpha_t^{(2)}, \alpha_t^{(3)}) = (\cos(\frac{\pi}{2}t), \cos(\frac{\pi}{2}t), \texttt{Sigmoid})$ and sampling with NR, FKC, and ACE.

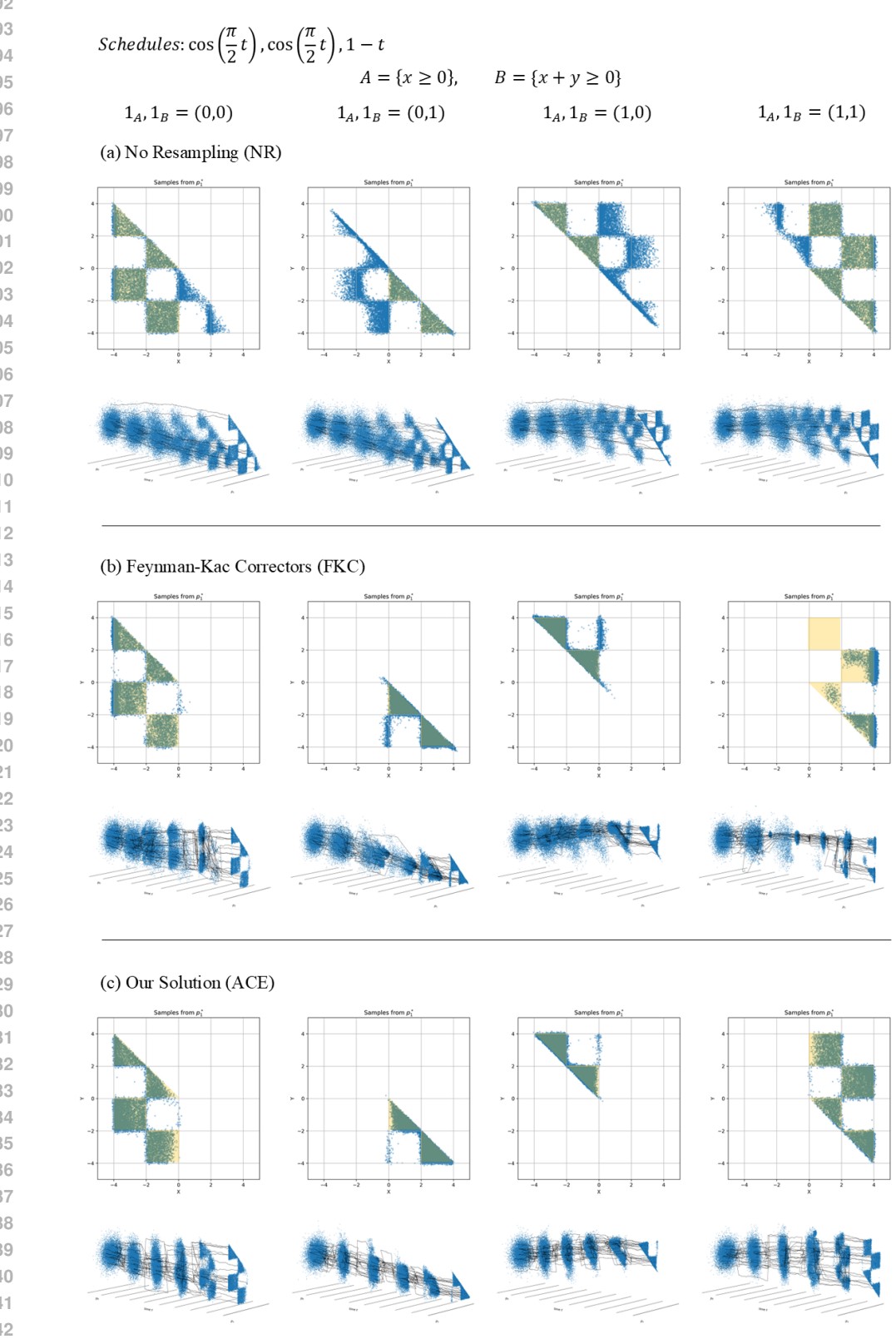

Figure E.11: **Visualization of generative trajectories and final samples.** We target the composite density $p^*(x,y) \propto p^{(1)}(x,y \mid B)p^{(2)}(x \mid A)/p^{(3)}(x)$ using the heterogeneous schedule configuration $(\alpha_t^{(1)}, \alpha_t^{(2)}, \alpha_t^{(3)}) = (\cos(\frac{\pi}{2}t), \cos(\frac{\pi}{2}t), 1 - t)$ and sampling with NR, FKC, and ACE.

### E.3 NECESSITY OF TASK-SPECIFIC NOISE SCHEDULERS

Reviewer naNy

A growing body of work on task-specific noise scheduling shows that different generative tasks require different allocations of noise across the denoising trajectory. This follows from the view that diffusion operates in two regimes, high-noise for global exploration and low-noise for local refinement, whose relative importance varies by task. We further validate this trend in molecular generation, and note that recent advances in learnable or adaptive noise schedulers also reinforce this perspective. Together, these observations motivate the use of adaptive or customized noise schedulers, and consequently, heterogeneous scheduler combinations.

Table E.7: Comparison of noise regimes and average regime statistics per tasks in Tables E.2,E.3,E.4. The comparison is based on the literature perspective (Choi et al., 2022; Rothchild et al., 2023; Chen, 2023; Pavlova & Wei, 2025). Asterisk(*) indicates significant difference from DN ($p < 0.001$). $|\mathcal{R}_H|$, $|\mathcal{R}_L|$ are the lengths of two regimes.

| (A) Comparison of High-Noise vs. Low-Noise Regimes (Choi et al., 2022; Rothchild et al., 2023; Chen, 2023; Pavlova & Wei, 2025) | | | |
|---|---|---|---|
| **Regime** | **Dynamics** | **Feature Scale Priority** | **Role in Denoising** |
| $\mathcal{R}_H$ **(High-Noise)** | Large Transitions | Global structure | Exploration |
| $\mathcal{R}_L$ **(Low-Noise)** | Small corrective steps | Local fine-grained details | Refinement |

| (B) Average Noise Regime Statistics per Molecular Tasks in Tables E.2,E.3,E.4 | | | | | |
|---|---|---|---|---|---|
| **Task** | **Global vs Local** | **Cond. vs Uncond.** | **Exploration vs Local Refinement** | $|\mathcal{R}_H|$ | $|\mathcal{R}_L|$ | $|\mathcal{R}_H|/|\mathcal{R}_L|$ |
| **DN** | Global | Uncond. | Exploration | 0.53 | 0.47 | 1.11 |
| **CONF** | Local | Cond. | Local Refinement | 0.47* | 0.53* | 0.89* |
| **SBDD** | Local | Cond. | Local Refinement | 0.48* | 0.52* | 0.91* |

**High-noise vs. low-noise regimes.** Recent analyses (Choi et al., 2022; Rothchild et al., 2023; Chen, 2023; Pavlova & Wei, 2025) in Table E.9 characterize two complementary denoising phases. The **high-noise regime** ($\mathcal{R}_H$) performs large transitions that support **global exploration** and formation of coarse structure. The **low-noise regime** ($\mathcal{R}_L$) reduces to **small corrective steps** that refine **local, fine-grained features**. Effective scheduling therefore requires balancing these phases according to task needs. This distinction is summarized in Table E.7(A).

**Emperically validated trends in conditional vs. unconditional molecular generation.** As shown in (Chen, 2023), the roles of $\mathcal{R}_H$ and $\mathcal{R}_L$ indicate that *larger images benefit from a longer $\mathcal{R}_H$*, reflecting their need for stronger exploratory behavior. The insights naturally transfer to molecules. Unconditional generation (DN) needs to explore many possible molecular shapes, so it naturally relies more on the broader, exploratory behavior of $\mathcal{R}_H$. Conditional tasks (CONF, SBDD), however, must fit specific structural requirements, and therefore gain more from the precise, detail-oriented corrections that occur in $\mathcal{R}_L$.

We empirically confirm that existing molecular schedulers adhere to this pattern (Tables E.2, E.3, E.4). Following Pavlova & Wei (2025), we divide noise phases using $\mathrm{SNR} \geq 1$ ($\mathcal{R}_H$) and $\mathrm{SNR} < 1$ ($\mathcal{R}_L$). For each model, we compute the interval lengths $|\mathcal{R}_H|$, $|\mathcal{R}_L|$, and their ratio, then average across models per task. The results in Table E.7(B) show clear trends: DN favors $\mathcal{R}_H$, whereas CONF and SBDD prioritize $\mathcal{R}_L$.

These results reinforce our central claim: *heterogeneous noise schedulers are not only reasonable but necessary* for the molecular tasks considered in Section E.2.

**Recent trends in task-specific scheduler design.** Recent works further emphasize that optimal schedules depend on task characteristics (Table E.8). Image models adopt learned or adaptive schedulers (Sahoo et al., 2024), while molecular models employ component-specific or trajectory-based schedules (Vignac et al., 2023; Seo et al., 2025). These methods directly challenge the assumption of a universal, fixed schedule.

**Foundational regime analysis.** Table E.9 compiles key works formalizing these roles. The low-noise regime ($\mathcal{R}_L$) is consistently identified as the phase responsible for *precise, fine-scale refinement*, while the high-noise regime ($\mathcal{R}_H$) drives *exploration and diversity*. These findings directly support task-specific allocations of denoising effort.

Table E.8: References on Task-Specific and Adaptive Scheduling

| Reference | Contribution |
| --- | --- |
| Vignac et al. (2023) | Introduces component-specific molecular noise scheduling. |
| Lee et al. (2024) | Presents a fully data-driven adaptive scheduler for time-series diffusion. |
| Sahoo et al. (2024) | Develops MuLAN: learned multivariate adaptive noise processes. |
| Seo et al. (2025) | Proposes per-element optimized forward trajectories for molecules. |
| Sorokin et al. (2025) | Demonstrates adaptive allocation between $\mathcal{R}_H$ (exploration) and $\mathcal{R}_L$ (refinement). |
| Choi et al. (2025) | Uses distinct channel-wise schedules to balance the diversity–accuracy trade-off. |

Table E.9: Foundational References for Noise Regime Mechanism

| Reference | Contribution |
| --- | --- |
| Choi et al. (2022) | Identifies that certain noise levels offer a proper pretext task for the model to learn rich visual concepts. |
| Rothchild et al. (2023) | Anaylze the diffusion dynamics in molecular generative models. |
| Chen (2023) | Shows that optimal schedules vary with task and resolution. |
| Pavlova & Wei (2025) | Identifies $\mathcal{R}_L$ as the phase in which precision and fine-scale structure dominate. |

## E.4   ACE BEYOND COLLAPSE: COMPOSITIONAL IMAGE GENERATION IN THE HOMOGENEOUS REGIME

In the main text, ACE is used primarily to *repair* heterogeneous compositions where the path-existence criterion fails and Marginal Path Collapse occurs. In this section, we show that the same theory also yields *gains beyond collapse avoidance* on a compositional image generation benchmark, even in a strictly homogeneous setting where $C(t) > 0$ everywhere.

Reviewer naNy

**Tail concentration in the homogeneous $\alpha_t$ regime.** Even when the path-existence criterion $C(t) > 0$ holds everywhere (i.e., no collapse), the coefficient $C(t)$ still controls how tightly the composed distribution $p_t^*$ is concentrated. The following result summarizes this dependence and motivates why time-varying exponents can improve sampling quality even in the homogeneous regime.

---

**Proposition E.1** (Tail control from a quadratic envelope). *Let $\{h_t\}_{t \in [0,1]}$ be a family of non-negative functions on $\mathbb{R}^d$ and let $p_t^*(x) := h_t(x)/Z_t$ be well defined, so $Z_t := \int_{\mathbb{R}^d} h_t(x)\,dx < \infty$ for all $t \in [0,1]$. Suppose there exist constants $m_* > 0$, $K > 0$, $B > 0$, and a function $C : [0,1) \to (0,\infty)$ such that $Z_t \geq m_*$ for all $t \in [0,1]$ and*

$$h_t(x) \leq K \exp\left(-\frac{1}{2} C(t)\,\|x\|^2 + B\|x\|\right) \quad \textit{for all } x \in \mathbb{R}^d,\ t \in [0,1].$$

*Then there exist constants $K_0 > 0$ and $R_0 > 0$, independent of $t$, such that for all $t$ with $C(t) > 0$ and all $R \geq R_0$,*

$$\mathbb{P}_{X \sim p_t^*}\left(\|X\| > R\right) \ \leq\ K_0\,\exp\left(\frac{B^2}{2C(t)}\right) C(t)^{-\frac{d}{2}}\,\exp\left(-\frac{1}{16} C(t)\,R^2\right). \tag{E.15}$$

*Consequently, for any $\varepsilon \in (0,1)$, the $(1-\varepsilon)$–quantile radius*

$$R_t(\varepsilon) := \inf\{R > 0 : \mathbb{P}(\|X\| \leq R) \geq 1 - \varepsilon\}$$

*satisfies*

$$R_t(\varepsilon) \ \leq\ R_0\,\vee\,\sqrt{\frac{16}{C(t)}\left[\log\frac{K_0}{\varepsilon} + \frac{B^2}{2C(t)} - \frac{d}{2}\log C(t)\right]}. \tag{E.16}$$

*In particular, for fixed $\varepsilon$ and $d$, larger values of $C(t)$ yield tighter tails and smaller effective radii $R_t(\varepsilon)$ for $p_t^*$, up to logarithmic corrections.*

---

*Proof.* Fix $t$ with $C(t) > 0$. Complete the square yields:

$$h_t(x) \ \leq \ K \exp\Big(\frac{B^2}{2C(t)}\Big) \exp\Big(-\frac{1}{2}C(t)\Big(\|x\| - \frac{B}{C(t)}\Big)^2\Big).$$

Let $C_{\min} := \inf_{s \in [0,1]} C(s) > 0$, which is positive on any compact subinterval where $C(t) > 0$. Choose $R_0 \geq 2B/C_{\min}$ such that for all $t$ and all $R \geq R_0$ we have $B/C(t) \leq B/C_{\min} \leq R_0/2 \leq R/2$. Then, for any $R \geq R_0$ and any $x$ with $\|x\| \geq R$,

$$\|x\| - \frac{B}{C(t)} \ \geq \ \frac{\|x\|}{2},$$

and thus

$$h_t(x) \ \leq \ K \exp\Big(\frac{B^2}{2C(t)}\Big) \exp\Big(-\frac{1}{8}C(t)\|x\|^2\Big).$$

Integrating over $\{\|x\| > R\}$ in radial coordinates ($dx = S_d r^{d-1} dr$) yields

$$\int_{\|x\|>R} h_t(x)\, dx \ \leq \ K \exp\Big(\tfrac{B^2}{2C(t)}\Big) \int_{r=R}^{\infty} \exp\Big(-\tfrac{1}{8}C(t)\, r^2\Big) S_d r^{d-1} dr.$$

By the change of variables $y = \sqrt{C(t)/8}\, r$, and the Gaussian tail bound $\mathbb{P}(z > s) \leq 2\exp(-s^2/2)$ for $z \sim \mathcal{N}(0,1)$ (Ross et al., 1998), there exists a constant $C_d > 0$ (depending only on $d$) such that

$$\int_{r=R}^{\infty} \exp\Big(-\tfrac{1}{8}C(t)\, r^2\Big) S_d r^{d-1} dr \ \leq \ C_d\, C(t)^{-\frac{d}{2}} \ \exp\Big(-\tfrac{1}{16}C(t)\, R^2\Big).$$

Hence

$$\int_{\|x\|>R} h_t(x)\, dx \ \leq \ KC_d \exp\Big(\frac{B^2}{2C(t)}\Big) C(t)^{-\frac{d}{2}} \ \exp\Big(-\frac{1}{16}C(t)\, R^2\Big).$$

By assumption, $Z_t \geq m_* > 0$ uniformly in $t$, so

$$\mathbb{P}_{p_t^*}(\|X\| > R) = \frac{1}{Z_t} \int_{\|x\|>R} h_t(x)\, dx \ \leq \ \frac{KC_d}{m_*} \exp\Big(\frac{B^2}{2C(t)}\Big) C(t)^{-\frac{d}{2}} \ \exp\Big(-\frac{1}{16}C(t)\, R^2\Big).$$

Setting $K_0 := KC_d/m_*$ gives the tail bound in Equation E.15. Finally, to get the quantile radius, it suffices to enforce the right-hand side to be at most $\varepsilon$ and solve for $R$. This yields

$$R^2 \ \geq \ \frac{16}{C(t)}\Big[\log\frac{K_0}{\varepsilon} + \frac{B^2}{2C(t)} - \frac{d}{2}\log C(t)\Big],$$

and taking $R \geq R_0$ proves the bound in Equation E.16. $\square$

**Application to heterogeneous ratio-of-densities.** In the setting of Theorem B.1, each lifted expert $\tilde{q}_t^{(i)}$ admits upper bounds of the form

$$\tilde{q}_t^{(i)}(x) \ \leq \ C_{+,i} \ \exp\Big(-\tfrac{1}{2}\sum_{k \in I_i} a_{i,k}(t)\, x_k^2 + B_i\|x\|\Big),$$

with $a_{i,k}(t) \geq 0$ and constants $C_{+,i}, B_i > 0$ independent of $t$. Multiplying these bounds and raising to the exponents $\gamma_i(t)$ yields

$$h_t(x) \ \leq \ K \exp\Big(-\tfrac{1}{2}\sum_{k=1}^{d} C_k(t)\, x_k^2 + B\|x\|\Big),$$

where $K, B > 0$ are uniform constants and $C_k(t)$ are precisely the coordinate-wise coefficients from Equation 4. Defining $C(t) := \min_k C_k(t)$, we have $\sum_k C_k(t)x_k^2 \geq C(t)\|x\|^2$, so the envelope matches the form in Proposition E.1. Theorem B.1 also implies that $h_t \in L^1(\mathbb{R}^d)$ and that the normalizing constants $Z_t$ are uniformly bounded below on any compact set where $C_k(t) > 0$. Therefore, all assumptions of Proposition E.1 hold for our heterogeneous ratio-of-densities path $p_t^* = h_t/Z_t$, and the tail and quantile-radius bounds apply directly with this choice of $C(t)$.

This proposition shows that even when $C(t) > 0$ everywhere (no collapse), the *magnitude* of $C(t)$ still governs how concentrated $p_t^*$ is: for fixed $\varepsilon$, the radius $R_t(\varepsilon)$ shrinks as $C(t)$ grows (up to logarithmic factors). Thus, increasing $C(t)$ at intermediate times (for example, via the time-varying exponents of ACE) tightens the tails of $p_t^*$ and reduces its effective spatial extent. In practice, this suppresses large intermediate spreads, stabilizes importance weights, and improves sample quality, as we observe both in the 1D ratio-of-Gaussians trajectory and image experiment described below.

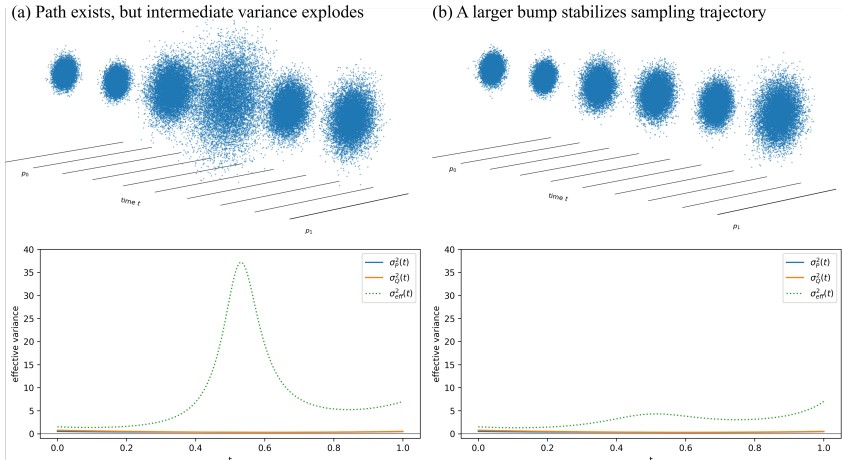

Figure E.12: **Stabilizing a ratio-of-Gaussians path via the bump parameter.** We illustrate the effect of the path-existence coefficient $C(t)$ on a 1D ratio-of-Gaussians example (Eq. 3). (a) With a small bump applied to $q_t^{(1)}$, $C(t) > 0$ but dips close to zero around $t = 0.5$, leading to a very broad intermediate density: the effective variance (green curve) exhibits a peak, and particle trajectories become highly dispersed. This behavior is consistent with Proposition E.1, which shows that small $C(t)$ yields weak tail decay and large quantile radii. (b) Increasing the bump parameter raises the lower bound of $C(t)$ throughout the trajectory, suppressing the peak in variance and producing a much more confined intermediate distribution. The resulting path smoothly transports samples from the Gaussian prior to the compactly supported target without large variance excursions.

---

**Example E.4** (Compositional Image Generation). We consider compositional text-to-image generation with region-wise prompts and bounding boxes. The task is generating an image $X$ conditioned on $n$ region-specific object prompts $c_{1:n}$ (with foreground masks $F_{1:n}$) and a global context prompt $C$. The target distribution factorizes via Bayes' rule into a product of expert likelihoods:

$$p(X \mid c_{1:n}, C) = \frac{1}{Z} \underbrace{p(X \mid C)^{\gamma_0(t)}}_{\substack{\text{Global} \\ \text{Consistency}}} \prod_{i=1}^{n} \underbrace{\left( \frac{p(F_i \mid c_i)}{p(F_i)} \right)^{\gamma_i(t)}}_{\substack{\text{Local} \\ \text{Specifics}}} \qquad (\text{E.17})$$

**ACE Implementation with a Single Backbone:** Crucially, we approximate all terms using a *single* pretrained text-to-image model (Stable Diffusion (Rombach et al., 2021) v1.5 / v2.1).

- $p_\theta(X \mid C)$: the base model conditioned on the global prompt.

- $p_\theta(F_i \mid c_i)$: the same model, applied to the cropped region $F_i$ (via masking) with prompt $c_i$.

- $p_\theta(F_i)$: the model on $F_i$ with a null prompt.

All experts share architecture and noise schedule, placing us in the homogeneous regime of Theorem 2.1, where the coefficient $C(t)$ remains strictly positive and no Marginal Path Collapse occurs. In this limit, path existence is guaranteed even for constant exponents.

**ACE vs. FKC and NR in the homogeneous regime.** We apply our generic ACE sampler (Algorithm 1) to this setup by treating each region-specific term in E.17 as an expert and composing their scores and drifts as in Section 2. Since the schedules are homogeneous, ACE does not need to "rescue" a collapsing path. Instead, we use a small bump $B = 5$ in the region exponents $\gamma_i(t)$, which increases $C(t)$ locally at intermediate times while preserving the endpoint distributions.

Proposition E.1 implies that this yields a tighter intermediate concentration for $p_t^*$, reducing mass in off-manifold regions and stabilizing importance weights.

We compare three steering schemes using the same backbone, schedules, and hyperparameters[3]:

- **NR (CFG-like):** score-difference heuristic without importance weights.
- **FKC:** constant exponents with Feynman–Kac weighting and resampling (Algorithm 2).
- **ACE:** time-varying exponents with a bump $B = 5$ (Algorithm 1).

Because $C(t) > 0$ everywhere, FKC and ACE both operate on valid probability paths; the only difference is the exponent schedule (i.e., constant vs. time-varying).

**Comparison with Existing Paradigms.** Existing approaches to this task generally fall into three categories: (1) Methods that train adapters like GLIGEN (Li et al., 2023b), Make-It-Count (Binyamin et al., 2025); (2) Auxiliary-guided methods like 3DIS (dewei Zhou et al., 2025) that rely on external signals (depth maps, LLMs); and (3) Architectural interventions such as box-layout guidance (Wang et al., 2024c) or attention-editing techniques (Chefer et al., 2023; Qiu et al., 2025) that modify model's internal signals. In contrast, ACE offers a sampling strategy that achieves unbiased modular composition using a single fundamental expert model without requiring retraining, external data, or architectural changes.

**Empirical results on COCO-MIG.** We evaluate on the COCO-MIG benchmark (Zhou et al., 2024b) and report the instance attribute success ratio (%) and mIoU (%) where the instance attribute success ratio measures the fraction of region-wise instructions correctly matched in the generated image. We compare ACE against NR and FKC, multiple diffusion baselines (Rombach et al., 2021; Podell et al., 2023; Esser et al., 2024; Labs, 2024), and task-specific adapters (GLIGEN (Li et al., 2023b), InstanceDiff (Wang et al., 2024b), MIGC (Zhou et al., 2024a), 3DIS (dewei Zhou et al., 2025)). As shown in Table E.10, we observe a consistent hierarchy *NR < FKC < ACE* across backbones: FKC improves substantially over the heuristic NR baseline, and ACE with $B = 5$ further improves both attribute alignment and spatial metrics, despite being applied in a regime with no path collapse. Remarkably, ACE matches or exceeds a task-specific adapter GLIGEN (Li et al., 2023b) on several metrics, while remaining completely training-free.

Qualitatively, Fig. E.13 shows that ACE produces sharply localized objects with correct attributes in their designated boxes, whereas the Stable Diffusion baseline struggles to localize objects or separate attributes. These results provide an empirical complement to Proposition E.1: even when the path-existence criterion is satisfied, carefully shaping the exponent schedule can yield *stronger* conditioning and improved sample quality.

Table E.10: Instance attribute success ratio (%) with the corresponding mIoU (%) shown in parentheses. L2–L6 denote the success ratios for tasks with 2 to 6 region-wise guidance conditions. 'CLIP' and 'Local CLIP' refer to CLIP scores computed on the entire image versus the full prompt, and on local crops versus the corresponding local prompts, respectively.

| Method | Backbone | L2 | L3 | L4 | L5 | L6 | Avg | CLIP↑ | Local CLIP↑ |
|---|---|---|---|---|---|---|---|---|---|
| *Base Diffusion Model with global prompt* | | | | | | | | | |
| SD1.5 | SD1.5 | 5.59 (18.83) | 4.79 (17.43) | 2.83 (14.95) | 2.41 (13.93) | 2.21 (15.94) | 3.11 (15.75) | 24.64 | 18.36 |
| SDXL | SDXL | 5.59 (19.78) | 4.48 (18.54) | 2.81 (16.67) | 2.14 (15.72) | 2.80 (18.42) | 3.17 (17.55) | 25.71 | 18.63 |
| SD3.5-M | SD3.5-M | 8.05 (21.57) | 8.46 (21.37) | 6.07 (18.98) | 5.02 (17.39) | 4.40 (17.80) | 5.86 (18.85) | 26.41 | 18.77 |
| Flux.1-dev | Flux.1-dev | 8.83 (22.00) | 7.50 (20.93) | 4.86 (17.75) | 4.53 (16.77) | 3.22 (16.49) | 5.08 (18.03) | 26.17 | 18.56 |
| *Inference-time Control using **only** the base Diffusion Model* | | | | | | | | | |
| NR | SD1.5 | 24.61 (30.50) | 24.19 (29.81) | 19.36 (25.64) | 17.67 (24.42) | 19.54 (25.91) | 20.24 (26.53) | 24.96 | 20.21 |
| FKC | SD1.5 | 33.02 (34.43) | 31.18 (33.34) | 28.80 (31.53) | 25.46 (29.63) | 22.88 (28.61) | 28.27 (31.51) | 25.44 | 20.66 |
| **ACE ($B$=5)** | SD1.5 | 45.31 (41.48) | 42.50 (40.60) | 36.25 (35.20) | 32.75 (32.66) | 32.40 (33.40) | 37.84 (36.67) | 25.59 | 21.18 |
| NR | SD2.1 | 26.52 (31.18) | 25.76 (30.50) | 20.94 (27.24) | 18.83 (24.77) | 18.78 (25.31) | 21.04 (26.93) | 24.89 | 19.96 |
| FKC | SD2.1 | 39.69 (37.78) | 34.93 (35.91) | 31.98 (33.95) | 28.42 (32.07) | 24.93 (30.02) | 31.99 (33.95) | 25.27 | 20.64 |
| **ACE ($B$=5)** | SD2.1 | 46.25 (42.24) | 41.04 (39.45) | 41.72 (37.96) | 38.38 (37.17) | 34.90 (34.82) | 40.46 (38.33) | 24.94 | 21.09 |
| *Adapter rendering methods (Requires Additional Training on new data or External Models)* | | | | | | | | | |
| GLIGEN | SD1.4 | 38.36 (33.96) | 32.79 (29.58) | 28.67 (25.95) | 25.02 (23.88) | 26.98 (24.93) | 28.84 (26.47) | 24.91 | 20.78 |
| InstanceDiffusion | SD1.5 | 68.24 (62.67) | 60.47 (55.75) | 59.88 (54.15) | 53.92 (49.02) | 57.14 (51.34) | 58.49 (53.12) | 25.97 | 21.90 |
| MIGC | SD1.4 | 66.37 (57.02) | 63.10 (54.47) | 61.27 (52.48) | 57.25 (49.49) | 59.13 (51.38) | 60.41 (52.16) | 25.39 | 21.42 |
| 3DIS | SD1.5 | 58.09 (52.76) | 51.48 (46.92) | 46.15 (42.46) | 40.39 (38.16) | 41.22 (38.47) | 45.23 (41.89) | 24.02 | 21.24 |

---

[3]For the image experiments, it sufficed to resample once at $t_s = 0.3$ due to a small batchsize of $N = 3$.

A photo of a red chair and a yellow chair and a white teddy bear and a brown dining table

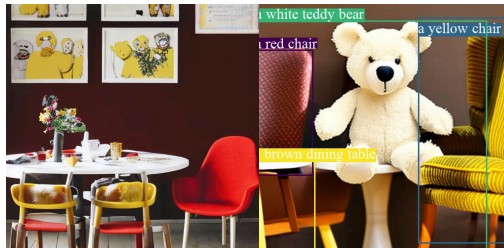

A photo of a black cup and a black donut and a green dining table and a blue donut

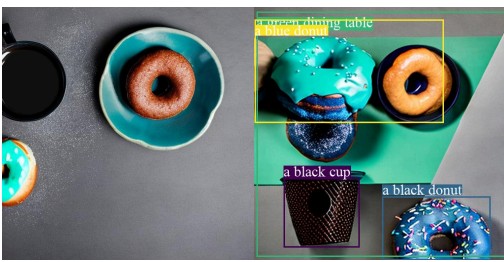

A photo of a blue boat and a red boat and a white boat and a green boat

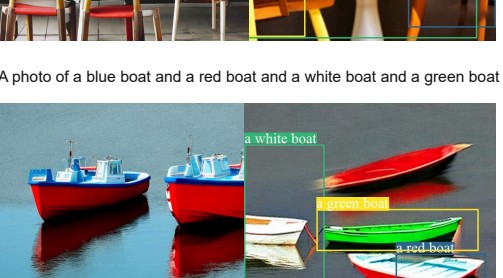

A photo of a white couch and a yellow dining table and a white oven and a brown chair

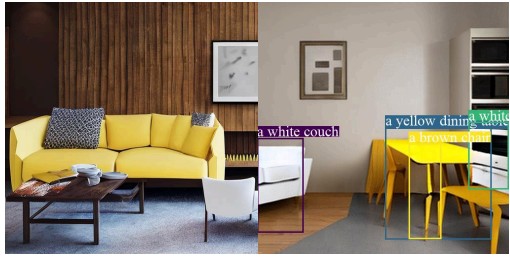

A photo of a brown airplane and a red airplane and a blue airplane

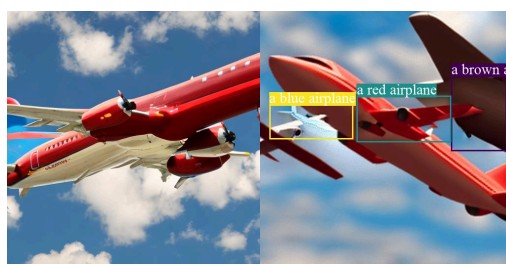

A photo of a white bird and a red boat and a blue boat

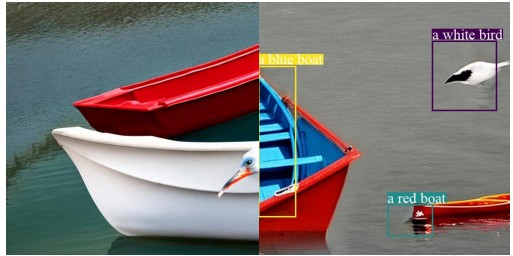

A photo of a blue dog and a white bed and a green tie

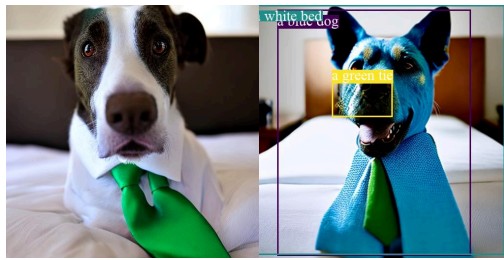

A photo of a black dog and a blue dog and a brown dog

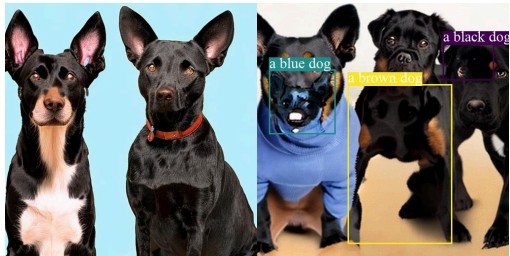

A photo of a white umbrella and a red umbrella

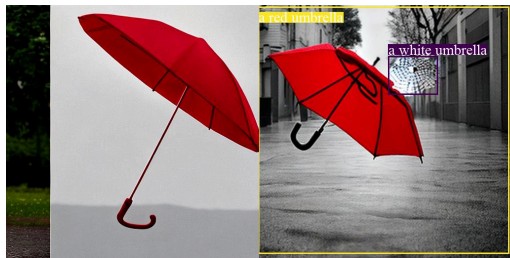

A photo of a brown boat and a brown boat and a yellow boat and a black boat and a white boat and a green boat

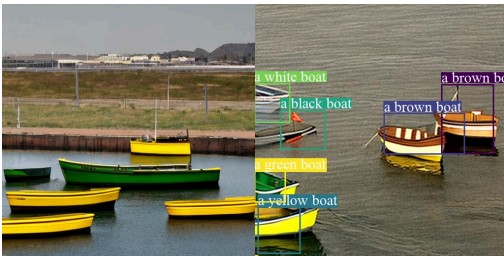

Figure E.13: Qualitative results of compositional image generation with ACE. Compared to the base Stable Diffusion 2.1 model (left), simulating the same model with ACE (right) yields better prompt alignment and layout guidance – completely without additional training or external models.

Table E.11: Runtime per image and peak GPU memory usage for generating a single image. Lower is better. Baselines were evaluated on an NVIDIA A100 GPU, whereas ACE and FKC were evaluated on an NVIDIA A6000. Since the A100 is approximately $\times 2.3$ faster than the A6000, the raw runtime gap between ACE, FKC and baselines is narrower than it appears.

| Method | Backbone | Time (s) ↓ | | | | | | VRAM (GB) ↓ | | | | | |
|---|---|---|---|---|---|---|---|---|---|---|---|---|---|
| | | L2 | L3 | L4 | L5 | L6 | Avg | L2 | L3 | L4 | L5 | L6 | Avg |
| SD1.5 | SD1.5 | 1.48 | 1.48 | 1.54 | 1.52 | 1.52 | 1.51 | 2.64 | 2.64 | 2.64 | 2.64 | 2.64 | 2.64 |
| NR | SD1.5 | 5.79 | 7.24 | 8.65 | 10.25 | 11.41 | 8.67 | 5.22 | 5.22 | 5.22 | 5.30 | 5.56 | 5.31 |
| FKC (A6000) | SD1.5 | 14.87 | 19.14 | 23.16 | 27.11 | 30.89 | 23.04 | 3.80 | 3.80 | 3.91 | 4.38 | 4.85 | 4.15 |
| **ACE** (A6000) | SD1.5 | 17.44 | 22.61 | 28.14 | 33.29 | 34.72 | 27.24 | 4.71 | 4.71 | 4.72 | 4.72 | 4.72 | 4.72 |
| SD2.1 | SD2.1 | 1.47 | 1.48 | 1.47 | 1.62 | 1.48 | 1.50 | 3.05 | 3.05 | 3.05 | 3.05 | 3.05 | 3.05 |
| NR | SD2.1 | 5.41 | 6.51 | 7.89 | 9.43 | 10.37 | 7.93 | 6.05 | 6.05 | 6.05 | 6.13 | 6.39 | 6.13 |
| FKC (A6000) | SD2.1 | 14.20 | 18.94 | 20.64 | 24.23 | 27.55 | 21.11 | 4.22 | 4.22 | 4.38 | 4.86 | 5.35 | 4.70 |
| **ACE** (A6000) | SD2.1 | 16.53 | 21.34 | 25.82 | 30.72 | 32.19 | 23.52 | 5.12 | 5.13 | 5.13 | 5.13 | 5.13 | 5.13 |
| SDXL | SDXL | 3.45 | 3.36 | 3.29 | 3.36 | 3.38 | 3.37 | 8.98 | 8.98 | 8.98 | 8.98 | 8.98 | 8.98 |
| SD3.5-M | SD3.5-M | 4.56 | 4.55 | 4.55 | 4.55 | 4.55 | 4.55 | 17.61 | 17.61 | 17.61 | 17.61 | 17.61 | 17.61 |
| Flux.1-dev | Flux.1-dev | 47.73 | 47.69 | 47.71 | 47.71 | 47.71 | 47.71 | 67.64 | 67.64 | 67.64 | 67.64 | 67.64 | 67.64 |
| GLIGEN | SD1.4 | 2.61 | 2.62 | 2.62 | 2.62 | 2.61 | 2.62 | 6.01 | 6.01 | 6.01 | 6.01 | 6.01 | 6.01 |
| InstanceDiffusion | SD1.5 | 6.77 | 8.22 | 9.70 | 11.14 | 12.67 | 9.70 | 6.60 | 6.60 | 6.60 | 6.60 | 6.60 | 6.60 |
| MIGC | SD1.4 | 2.58 | 2.58 | 2.56 | 2.57 | 2.58 | 2.57 | 5.43 | 5.43 | 5.43 | 5.43 | 5.43 | 5.43 |
| 3DIS (A6000) | SD1.5 | 10.56 | 10.84 | 11.03 | 11.14 | 11.38 | 10.99 | 7.55 | 7.55 | 7.55 | 7.55 | 7.55 | 7.55 |

# F FUTURE DIRECTIONS FOR ACE

We conclude by outlining limitations and several promising directions for further development of the ACE framework and heterogeneous model composition, both in theory and in practice.

**Error propagation in model composition.** A natural open question concerns the propagation of errors when composing multiple expert models using ACE. In practice, does the modular composition of experts lead to error accumulation that degrades performance compared to training a single, larger network? A systematic error analysis could clarify whether modularity introduces compounding approximation error or whether the benefits of specialization dominate in practical scenarios.

**Extension to arbitrary transport.** We assumed that the stochastic interpolation is between a Gaussian and compactly supported distribution, which is a common formulation in generative modeling. However, tasks such as molecule or image editing, where a sample from the source distribution is given and the task is to transport that sample to a target distribution, require a different formulation since these models interpolate between two compactly supported distributions. Future work could explore when Marginal Path Collapse occurs in these general transport problems. Reviewer gBQy

**Extension to infinite-dimensional spaces.** Our reasoning was kept as general as possible so that it may be naturally extended to infinite-dimensional settings. Viewing each data sample as a function, rather than a point in finite-dimensional space, is both theoretically appealing and practically relevant in certain generative modeling applications. For instance, by replacing $\mu$ in definition A.1 with a Gaussian measure on a Hilbert space, one can develop a function-space analogue of ACE with direct implications for models defined over functional data.

**Practical efficiency via distillation.** A key limitation of ACE in practice is that the inference cost scales linearly with the number of experts. However, the weighted SDE/ODE formulation of theorem A.1 suggests that existing techniques for model distillation—such as consistency models, flow map matching, or efficient SDE/ODE solvers—could be adapted to mitigate this cost.

*Remark* (Distillation of Weighted SDE). The probability path $\{p_t^*(X_t)\}_{t \in [0,1]}$, $X_t \in \mathbb{R}^d$ can equivalently be simulated by solving the ODE

$$Y_0 \sim p_0^* \otimes \delta(\mathbf{1}), \qquad dY_t = \left( \begin{bmatrix} v_t^* \\ g_t \end{bmatrix} \circ \pi \right) (Y_t) dt. \tag{F.1}$$

where $Y_t = \begin{bmatrix} X_t \\ \log w_t \end{bmatrix} \in \mathbb{R}^{d+1}$ and $\pi$ projects $Y_t$ to $X_t$. Applying distillation techniques to this formulation will enable efficient sampling while maintaining fidelity to the target distribution.

**Expanding modular generation to scientific frontiers.** We demonstrated ACE's capacity to resolve path collapse in high-dimensional, multi-modal settings through the scaffold decoration task, where it successfully coordinated distinct modalities (bond topology, protein pocket, and 3D conformation). This success suggests that ACE can extend to even more complex scientific tasks, such as Reviewer gBQy, naNy

protein glue generation and fragment linking, by decomposing them into families of existing models (de novo, conformer, and SBDD), though empirical validation on these tasks remains future work.

**Optimizing exponent schedules for scientific and creative tasks.** Our current ACE schedules use simple bump functions chosen to satisfy the path-existence criterion and, in the homogeneous regime, to modestly increase $C(t)$ at informative times. The COCO-MIG experiments in Appendix E.4 show that even such simple time-varying exponents can significantly improve sample quality over both NR and FKC, despite the absence of path collapse. This suggests a promising direction: treating the exponent schedules $\gamma_i(t)$ themselves as objects to be optimized for downstream objectives such as alignment or diversity. A complementary line of work is to extend ACE to truly heterogeneous creative pipelines that combine distinct backbones and modalities (e.g., global image-editing models with local text-conditioned generators), where incompatible noise schedules would otherwise induce collapse. In both cases, ACE provides the theoretical guarantees needed to explore richer steering schemes without sacrificing validity.

**Extension to hybrid process in mixed continuous-categorical domains.** As we present in E.1, many scientific and molecular generative tasks involve data living in a product space of continuous coordinates (e.g., 3D positions) and categorical variables (e.g., atom and bond types). Current diffusion-based steering methods, including ACE, operate on continuous domains where intermediate densities are well defined under Gaussian interpolation. However, categorical variables do not admit Gaussian smoothing, and thus lack a natural notion of intermediate densities or scores along the diffusion path. This mismatch makes ratio-of-densities steering, and the study of Marginal Path Collapse, substantially more challenging in hybrid domains. Future work could develop principled stochastic interpolants for mixed continuous–discrete representations, identify conditions under which well-defined joint paths exist, and characterize when collapse arises in such hybrid settings. Such an extension would enable theoretically sound steering for full molecular structures, jointly evolving coordinates and atom/bond types, thereby broadening ACE's applicability to richer molecular generation pipelines.

Reviewer naNy

**A long-term vision: Towards a modular generative ecosystem.** ACE demonstrates that complex conditional generation can be achieved through the modular composition of pretrained expert models, offering a scalable alternative to monolithic, task-specific architectures. In the long run, this suggests a paradigm shift toward a standardized practice for generative modeling on continuous domains. Instead of training a bespoke model for every novel application, the community could curate a library of highly efficient expert models for fundamental marginal densities, composing them ad hoc using ACE. This modular paradigm opens the door to a principled framework for the collective intelligence of AI, where distinct models collaborate to solve complex tasks beyond the scope of their individual training.