# OpenReview forum: "On the Collapse of Generative Paths: A Criterion and Correction for Diffusion Steering"
_ICLR.cc/2026/Conference — ICLR 2026 Conference Desk Rejected Submission_

### Official Review · Reviewer_naNy · 2025-10-30

**Soundness:** 2
**Presentation:** 1
**Contribution:** 2
**Rating:** 2
**Confidence:** 2

**Summary:**

This paper identifies a phenomenon in diffusion model guidance termed "Marginal Path Collapse" and argues for its negative impact on generation quality. The authors theoretically propose a criterion to predict such collapse and introduce an Adaptive Path Correction with Exponents, ACE method. ACE dynamically adjusts steering weights to ensure a valid probability path, thereby eliminating marginal path collapse.

**Strengths:**

The theoretically inspired perspective of this work is valuable. By highlighting "Marginal Path Collapse", a phenomenon previously overlooked in the literature, the paper provides a complementary viewpoint for inference-time control in diffusion models.

**Weaknesses:**

1. Substantial improvements to writing clarity are needed:
Background context for foundational concepts is insufficiently explained before introducing abstract formulations. For example:
    - The origin of the "ratio-of-densities" problem (e.g., the core principles of classifier-free guidance and contrastive decoding, and how these methods lead to the ratio-of-densities form) remains unclear in the main text. The definition of "experts" in the context of these methods is not specified.
    - The Feynman–Kac Correctors method, which is repeatedly compared throughout the paper, lacks explicit mathematical formulations.

2. The significance of Marginal Path Collapse problem are not convincing. See specific questions below for further details.

3. Experimental design is either unreasonable or inadequately described:Key aspects of the experimental setup are unclear, making it difficult to assess the validity of results. See specific questions below.

4. Minor notation issues:
    - Line 408: The conditional independence notation is better be standardized to A⫫ Y | X
    - Lines 417 and 432: Extra "." is present.

**Questions:**

1. Is the key cause of "collapse" the use of different noise schedules across experts? Would this scenario hold in practical applications?

2. I also have some intuitive observations to discuss with the authors: If all experts adopt the same noise schedule, collapse seems unlikely, for the following reasons:
    - In Classifier-Free Guidance (CFG), conditional and unconditional generative models are trained simultaneously. The support of the conditional distribution p is typically contained within the support of the unconditional distribution q—since data used for conditional training can always be reused for unconditional training, the density ratio of \(x_0\) (i.e., \(p(x_0)/q(x_0)\)) should never involve division by zero.
    -  The conditional distribution p often exhibits sharper concentration with lighter tails than the unconditional distribution q. For example, if q is a Gaussian distribution with a large variance, p might be a Gaussian with a smaller variance. Since p has lighter tails, the density ratio \(\log(p(x_t)/q(x_t))\) would still integrable.
    - At intermediate diffusion timesteps t, the shared noise schedule ensures p still has lighter tails, having no occurrence of collapse.

1. For Figure 3, could visualizations of generated results from different methods be added? Such visualizations would intuitively illustrate how Marginal Path Collapse affects generation quality.

2. Even if Marginal Path Collapse occurs, the Classifier-Free Guidance (CFG) algorithm itself seems unaffected—given that CFG only relies on log-gradient information of densities. In the Gaussian example provided: even if the density is non-integrable  due to a positive coefficient in the exponent, taking the log and computing the gradient can still yields a finite gradient.  If collapse occurs over a small number of steps or with small step sizes, generation quality may still remain acceptable. Can the authors use experimental results to show how the duration of the collapse interval influences generation quality? This would help demonstrate that collapse can truly impairs CFG performance.

3. Out of curiosity: Does Classifier Guidance avoid the collapse problem? Its marginal density involves multiplying (rather than dividing) probability densities, which may prevent the issues caused by division in ratio-of-densities forms.

4. The experimental design of the "Synthetic Checker Dataset" is unclear, and its rationality needs clarification:
    - Among the pre-trained experts, which one learns the joint distribution of x and y as a checkerboard distribution? Lines 377–408: Mathematical notations do not mention the checkerboard distribution or the introduction of the [-4, 4] region boundary. Is it assumed that these experts learn checkerboard distributions within a square region under specific conditions—e.g., the first two experts operate under the conditions $x \geq 0$ and $x + y \geq 0$? If this is the case, is the third expert (a 1D unconditional model for p(x)) a uniform distribution over \([-4, 4]\)?
    - This setup seems counterintuitive: In typical CFG applications, experts usually model distributions of the same dimensionality. For instance, in this 2D problem, all experts should ideally model 2D distributions (e.g., checkerboard distributions within a square region, with some experts incorporating additional conditions like $x \geq 0$ and $x + y \geq 0$.

5. For the "Flexible-pose scaffolding decoration" experiment, what are the specific definitions of each expert? The appendix mentions DiffSBDD, GeoDiff, and EDM—but the latter two models do not natively incorporate pocket information. How were they adapted to achieve the dual goals of "preserving topology while adapting to the pocket"? Additionally, citations for these methods are missing.

---

> ### Author Response · Authors · 2025-11-27
> **[1/6] Response to Reviewer naNy**
>
> We thank the reviewer for the detailed and thoughtful comments. We have substantially revised the introduction, background, and experimental sections to improve clarity and to address all technical and conceptual questions. All changes are highlighted in blue and annotated with reviewer IDs.
>
> ---
>
> ### W1(a). *“The origin of the ratio-of-densities problem remains unclear.”*
>
> **Author actions.** We revised the **Introduction**, added **Appendix A.2: Algorithmic Formulations and Comparison: FKC vs. ACE**, and **Appendix D.1: Common Steering Objectives as Ratio-of-Densities Paths**. We now provide explicit examples and a mathematical demonstration that CFG, product-of-experts conditioning, contrastive decoding, reward-tilting, and Bayesian conditioning with pretrained experts all reduce to a ratio-of-densities path.
>
> **Details.** Many standard inference-time control techniques can be unified under a ratio-of-densities framework, where the target density takes the form $p^\star(x) \propto \prod_i q^{(i)}(x)^{\gamma_i}$. Along the generative trajectory, this induces a time-indexed family $p_t^\star(x) \propto \prod_i q_t^{(i)}(x)^{\gamma_i}$, whose intermediate densities must remain normalizable for valid sampling.
>
> * **Classifier-free guidance (CFG).** To enhance samples consistent with condition $y$, CFG reweights the conditional model relative to the unconditional one: $p^\star(x) \propto p_\theta(x\mid y)^{\gamma},p_\theta(x)^{1-\gamma}$, implying the path $p_t^\star(x)\propto q_t(x\mid y)^{\gamma},q_t(x)^{1-\gamma}$ with score $\gamma \nabla\log q_t(x\mid y)+(1-\gamma)\nabla\log q_t(x)$.
> * **Product-of-experts.** To enforce multiple constraints $c_1,\dots,c_K$, one multiplies separate experts $q^{(k)}(x)=p_\theta(x\mid c_k)$, yielding $p^\star(x)\propto \prod_{k=1}^K q^{(k)}(x), \quad  p_t^\star(x)\propto \prod_{k=1}^K q_t^{(k)}(x)$.
> * **Reward-tilted / RL-style guidance.** Combining a base model $p_\theta(x)$ with a reward $r(x)$ gives $p^\star(x)\propto p_\theta(x)\exp(\beta r(x))$. This can be written as a product of a base density $q_{\text{base}}(x)$ and a reward density $q_{\text{rew}}(x)\propto\exp(\beta r(x))$, leading to $p_t^\star(x)\propto q_t^{\text{base}}(x),q_t^{\text{rew}}(x)$.
> * **Contrastive / reference-model decoding.** To suppress generic patterns from a reference model $p_{\mathrm{ref}}$, the target is $p^\star(x)\propto p_{\mathrm{LM}}(x)^{\gamma},p_{\mathrm{ref}}(x)^{-\gamma}$, inducing $p_t^\star(x)\propto q_t^{\mathrm{LM}}(x)^{\gamma},q_t^{\mathrm{ref}}(x)^{-\gamma}$.
> * **Bayesian composition.** When only conditionals $p_\theta(x\mid c_k)$ are available, Bayes’ rule yields $$p(x\mid c_{1:K})\propto {p(x)}\prod_k p(c_k\mid x)\propto \underset{\text{\tiny prior}}{p(x)}\prod_k p_\theta(x\mid c_k)/\underset{\text{\tiny marginal}}{p_\theta(x)}$$The assumptions and derivation under this heterogeneous conditioning structure are given in **Eq. 11**. This induces the path $p_t^\star(x)\propto q_t^{\mathrm{prior}}(x),\prod_k q_t^{(k)}(x),q_t^{\mathrm{marg}}(x)^{-1}$, which fits our general ratio-of-densities template.
> ---
> ### W1(b). *“Definition of experts not specified; FKC lacks explicit mathematical formulations.”*
>
> **Author actions.** We added explicit mathematical and algorithmic formulations of FKC and ACE in **Appendix A.2**, including pseudo-code, and clarified the notion of “experts” in **Section 2.1**.
>
> **Details.**
>
> * **Definition of experts.** In **Section 2.1**, we now define “experts” as generative models ${q_t^{(i)}}$ (possibly on different domains and trained with different schedules) whose outputs are multiplicatively combined at inference time. In our molecular tasks, DN/CONF/SBDD models serve as such experts.
>
> * **FKC formulations.** In **Appendix A.2**, we introduce FKC as a special case of ACE with constant exponents and provide:
>   (i) the Feynman–Kac PDE,
>   (ii) the associated SDE with weight process, and
>   (iii) **Algorithm 2** (pseudo-code).
>   This makes explicit how FKC operates and why it fails when the Path Existence Criterion (PEC) is violated (i.e., when no valid target $p^\star _t$ exists to correct toward).
> ---
> (continued)

---

> ### Author Response · Authors · 2025-11-27
> **[2/6] Response to Reviewer naNy**
>
> ### W2–Q1/Q2.
>
> *“Is collapse mainly due to different noise schedules across experts? Would this scenario hold in practical applications? The significance of Marginal Path Collapse is not convincing. If all experts adopt the same schedule, collapse seems unlikely.”*
>
> **Author actions.** We added **Appendix E.2, E.3, and E.4**.
>
> **Answer.** Yes. The primary cause of Marginal Path Collapse is the mismatch in noise contraction rates $\alpha^{(i)}_t$ between component models. **Appendix E.2** quantifies how often this occurs, and **Appendix E.3–E.4** explain why heterogeneous schedules are effectively unavoidable in realistic DN/CONF/SBDD pipelines.
>
> This scenario is highly prevalent in practice for two reasons:
>
> 1. **Task-specific schedulers are necessary (Appendix E.3).** Each task achieves its best performance under a distinct, task-specific noise schedule.
>
> 2. **DN/CONF/SBDD heterogeneity is inevitable (Appendix E.2).** Our scaffold-decoration pipeline necessarily composes models with heterogeneous schedules.
>
> High-level, there is no single “universal” scheduler: the optimal schedule depends strongly on data modality and resolution. Different model families thus use diverse, incompatible schedules. In realistic pipelines, practitioners cannot simply standardize schedules without harming individual model performance. They are therefore forced to compose heterogeneous schedules, making path collapse a *systematic* hurdle that ACE addresses.
>
> We also emphasize that ACE’s contributions go beyond preventing collapse:
>
> 1. **Path Existence Criterion (PEC).** A simple test for path normalizability.
> 2. **Bump-function protocol.** A simple, automatic fix when PEC fails.
> 3. **Generalized Feynman–Kac PDE for time-varying exponents.** This enables optimizing a time-varying exponent schedule *even when collapse does not occur*, improving sample quality by stabilizing the effective sampling radius (**Appendix E.4**).
>
> We detail these points below.
>
> ---
>
> ### (1-1) Necessity of task-specific noise schedulers (Appendix E.3)
>
> A growing body of work on task-specific noise scheduling shows that different generative tasks require different allocations of noise across the denoising trajectory. This follows the standard view that diffusion operates in two regimes:
>
> * a **high-noise** regime for global exploration, and
> * a **low-noise** regime for local refinement,
>
> whose relative importance is task-dependent. We validate this trend in molecular generation and note that recent advances in learnable or adaptive schedulers reinforce this perspective. Together, these observations motivate adaptive/customized schedulers and, by extension, heterogeneous combinations of schedulers.
>
> **Table E.7(A): Comparison of high-noise vs. low-noise regimes ([7–10]).**
>
> |**Regime**|**Feature scale priority**|**Role in denoising**|**Dynamics**|
> |-|-|-|-|
> |$\mathcal{R}_H$ (high noise)|Global structure|Exploration|Large transitions, broad search|
> |$\mathcal{R}_L$ (low noise)|Local fine-grained details|Refinement|Small corrective steps, manifold-sensitive|
>
> **Table E.7(B): Average noise-regime statistics per task in Tables E.2–E.4.**
> *Asterisk indicates a significant difference from DN over Tables E.2–E.4 ($p<0.001$).*
>
> |**Task**|**Cond. vs Uncond.**|**Global vs Local**|**Exploration vs Refinement**|Length $R_H$|Length $R_L$|$\text{len}(R_H)/\text{len}(R_L)$|
> |-|-|-|-|-|-|-|
> |DN|Uncond.|Global|Exploration|0.53|0.47|1.11|
> |CONF|Cond.|Local|Local refinement|0.47*|0.53*|0.89*|
> |SBDD|Cond.|Local|Local refinement|0.48*|0.52*|0.91*|
>
> **Foundational regime analysis: high vs. low noise.**
> Recent analyses ([7–10], summarized in **Table E.9**) identify two complementary phases:
>
> * The **high-noise regime** $R_H$ induces large transitions that enable global exploration and the formation of coarse structure, supporting sample diversity.
> * The **low-noise regime** $R_L$ consists of small corrective steps that refine local, fine-grained features, enabling precise reconstruction.
>
> Effective scheduling must balance these phases according to the task (**Table E.7(A)**).
>
> **Heterogeneous schedulers for conditional vs. unconditional molecular generation.**
> As shown in [7], larger images benefit from a longer $R_H$, reflecting their need for stronger exploration. This insight transfers naturally to molecules:
>
> * **Unconditional generation (DN)** must explore many molecular shapes and therefore relies more on the broad exploratory behavior of $R_H$.
> * **Conditional tasks (CONF, SBDD)** must satisfy structural constraints and thus benefit more from the precise corrections in $R_L$.
>
> Empirically, existing molecular schedulers follow this pattern (**Tables E.2–E.4**). We split the noise phases using $\mathrm{SNR} \ge 1$ for $R_H$ and $\mathrm{SNR} < 1$ for $R_L$. For each model in **Tables E.2–E.4**, we compute $|R_H|$, $|R_L|$, and their ratio, then average across models per task. **Table E.7(B)** shows that DN favors $R_H$, whereas CONF and SBDD prioritize $R_L$.
>
> (continued)

---

> ### Author Response · Authors · 2025-11-27
> **[3/6] Response to Reviewer naNy**
>
> These results support our central claim: *heterogeneous noise schedulers are not only reasonable but necessary* for the molecular tasks considered in **Section E.2**.
>
> ---
>
> **Recent trends in task-specific scheduler design.**
> Recent works further emphasize that optimal schedules depend on task characteristics (**Table E.8**).
>
> **Table E.8: References on task-specific and adaptive scheduling.**
>
> |**Reference**|**Contribution**|
> |-|-|
> |Vignac et al. (2023) [1]| Component-specific molecular noise scheduling.|
> |Lee et al. (2024) [2]| Fully data-driven adaptive scheduler for time-series diffusion.|
> |Sahoo et al. (2024) [3]| MuLAN: learned multivariate adaptive noise processes.|
> |Seo et al. (2025) [4]| Per-element optimized forward trajectories for molecules.|
> |Sorokin et al. (2025) [5]| Adaptive allocation between $R_H$ (exploration) and $R_L$ (refinement).|
> |Choi et al. (2025) [6]| Distinct channel-wise schedules to balance the diversity–accuracy trade-off.|
>
> ---
>
> **Table E.9: Foundational references for noise-regime mechanisms.**
>
> |**Reference**|**Contribution**|
> |-|-|
> |Chen (2023) [9]|Shows that optimal schedules vary with task and resolution.|
> |Choi et al. (2022) [7]|Identifies noise levels that serve as effective pretext tasks for learning rich visual concepts.|
> |Rothchild et al. (2023) [8]|Analyzes diffusion dynamics in molecular generative models.|
> |Pavlova & Wei (2025) [10]|Identifies $R_L$ as the phase in which precision and fine-scale structure dominate.|
>
> ---
>
> ### (1-2) Inevitable heterogeneity of DN/CONF/SBDD models
>
> As discussed in **Appendix E.1**, complex tasks such as scaffold decoration or protein glue generation exhibit a **heterogeneous conditioning structure** that necessitates composing distinct model families (e.g., De Novo, Conformer, and SBDD). Our survey in **Tables E.2–E.4** shows that existing DN/CONF/SBDD models use heterogeneous schedules by design.
>
> Since some schedulers appear across tasks, one might ask whether composing models under a shared (homogeneous) scheduler could avoid path collapse. In molecular generation, this is generally infeasible due to two fundamental constraints:
>
> * **Heterogeneity in representations.**
>   Unlike images, molecules admit multiple incompatible representations. Some models encode atom types as continuous one-hot vectors $H^{\mathrm{one\text{-}hot}} \in \mathbb{R}^N$ evolving under Gaussian convolution; others use categorical variables with discrete states or operate in latent spaces. These spaces differ in dimensionality, continuity, and semantics, and cannot be embedded into a unified ambient space with a single, consistently defined diffusion path. This fundamentally constrains which experts can be mathematically composed.
>
> * **“Empty intersection” of schedules.**
>   Even if we restrict to models with compatible continuous representations, a homogeneous DN/CONF/SBDD triplet often does not exist. In our scaffold-decoration experiment:
>
>   * DN models (EDM, GCDM) use **quadratic** schedulers.
>   * SBDD models (DiffSBDD, DualDiff) use quadratic or sigmoid schedulers.
>   * State-of-the-art CONF models predominantly use **sigmoid** schedules rather than quadratic ones.
>
>   Consequently, no common schedule simultaneously matches all three families. This structural mismatch necessitates heterogeneous schedules (quadratic for DN/SBDD, sigmoid for CONF), making ACE’s path-existence guarantees a prerequisite for reliable generation.
>
> ---
>
> ### (1-3) ACE advantages beyond path collapse
>
> In **Section E.4**, we show that ACE provides clear benefits even when path collapse does *not* occur (e.g., standard CFG setups). The key reason is that the time-dependent exponents introduced by ACE’s bump function yield a more stable trajectory by preventing excessive growth of the effective radius along the denoising path.
>
> This is supported theoretically by **Proposition E.1**, which interprets $C(t)$ as a factor controlling an upper bound on the effective radius of the intermediate densities. Increasing $C(t)$ via the bump term not only prevents divergence but also reduces variance expansion, as illustrated in **Figure E.12**.
>
> We further validate this in image experiments (also framed under the ratio-of-densities setting). As shown in **Table E.10**, ACE with bump parameter $B=5$ consistently outperforms the conventional FKC path (constant exponents) and NR by a significant margin. Thus, both PEC and ACE constitute novel and technically valuable contributions.
>
> ---
> (continued)

---

> ### Author Response · Authors · 2025-11-27
> **[4/6] Response to Reviewer naNy**
>
> ### Q3. *“For Figure 3, could visualizations of generated results from different methods be added?”*
>
> For three-expert compositions using five standard schedules (the setup of **Figure 3**, excluding the custom schedule), there are 41–80 different schedule combinations that violate PEC, depending on the guidance scale (full list in **Figure E.5**). In **Figures E.6–E.11**, we visualize generated results from five representative combinations across multiple conditioning scenarios. These visualizations are intended to give an intuitive view of how Marginal Path Collapse affects generation quality in NR and FKC.
>
> ---
>
> ### Q4.
>
> *“Even if Marginal Path Collapse occurs, CFG itself seems unaffected—CFG only uses log-gradients. In the Gaussian example, even if the density is non-integrable, the log-gradient can still be finite. If collapse occurs over a small number of steps, generation quality might remain acceptable. Can you show how the duration of the collapse interval influences quality, to demonstrate that collapse truly impairs CFG performance?”*
>
> We added experiments showing that even short collapse intervals (7–11% of the path) cause severe degradation in sample quality.
>
> **Why finite gradients are not enough.**
> CFG uses log-gradients, but finite mixed gradients alone do not guarantee that the sampler follows the intended ratio-of-densities path.
>
> Let ${p_t^\star(x)\propto h_t(x)}$ denote the ideal target path, where $h_t$ is the ratio-of-densities integrand from **Section 2.1**, and let ${p'_t}$ be the actual marginal law induced by the FKC (or CFG) dynamics. The Feynman–Kac derivation (Remark in **Appendix A.2**) shows that $p'_t = p_t^\star$ only if:
>
> 1. $h_t \in L^1$ so that $p_t^\star$ is a normalized density for all $t$; and
> 2. the sampler’s drift coincides with the score field $\nabla_x \log p_t^\star(x)$, so that $p_t^\star$ solves the corresponding Fokker–Planck equation.
>
> When Marginal Path Collapse occurs (i.e., $h_t \notin L^1$ and $Z_t=\int h_t = \infty$ at some $t$), no such $p_t^\star$ exists. The mixed field $s_t^\star = \sum_i \gamma_i(t),\tilde s^{(i)}_t$ is still a well-defined vector field, but it is no longer the gradient of the log-density of any normalizable $p_t^\star$. In this regime, FKC/CFG still define a valid stochastic process with some probability path ${p'_t}$, but this path is *no longer constrained* to be the intended ratio-of-densities path.
>
> Empirically, whenever our path-existence criterion detects $h_t\notin L^1$ (i.e., $C_k(t)<0$), the discrepancy between $p'_t$ and any reasonable target becomes stark: FKC and NR exhibit severe instability and loss of fidelity (**Tables 2, 3, 4; Figures E.4–E.11**), whereas ACE enforces normalizability at all times and recovers the desired target distributions.
>
> **Impact of collapse-interval duration.**
> In **Figure E.6**, we analyze schedule combinations whose collapse intervals range from 7.5% to 48.8% of the trajectory. We show that even short collapse intervals (7.5% of the path) cause substantial degradation in FKC performance (metrics and validity). In all cases where $C(t)<0$ occurs, NR and FKC show strong deterioration in Wasserstein/MMD metrics and sample validity. This suggests that the *presence* of collapse, not just its duration, is critical.
>
> Conversely, in a homogeneous control experiment (**Table E.6**, reproduced below) where collapse does not occur, ACE with bump parameter $B=0$ reduces to FKC. FKC/ACE then achieve near-optimal metrics, while NR (CFG) remains suboptimal:
>
> **Table E.6. Distributional metrics under homogeneous composition (no path collapse).**
> Here, ACE with bump $B=0$ coincides with FKC; NR (CFG) performs significantly worse across 5 seeds.
>
> |Method|`W1`|`W2`|`MMD_RBF`|
> |-|-|-|-|
> |NR (CFG)|0.859 ± 0.023|1.113 ± 0.023|0.077 ± 0.003|
> |FKC|0.162 ± 0.019|0.282 ± 0.017|0.012 ± 0.001|
> |ACE|0.162 ± 0.019|0.282 ± 0.017|0.012 ± 0.001|
>
> Taken together, these results show:
> (i) FKC performs well when its assumptions hold but fails precisely when PEC is violated (even briefly), and
> (ii) NR’s limitations are independent of collapse (it remains suboptimal even when the path is valid).
>
> ---
>
> ### Q5. *“Out of curiosity: Does Classifier Guidance avoid the collapse problem?”*
>
> Yes. Pure classifier guidance multiplies densities with **nonnegative** exponents under a **shared** schedule, so $C_k(t)>0$ and collapse is unlikely. Our work targets the broader ratio-of-densities setting, particularly with **negative exponents**, where PEC predicts collapse and ACE is needed.
>
> ---
>
> (continued)

---

> ### Author Response · Authors · 2025-11-27
> **[5/6] Response to Reviewer naNy**
>
> ### W3–Q6. *“Experimental design of synthetic checker dataset and flexible-pose scaffold decoration is unclear / counterintuitive.”*
>
> **Author actions.** We revised **Appendix C.1: Synthetic Dataset: 2D Checker** and added **Appendix E.1: Heterogeneous Conditioning Structures in Scientific Tasks**. We now explicitly define the checker distribution, the three experts, and the Bayes factorization. We also formalize *heterogeneous conditioning structures* and show that the same pattern appears in scaffold decoration, linker generation, and protein glue design. For scaffold decoration, we specify the roles of DN/CONF/SBDD experts and add citations.
>
> ---
>
> ### Q6-1. *“Among the pre-trained experts, which one learns the joint distribution of x and y as a checkerboard distribution?”*
>
> We now define the ground-truth 2D checkerboard distribution on $\mathcal{D}=[-4,4]^2$ in **Eq. C.1** as $$p_{\texttt{Checker}}(x,y) = \frac{1}{32}\mathbf{1}\left( \left\lfloor \frac{x}{2} \right\rfloor + \left\lfloor \frac{y}{2} \right\rfloor \text{ is odd} \right)\mathbf{1}\left( (x,y)\in \mathcal{D} \right)$$where $\lfloor \cdot \rfloor$ is the floor function and $\mathbf{1}(\cdot)$ is the indicator. The support consists of alternating $2\times2$ squares.
>
> The three experts are:
> * $q^{(1)}(x \mid x \ge 0)$ (1D): a uniform distribution over $[0,4]$;
> * $q^{(2)}(x,y \mid x+y \ge 0)$ (2D): the joint checkerboard distribution on the half-plane $x+y \ge 0$;
> * $q^{(3)}(x)$ (1D): a uniform distribution over $[-4,4]$.
>
> Both $q^{(1)}$ and $q^{(3)}$ are **1D** models over $x\in[-4,4]$ and model only a 1D uniform distribution. Only $q^{(2)}$ learns the *joint* checkerboard distribution of $(x,y)$, conditioned on $x+y\ge 0$. The first model does **not** learn the checkerboard distribution on the full 2D square (it has no knowledge of the joint). We updated the experiment section and **Appendix C.1** to make this explicit.
>
> ---
>
> ### Q6-2. *“This setup seems counterintuitive. In typical CFG applications, experts usually model distributions of the same dimensionality. In this 2D problem, all experts would ideally be 2D models.”*
>
> Typical CFG applications assume a simple single-condition structure. In contrast, we consider a **multi-condition** problem $p(x,y\mid A,B)$ with a **heterogeneous conditioning structure**.
>
> In **Appendix E.1** and **Figure E.2**, we define a heterogeneous conditioning structure as one where some conditions constrain only part of the state (e.g., $A$ on $X$) while others constrain the full state (e.g., $B$ on $(X,Y)$). This is exactly the structure of both the synthetic checker experiment and our scaffold-decoration task.
>
> Using Bayes’ rule with
>
> * $A = { X \ge 0 }$ and
> * $B = { X+Y \ge 0 }$,
> we obtain $$p(x,y \mid A,B) \propto p(x \mid A)\frac{p(x,y \mid B)}{p(x)}$$ which matches **Eq. 11**. This mirrors the molecular setting, where some conditions act only on subsets of variables (e.g., scaffold topology acts only on the scaffold) while others act globally (e.g., pocket binding acts on the entire molecule).
>
> Unlike standard CFG setups, we show that **not all experts must model the full joint** to sample from the constrained joint. The 1D conditional model constrains $x\ge 0$, thereby influencing the joint $(x,y)$, and the resulting samples still respect the underlying joint distribution. This “counterintuitive” design is a key conceptual contribution: it mirrors practical heterogeneous conditioning in scientific tasks.
>
> ---
>
> (continued)

---

> ### Author Response · Authors · 2025-11-27
> **[6/6] Response to Reviewer naNy**
>
> ### W3–Q7.
> *“For the ‘Flexible-pose scaffold decoration’ experiment, what are the specific definitions of each expert? DiffSBDD, GeoDiff, and EDM are mentioned—but the latter two do not natively incorporate pocket information. How were they adapted to achieve the dual goals of ‘preserving topology while adapting to the pocket’?”*
>
> **Flexible-pose scaffold decoration** has the same heterogeneous conditioning structure as the checker example. We make the analogy explicit by substituting $(X,Y)$, $A$, and $B$ as:
>
> * $(X,Y) = (\mathcal{M}^{\mathrm{sc}}, \mathcal{M}^{\mathrm{R}})$;
> * $A$ = preserve the topology of $\mathcal{M}^{\mathrm{sc}}$ as $\mathcal{T}^{\mathrm{sc}}$;
> * $B$ = adapt the entire molecule $(\mathcal{M}^{\mathrm{sc}}, \mathcal{M}^{\mathrm{R}})$ to bind to the pocket $\mathcal{P}$.
>
> The task is to sample from $p((\mathcal{M}^{\mathrm{sc}}, \mathcal{M}^{\mathrm{R}})\mid A,B)$, which encodes the dual objective “preserving topology while adapting to the pocket.” (Other scientific tasks like **linker design** and **protein glue generation** admit analogous decompositions; **Examples E.2 and E.3**).
>
> For flexible-pose scaffold decoration, **Eq. (9)** and **Example E.1** specify the experts:
>
> * **Topology-aware expert (CONF, GeoDiff):** $p(\mathcal{M}^{\mathrm{sc}} \mid \mathcal{T}^{\mathrm{sc}})$
>   is a conformer-generation model producing 3D molecules whose topology matches the given $\mathcal{T}^{\mathrm{sc}}$.
>
> * **Pocket-aware expert (SBDD, DiffSBDD):** $p(\mathcal{M} \mid \mathcal{M} \leftrightarrow \mathcal{P})$ is an SBDD model generating 3D molecules that bind to the protein pocket $\mathcal{P}$.
>
> * **Unconditional prior expert (DN, EDM):** $p(\mathcal{M}^{\mathrm{sc}}), p(\mathcal{M})$ are De Novo models generating 3D molecules unconditionally.
>
> We emphasize that **CONF and DN are not adapted to the pocket**. Pocket information is captured solely by the SBDD expert, analogous to how the checkerboard constraint $x+y\ge 0$ is captured solely by $q^{(2)}$ in the synthetic experiment. The ratio involving DN and CONF then enforces the “preserve topology” condition *in addition* to “adapt to the pocket.”
>
> We added citations for DiffSBDD, GeoDiff, and EDM in both the main text and appendix, and we summarize their schedulers in **Tables E.2–E.4**.
>
> ---
>
> ### W4. *Typos and minor issues.*
>
> We thank the reviewer for pointing out these issues and have corrected all identified typos.
>
> ---
>
> ### References
>
> [1] Vignac et al., *MiDi: Mixed Graph and 3D Denoising Diffusion for Molecule Generation*, ICLR MLDD Workshop, 2023.\
> [2] Lee et al., *ANT: Adaptive Noise Schedule for Time Series Diffusion Models*, NeurIPS, 2024.\
> [3] Sahoo et al., *Diffusion Models with Learned Adaptive Noise*, NeurIPS, 2024.\
> [4] Seo et al., *Learning Flexible Forward Trajectories for Masked Molecular Diffusion*, arXiv preprint, 2025.\
> [5] Sorokin et al., *ImageReFL: Balancing Quality and Diversity in Human-Aligned Diffusion Models*, arXiv preprint, 2025.\
> [6] Choi et al., *Channel-wise Noise Scheduled Diffusion for Inverse Rendering in Indoor Scenes*, CVPR, 2025.\
> [7] Choi et al., *Perception-Prioritized Training of Diffusion Models*, CVPR, 2022.\
> [8] Rothchild et al., *Investigating the Behavior of Diffusion Models for Accelerating Electronic Structure Calculations*, Chemical Science, 2023.\
> [9] Chen, *On the Importance of Noise Scheduling for Diffusion Models*, arXiv preprint, 2023.\
> [10] Pavlova and Wei, *Diffusion Models Under Low-Noise Regime*, arXiv preprint, 2025.

---

> > ### Comment · Reviewer_naNy · 2025-11-28
> >
> > Thank you for your detailed responses and meticulous revisions to the manuscript. All my previous questions have been fully addressed, and the logical clarity of the paper has seen a substantial improvement. I am pleased to raise my evaluation score to 8. (I cannot revise now and will revise the score once the system is fixed.) I hope the author can release the code soon.

---

> > > ### Author Response · Authors · 2025-11-28
> > >
> > > Thank you for your positive follow-up and for raising your score. We are glad that the revisions clarified the logic and fully addressed your concerns. We will release the codebase soon to support full reproducibility of both the main and additional experiments.

---

### Official Review · Reviewer_jeuE · 2025-10-30

**Soundness:** 3
**Presentation:** 2
**Contribution:** 3
**Rating:** 8
**Confidence:** 2

**Summary:**

The paper investigates a fundamental failure mode of ratio-of-densities diffusion steering, termed Marginal Path Collapse (MPC), which occurs when intermediate densities along a steered generative path become non-normalizable. This leads to catastrophic sampling failures, particularly when composing heterogeneous pretrained models.

To address this, the authors first introduce the Path Existence Criterion (PEC), which predicts when MPC will occur, and then propose Adaptive path Correction with Exponents (ACE), which dynamically adjusts the steering exponents via a bump function to ensure the PEC holds throughout the entire trajectory, while preserving the endpoint marginals.

The method is evaluated on both a synthetic 2D Checkerboard benchmark and a flexible-pose scaffold decoration task for molecular design, demonstrating theoretical soundness and strong empirical performance.

**Strengths:**

- The contribution is twofold and novel within diffusion steering: the paper clearly identifies MPC as a failure mode and introduces PEC and ACE as elegant, principled solutions.

- The method can be applied in general ratio-of-densities steering settings.

- The synthetic benchmarks convincingly demonstrate the MPC phenomenon and correction.

- The molecular design experiments (Tables 2–3) show ACE enabling valid and chemically meaningful samples where FKC fails, with improved docking scores.

**Weaknesses:**

- The notation and derivations are somewhat heavy; for instance, Theorem 2.3 could be presented with more clarity. A pseudo-code summary of ACE would greatly improve readability.

- The paper does not clearly explain how to choose the bump coefficient B.

- A few minor typos remain: "frequently violate the ..." (around line 239), "Assume that the sum Assume ..." (around line 263), beginning of 3.1.

**Questions:**

- Are there any computational challenges associated with ACE? For example, does it require finer path discretization at inference time?

- What are the trade-offs involved in selecting B? It seems that choosing a sufficiently large B will always satisfy the PEC, but could that introduce undesirable irregularities in the path or increased inference cost?

- Is there a particular reason not to use ACE with ODE samplers?

---

> ### Author Response · Authors · 2025-11-27
> **[1/1] Response to Reviewer jeuE**
>
> We thank the reviewer for the positive evaluation and for recognizing PEC and ACE as elegant, principled solutions. We have updated both the main manuscript and the supplementary materials in response to the comments. All changes are highlighted in blue, with reviewer IDs annotated for clarity.
>
> 1. ***“The notation and derivations are somewhat heavy; Theorem 2.3 could be presented with more clarity. A pseudo-code summary of ACE would greatly improve readability.”***
>
> **Author Actions:**
> - We simplified **Theorem 2.3** and rewrote it to separate the path definition ($p_t^\star \propto \prod_i q_t^{(i)}{}^{\gamma_i(t)}$) from the SDE/ODE dynamics and the log-weight update, thereby reducing the notational burden. We also fixed the typos.
> - We now provide the sampling algorithms for ACE and FKC in **App. A.2**. The main difference between ACE and FKC is that ACE incorporates a time-varying exponent schedule, obtained by adding a bump function to ensure the PEC is satisfied. This sampling procedure is illustrated in **Fig. 4**.
>
> 2. ***“How to choose the bump coefficient $B$ and what are the trade-offs?”***
>
> **Author Actions:** We added a figure (**Fig. 6**) and a paragraph in the discussion section (**Sec. 4**) discussing the choice of $B$ and its trade-offs.
>
> **Details.** **Fig. 6** shows the empirical trade-off: performance peaks near $B=30$ and remains stable for larger values ($B=100$), while too-small choices ($B=10$) fail to satisfy the PEC and lead to sharp degradation. Since the PEC depends only on analytic schedules and exponents, candidate $B$ values that satisfy $C(t)>0$ can be checked cheaply, without any model evaluation. We note that $B$ acts as a scalar multiplier on the guiding vector field and thus a large $B$ may amplify any network approximation errors. This reflects a trade-off: a large $B$ will always satisfy the PEC, but can slightly amplify the network's approximation error. We found no additional inference costs or numerical irregularities for using a larger $B$. In practice, $B=30$ emerges as a good default across most scenarios, both on the synthetic Checker benchmark (**Fig. E.6**) and on scaffold decoration (**Tab. 3-4**).
>
> **Beyond collapse.** Although we introduce $B$ primarily to enforce the PEC in collapsing regimes, we also observe gains in settings where $C(t)>0$ for all $t$. In the compositional image generation task (**App. E.4**), all experts share the same backbone and noise schedule, so ACE with $B=0$ already satisfies the PEC and reduces to FKC. Nonetheless, a small bump ($B=5$) significantly improves attribute alignment and spatial metrics over constant-exponent steering. This behavior is consistent with our radius bound (**Prop. E.1**), which shows that larger $C(t)$ tightens the effective radius of $p_t^\star$, reducing mass in off-manifold regions. Systematically optimizing time-varying exponent schedules $\gamma_i(t)$ for quality, even in homogeneous regimes, is an interesting direction for future work.
>
> 3. ***“Are there computational challenges associated with ACE? Does it require finer path discretization?”***
>
> We clarify that we did not use finer discretization, but instead followed the standard step counts of the pretrained models (1,000 steps for the synthetic checker, 500 for scaffold decoration, 50 for image), and we observed no need for denser grids. In terms of computational cost, ACE has the same order of complexity as FKC: the additional divergence term is efficiently handled using Hutchinson’s trace estimator[1], requiring only one extra backward pass per step. Thus, ACE remains practical and scalable for high-dimensional DN/CONF/SBDD or image models.
>
> **Author Actions:** All relevant details are now added to **Sec. C.1**.
>
> [1] Hutchinson, M. F., “A stochastic estimator of the trace of the influence matrix for Laplacian smoothing splines,” Communications in Statistics – Simulation and Computation, 18(3):1059–1076, 1989.
>
> 4. ***“Is there a particular reason not to use ACE with ODE samplers?”***
>
> ACE can be used with ODE samplers, but its resampling step can lead to severe sample degeneracy in deterministic dynamics. For this reason, our main experiments use SDE samplers where sample diversity is preserved under resampling with stochastic dynamics.
>
> **Details.** ACE uses importance resampling to remove low-weight trajectories and duplicate high-weight regions of the path. In SDE samplers, each resampling step is followed by a diffusion step that injects noise, so duplicated particles quickly spread into diverse trajectories (as illustrated in **Figs. 1, 4, E.7-E.11**). In contrast, ODE samplers have deterministic dynamics, so any duplicated particles produced during resampling stay identical for the rest of the trajectory. This rapidly reduces the effective number of distinct samples. We note that ACE could still be used in the ODE setting as a form of early-stage rejection/filtering (discard low-weight paths without duplication).

---

> ### Comment · Reviewer_jeuE · 2025-11-28
> **Reply to Authors**
>
> I thank the authors for taking the time to answer my questions and for their revised submission. My concerns are addressed, I will maintain my positive evaluation, and raise my confidence score to 3.

---

> > ### Author Response · Authors · 2025-11-28
> >
> > Thank you for taking the time to revisit our submission and for increasing your confidence score. We are happy to hear that our revisions and answers resolved your concerns. We will also release the full codebase soon to facilitate reuse and reproducibility.

---

### Official Review · Reviewer_TtEF · 2025-10-31

**Soundness:** 3
**Presentation:** 4
**Contribution:** 3
**Rating:** 6
**Confidence:** 3

**Summary:**

The paper first identifies Marginal Mode Collapse, a failure mode of current ratio-of-densities based inference-time steering schemes for diffusion models. Then, the authors provide a diagnostic test to predict whether this problematic phenomenon will occur during the sampling path. Next, they present ACE, a framework that extends FKC with time-varying exponents dynamically scheduled to prevent such model collapse issue. Ultimately, they evaluate the method in synthetic as well as molecular design experiments.

**Strengths:**

- The paper is very well structured and written. All sections and contributions are both well-motivated and well presented.

- The problem identified and tackled seems practically relevant, especially from a conceptual standpoint.

- The presented diagnostic tool as well as algorithmic solution seem principled, practical, and useful.

**Weaknesses:**

- (main concern) The relevance of the entire work relies on the existence of a theoretically-identified phenomenon. Nonetheless, concrete evidence of it seems somewhat weak. At least to my understanding, this is not well investigated in the current experimental sec., while in principle this should be possible by e.g. measuring how frequently it occurs in practice with near-optimal parameters. Currently, (1) the synthetic experiments seems to be specifically designed s.t. it occurs (as mentioned in lines 409-410), for which I fail to see how this is 'a realistic testbed' (as mentioned in line 409) - especially the third scheduler within the note ($\alpha^3$) seems particularly adversarial. Moreover, (2) in the molecular design experiments it is somewhat unclear if the gains are due to better tuning of the scheme or effective occurrence of the presented mode collapse phenomenon, as this, again, does not seem to be measured.


Although it tackles a very specific problem, I believe that the overall work is relevant, presents potentially useful contributions, and well-written. Nonetheless, the entire relevance relies on the aforementioned point, which currently seems presented/argued in a seemingly very brittle fashion not sufficiently supported by experimental evidence.

**Questions:**

Answers to the following points would be quite essential to better assess this work.

- Could you clarify the points mentioned within the weaknesses section?

- Could you evaluate experimentally the occurrence of the presented phenomenon in (1) truly realistic synthetic settings, and (2) in the molecular design task without using ACE?

---

> ### Author Response · Authors · 2025-11-27
> **[1/1] Response to Reviewer TtEF**
>
> We thank the reviewer for the positive overall assessment and for highlighting the relevance and structure of the work. We have updated both the main manuscript and the supplementary materials and we believe the revised version addresses all identified concerns. All changes made in response to reviewer comments are highlighted in blue and annotated with reviewer IDs.
>
> 1. ***“The relevance relies on the existence of a theoretically-identified phenomenon … concrete evidence of it seems somewhat weak … synthetic experiments seem designed so that it occurs; unclear if gains in molecular design are due to better tuning or actual occurrence of collapse.”***
>
> **Author Actions:** We agree that empirical support for Marginal Path Collapse must be convincing. In the revision, we: (i) systematically measure collapse frequency across all combinations of five standard noise schedules (**App. E.2, Tab. E.5**), (ii) evaluate several realistic synthetic compositions using only widely-used schedules (no adversarial designs) (**Figs E.6-E.11**), and (iii) show, in molecular experiments, that NR and FKC fail precisely in settings where the Path Existence Criterion (PEC) is violated, under **identical** hyperparameters to ACE (**Sec. 2.5-3, App. C.2, E.2, Fig. E.4**). Together, these results show that the phenomenon is common and that ACE’s gains are not due to extra tuning.
>
> **Details.**
>
> **(i) Frequency of collapse in realistic schedule compositions.**
> In **Section E.2** and **Figure 3**, we consider three-expert compositions $h_t = q_t^{(1)}(q_t^{(2)}/q_t^{(3)})^w$ formed from five standard schedules (DDPM, sigmoid, cosine, linear, $1-t^2$). This yields $5^3=125$ schedule triples; excluding homogeneous cases where $\alpha^{(2)}=\alpha^{(3)}$ leaves 100 heterogeneous compositions. As reported in **Table E.5**, the fraction of compositions with $C(t)<0$ on the path grows rapidly with guidance scale: 41\% ($w{=}1.0$), 47\% ($w{=}1.1$), 52\% ($w{=}1.5$), 66\% ($w{=}2.0$), 77\% ($w{=}7.5$), and 80\% ($w{=}15$). Thus, for heterogeneous experts, collapse is frequent even under standard
> schedules and moderate guidance weights.
>
> **(ii) Realistic synthetic settings (non-adversarial).**
> To address the concern that our original synthetic scheduler might be “adversarial,” we added a new suite of experiments using only canonical schedules (DDPM, sigmoid, linear, cosine, polynomial), as shown in **Figure E.6** and **Figures E.7-E.11**. These combinations correspond to realistic mixtures of diffusion and flow-matching schedules. We compute the PEC and observe collapse intervals $C(t)<0$ in many of these settings; quantitatively, the NR and FKC baselines degrade exactly where the PEC predicts violation. Qualitatively, NR leaves out-of-distribution samples in the batch, while FKC exhibits mode collapse and unstable weights. ACE, with the same discretization and ESS threshold, restores valid paths and matches the target Checker distribution across all these realistic combinations.
>
> **(iii) Molecular design: ACE vs NR/FKC under identical tuning.** For the flexible-pose scaffold decoration task, we clarified the experimental design in **Sections 2.5, 3, 3.1, and Appendix C.2**, and we emphasize that NR, FKC, and ACE share identical hyperparameters: same number of particles, steps, noise level, resampling threshold, and hyperparameters of the individual expert models; ACE adds only the bump parameter $B$, fixed to 30. **Tables 3-4** show that NR and FKC suffer from low validity and poor docking scores at high guidance scales, exactly where the PEC indicates path collapse, while ACE remains stable and achieves substantially better docking. **Figures 5, E.4** visualize typical outputs: ACE yields consistent, chemically valid molecules, whereas FKC (with the same seed) produces fragmented, invalid structures. This demonstrates that the gains come from preventing collapse, not from extra parameter tuning.
>
> 2. ***“Could you evaluate experimentally the occurrence of the phenomenon in truly realistic synthetic settings, and in the molecular design task without ACE?”***
>
> Yes. The new synthetic experiments use only standard schedules; the molecular analysis compares NR, FKC, and ACE under identical hyperparameters and analyzes when the PEC is violated. In both domains, the presence of PEC violations tightly correlates with degraded performance for NR/FKC and with the regimes where ACE is necessary. Please refer to the revised **Section 3, Appendix E.2, Tables 3, E.5, and Figures E.4-E.11** for the detailed experimental evidence.

---

### Official Review · Reviewer_gBQy · 2025-11-01

**Soundness:** 2
**Presentation:** 2
**Contribution:** 2
**Rating:** 4
**Confidence:** 3

**Summary:**

This paper identifies the so-called path-collapse phenomenon in generative methods using stochastic interpolation $q_t \sim \alpha_t X_0 + \beta_t X_1$ between a Gaussian $X_0 \sim \mathcal{N}(0,I)$ and a compactly supported distribution $X_1 \sim q_1$.
This phenomenon appears when trying to combine multiple generative models (with different dimensions and interpolation schedules) to sample from a quotient of densities
$h_1 = \prod_{i=1}^M (q_1^{(i)})^{\gamma_i}$ for $\gamma_i \in \mathbb{R}$
of the constituent models due to possible non-integrability of the ratio $h_t = \prod_{i=1}^M (q_t^{(i)})^{\gamma_i}\,,$ of the interpolated densities $q_t^{(i)}$ (even if the endpoints are integrable)
i.e. $h_t \notin L^1$ even if $h_0, h_1 \in L^1$.
The paper provides a necessary and sufficient criterion to identify when $h_t \in L^1$ in terms of $\gamma_i$ and $\alpha_t$.
Furthermore, in case $h_t \notin L^1$, the authors show that one can always, as a remedy, add a time-varying bump function $\tilde{\gamma}_i(t) = \gamma_i + B_i t(1-t)$ for some constants $B_i > 0$ to the exponents such that after this modification $\tilde{h}_t \in L^1$ and the endpoints are preserved: $\tilde{h}_0 = h_0$ and $\tilde{h}_1 = h_1$.
Moreover, it is shown how to modify the Feynman-Kac PDE to sample from $\tilde{h}_1$, while evolving only via the marginal
Finally the authors provide toy and real-world examples in conditional molecule and image generation showing that preventing this path-collapse improves existing results such as No Resampling in CFG.

**Strengths:**

Calling attention to the potential existence of degeneracies of marginal distributions when combining multiple generative models.

**Weaknesses:**

The authors introduce the ratio-of-density paradigm, namely sampling from $p(x)^{\gamma_1}/q(x)^{\gamma_2}$, and propose remedies within this framework.
However, they spend little time in explaining what generative tasks and processes fit into this framework in general, both in the main text and well as the appendix.
Only from the examples in Section 2.5 or 3 can the reader infer some of the concrete use cases, which are, however, difficult to parse for readers without intuition and background in molecule and image generation.
The authors should first provide a list of general problems (such as e.g. CFG and product of experts) and show mathematically (!) how these can be posed as ratio-of-densities steering problems that require (!) the intermediate densities $h_t$ for $t \in [0,1]$ in the corresponding SDE/ODE flow.
Only then it is clear why the problem the authors solve is relevant, which is not the case of for the current version.

Minor details:
* Add reference for Proposition B.2 in the appendix
* Fix Remark in Appendix B.2: $P_0(x) = 1_{[0,1]}(x) \text{\ and\ } Q_0(x) = 2(1-x) 1_{[0,1]}(x)$
are two probability densities with compact support, but $P_0(x)/Q_0(x)$ is not integrable. Also Lemma 4.6 is not defined.
* More clear wording (e.g. unbounded domain instead of infinite domain)
* Proof of Corollary B.1 shows only one direction. Mention this or add other implication as well.
* Proof of Proposition B.5 is unfinished (see last few lines of the proof). Also not clear why one shows a lower bound, since the statement in the proof concerns with upper bound only. Lower bound should be established separately if needed later.
* Duplicate wording in Theorem 2.2
* Unclear symbol S(t) on page 5, did they mean C(t) ?
* Unexplained symbols in Section 2.5 such as $\mathcal{T}(\mathcal{M})$ or $\leftrightarrow$ and equation 2.1 or 2.2 undefined
* In the proof of Theorem B.2 it is stated that $q_t^{(i)}$ is the result of the convolution of a Gaussian and a compactly supported distribution. But then the support of $q_t^{(i)}$ is unbounded, contradicting the definition of $R_t^{(i)}$

**Questions:**

* How exactly are the intermediate densities $h_t$ for $t \in [0,1]$ required for sampling from $h_1$?


Limitations:
It is assumed that the stochastic interpolation is between a Gaussian and compactly supported distribution.

---

> ### Author Response · Authors · 2025-11-27
> **[1/2] Response to Reviewer gBQy**
>
> We thank the reviewer for the careful reading and constructive feedback. We have updated both the main manuscript and the supplementary materials, and we believe the revised version addresses all identified concerns. All changes made in response to reviewer comments are highlighted in blue, and the corresponding reviewer IDs are noted alongside each revision for clarity.
>
> ---
>
> 1. ***“The authors spend little time in explaining what generative tasks and processes fit into this framework in general … The authors should first provide a list of general problems (such as CFG and product of experts) and show mathematically (!) how these can be posed as ratio-of-densities steering problems …”***
>
>
> **Author Actions:** We added a dedicated discussion of common steering methods and their ratio-of-densities form in the **Introduction** and in a new subsection **Appendix D.1**. This now includes explicit examples and a mathematical demonstration that CFG, product-of-experts conditioning, contrastive decoding, reward-tilting, and Bayesian conditioning with pretrained experts all reduce to the ratio-of-densities path, which requires validity of intermediate densities.
>
>
> **Details.** Many standard inference-time control techniques can be unified under the ratio-of-densities framework, where the target density takes the form $p^\star(x) \propto \prod_i q^{(i)}(x)^{\gamma_i}$. Along the generative trajectory, this induces a time-indexed family $p_t^\star(x)\propto\prod_i q_t^{(i)}(x)^{\gamma_i}$, whose intermediate densities must remain normalizable for valid sampling.
>
> - **Classifier-free guidance (CFG)**: To enhance samples consistent with condition $y$, CFG reweights the conditional model relative to the unconditional one, $p^\star(x) \propto p_\theta(x\mid y)^{\gamma}\,p_\theta(x)^{1-\gamma}$, implying the path $p_t^\star(x)\propto q_t(x\mid y)^{\gamma}\,q_t(x)^{1-\gamma}$ with score $\gamma\nabla\log q_t(x\mid y)+(1-\gamma)\nabla\log q_t(x)$.
>
> - **Product-of-experts:** To enforce multiple constraints $c_1,\dots,c_K$, one multiplies separate experts $q^{(k)}(x)=p_\theta(x\mid c_k)$, yielding $p^\star(x)\propto \prod_{k=1}^K q^{(k)}(x)$ and the path $p_t^\star(x)\propto \prod_{k=1}^K q_t^{(k)}(x)$.
>
> - **Reward-tilted / RL-style guidance:** Combining a base model $p_\theta(x)$ with a reward $r(x)$ creates a target $p^\star(x)\propto p_\theta(x)\exp(\beta r(x))$. This can be viewed as a product of base density $q_\text{base}(x)$ with a reward-density $q_{\text{rew}}(x)\propto\exp(\beta r(x))$, leading to the path $p_t^\star(x)\propto q_t^{\text{base}}(x)\,q_t^{\text{rew}}(x)$.
>
> - **Contrastive / reference model decoding:** To suppress generic patterns from a reference model $p_{\mathrm{ref}}$, the target is defined as $p^\star(x)\propto p_{\mathrm{LM}}(x)^{\gamma}\,p_{\mathrm{ref}}(x)^{-\gamma}$, inducing the path $p_t^\star(x)\propto q_t^{\mathrm{LM}}(x)^{\gamma}\,q_t^{\mathrm{ref}}(x)^{-\gamma}$.
>
> - **Bayesian composition:** When only conditionals $p_\theta(x\mid c_k)$ are available, Bayes yields $$p(x\mid c_{1:K})\propto {p(x)}\prod_k p(c_k\mid x)\propto \underset{\text{\tiny prior}}{p(x)}\prod_k p_\theta(x\mid c_k)/\underset{\text{\tiny marginal}}{p_\theta(x)}$$The assumptions and derivation under the heterogeneous conditioning structure is given in **Eq. 11**. This induces the path $p_t^\star(x)\propto q_t^{\mathrm{prior}}(x)\,\prod_k q_t^{(k)}(x)\,q_t^{\mathrm{marg}}(x)^{-1}$ which fits our general ratio-of-densities template.
>
> - **Scientific applications.** Our molecular compositions for scaffold decoration, protein glue generation, fragment linking are the examples of Bayesian composition; the full derivations are given in **Section E.1**.
>
> ---
>
> 2. ***“How exactly are the intermediate densities $h_t$ for $t\in[0,1]$ required for sampling from $h_1$?”***
>
> **Author Actions:** We added a paragraph in the **Introduction, Sec. 2.1, and Appendix A.2** to make this requirement explicit.
>
> **Details.** In our notation (**Sec. 2.1**), the intended ratio-of-densities path is defined by
> $$
> p_t^\star(x) = \frac{h_t(x)}{Z_t}, \quad h_t(x) = \prod_{i} \tilde q_t^{(i)}(x)^{\gamma_i(t)}, \quad Z_t = \int h_t(x)\,dx.
> $$
>
> SDE/ODE samplers and FKC do not only depend on the endpoint density $p_1^\star$: they are defined to follow an entire time-indexed family of target densities $\\{p_t^\star\\}_{t\in[0,1]}$ whose scores are used in the drift and weight updates. These intermediate densities are obtained (up to normalization) from the ratio-of-densities integrand $h_t$. If $h_t$ fails to be integrable at any intermediate time, the path $\{p_t^\star\}$ ceases to exist; the sampler still produces some probability path $\{p_t'\}$, but there is no longer a guarantee that $p_t' = p_t^\star$ or that $p_1'$ equals the desired target $p_1^\star$.
>
> (continued)

---

> ### Author Response · Authors · 2025-11-27
> **[2/2] Response to Reviewer gBQy**
>
> The derivation of the SDE/ODE, and Feynman–Kac corrector for sampling from the target distribution (**App. A.2**) assumes that each $Z_t$ is finite, so that $p_t^\star$ is a normalized density and solves the associated Fokker–Planck equation. Under these assumptions, the law of the sampler, $\{p_t'\}$, is equal to $\{p_t^\star\}$ and the terminal law $p_1'$ coincides with the desired ratio-of-densities target $p_1^\star$.
>
> However, when Marginal Path Collapse occurs (i.e., there exists $t^\star$ such that $h_{t^\star}\notin L^1$ and $Z_{t^\star}=\infty$) no normalized density $p_{t^\star}^\star$ exists. The mixed field $s_t^\star = \sum_i \gamma_i(t)\,\tilde s_t^{(i)}$ is still a well-defined vector field, so the SDE/ODE and FKC updates are *numerically computable*, but there is no density whose log-gradient this field represents at $t^\star$. In this regime, the sampler still induces a probability path $\{p_t'\}$ via its own Fokker–Planck equation, yet that path is no longer constrained to match any ratio-of-densities path $\{p_t^\star\}$. Empirically (**Tab. 2,3,4, and Fig.4**), we observe that precisely when our path-existence criterion (**Theorem 2.1**) detects $h_t\notin L^1$, NR and FKC become unstable and fail to approximate the target distribution, whereas ACE modifies the exponents (**Theorem 2.2, 2.3**) to ensure $Z_t<\infty$ for all $t$ and thus *keeps the sampler aligned with the intended ratio-of-densities path*.
>
> ---
>
> 3. ***Minor issues (Proposition B.2 reference, Remark in B.2, Lemma 4.6, wording, Corollary B.1 direction, Proposition B.5, duplicate wording, symbol $S(t)$, unexplained symbols, support conflict in original submission).***
>
>
> **Author Actions:** We carefully revised the main text and the appendix to fix all of these issues: we corrected references, clarified notation, removed nonessential or incorrect statements, and streamlined the proofs.
> - Regarding the integrability condition for product of GMMs, we unified partial results (previously Proposition B.2, Corollary B.1) into a single theorem (**Theorem B.4** in the revised version).
> - We removed the Remark in Appendix B.2 (previously discussing a counter example for the compactly supported target case) where the assumption of the integrability at the endpoints ($P_0/Q_0$) was missing and Lemma 4.6 was not defined. We decided that the synthetic checker experiments and real-world evaluations are better examples that show path collapse.
> - For Proposition B.5 (**Proposition B.2** in the revised appendix), the aim was to prove exponential upper and lower bounds for $p_t$, which we make explicit in the current version.
> - The duplicated wording in Theorem 2.2 and the symbol $S(t)$ (which was intended to be $C(t)$) have been fixed.
> - Symbols such as $\mathcal{T}(\mathcal{M})$ and $\leftrightarrow$ are now defined near **Eq. 9** in the main text.
> - The definition of $R_t$ is intended to denote radius of the support of the *scaled* compactly supported density $\beta_tX_1$ prior to convolution. This interpretation is clarified in **Proposition B.2** of the revised manuscript. Moreover, the proof of Theorem B.2 (**Theorem B.1** in the revision) now explicitly relies on the revised **Proposition B.2**, thereby resolving the apparent contradiction concerning unbounded support following Gaussian convolution.
>
> 4. ***“Limitations: It is assumed that the stochastic interpolation is between a Gaussian and a compactly supported distribution.”***
>
> We acknowledge that our analysis focuses on interpolants between a Gaussian prior and a compactly supported data distribution, following the standard diffusion setup. As discussed in **Section F**, extending PEC and ACE to more general interpolants to support emerging applications (e.g., molecule or image editing) is an important direction for future work. We believe this extension is feasible, as the core structure of ACE (adaptive exponents + Feynman–Kac sampling) does not depend on the specific choice of interpolant.

---

### Author Response · Authors · 2025-11-27
**Summary of Revisions**

Dear Reviewers,

We sincerely thank you for your detailed, insightful, and constructive feedback. We have uploaded a revised version of the paper; all changes are highlighted in blue and annotated with reviewer IDs in both the main text and the appendix for ease of navigation. While we respond to each comment individually below, we would like to briefly summarize the revisions.


### **Revisions in the main paper**

* Expanded **Introduction** to explain how common steering tasks reduce to ratio-of-densities construction, why heterogeneous experts and schedules arise in practice, and why valid sampling (SDE/ODE/FKC) requires normalized intermediate densities along the entire generative path.
* Added clear motivation for the role and prevalence of **heterogeneous noise schedules** (supported by our task-specific scheduler survey).
* Reorganized **Theorem 2.3** for clarity and readability.
* Provided clearer descriptions of the **synthetic** and **scaffold-decoration** experiments, explicitly defining each expert and the conditioning structure.
* Added new empirical results on **collapse prevalence**, **collapse duration**, and **bump parameter selection**, strengthening the practical relevance of Marginal Path Collapse.

### **Additions in the Appendix** (Subsections)

* **A.2:** Complete pseudo-code for ACE and FKC with a clear comparison.
* **D.1:** Detailed derivations showing how CFG, PoE, contrastive decoding, reward-tilting, and Bayesian composition all instantiate ratio-of-densities paths.
* **E.1:** Formal treatment of heterogeneous conditioning structures and their scientific applications.
* **E.2:** Systematic analysis of collapse frequency and empirical consequences.
* **E.3:** Survey and analysis of task-specific and adaptive noise schedulers.
* **E.4:** Results demonstrating that ACE also improves performance in *non-collapsing, homogeneous* image settings.

### **Core contributions.**
1. **Path Existence Criterion (Theorem 2.1):** a necessary and sufficient condition that characterizes when the intended ratio-of-densities path exists and predicts Marginal Path Collapse.
2. **Bump-function Correction (Theorem 2.2):** a simple, constructive modification of the exponents that restores path existence while preserving both endpoint marginals.
3. **Adaptive Path Correction with Exponents (Theorem 2.3):** a generalized, adaptive extension of Feynman–Kac steering dynamics that supports time-varying exponent schedules.

Notably, this adaptive correction (3) provides benefits beyond collapse prevention and supports more flexible, time-varying steering schedules.

We believe these revisions substantially strengthen the paper in terms of clarity, scope, and empirical support. We are grateful for your thoughtful feedback and are happy to clarify any remaining questions.


Sincerely,

The Authors

---

### Author Response · Authors · 2025-12-02
**Summary of Rebuttal Outcomes: Updated Reviewer Scores & Key Revisions**

Dear Area Chair,

We understand the challenges created by the recent OpenReview security incident and score rollback, and we appreciate your effort in evaluating our submission under these circumstances. We would like to provide a concise summary of the post-review discussion phase to assist your assessment.

---

### 1. Integrity and Process

We confirm that we have strictly adhered to the ICLR Code of Conduct. We have had **no contact with any reviewer outside of OpenReview**, and all changes in reviewer stance reflected on the forum were made voluntarily by the reviewers, based solely on our revisions and technical responses.

---

### 2. Reviewer Positions After Discussion (Pre-Reversion)

Although the numeric scores have been reverted to their pre-discussion values, the **official reviewer comments** on OpenReview reflect substantially improved evaluations. The table below summarizes the reviewer stances *just before* the system freeze, based on their official comments:

|Reviewer ID|Original Score|**Updated Stance (In Comments)**|Status|
|-|-|-|-|
|**naNy**|2|**8 (Raised to Accept)**|Wrote an official follow-up stating that all previous questions have been *fully addressed*, that the *logical clarity has substantially improved*, and that they are *“pleased to raise [the] score to 8”* (explicitly noting that they cannot change the score due to system restrictions)|
|**jeuE**|8|**8 (Accept) + Confidence 2->3**|Posted an official comment confirming that all concerns have been addressed, that they *maintain their positive evaluation*, and that they *raise their confidence score to 3*.|
|**TtEF**|6|6 (No reply due to freeze)|Initially expressed concern about empirical evidence. In our response and revision, we added the requested realistic experiments and frequency analysis.|
|**gBQy**|4|4 (No reply due to freeze)|Requested a clearer list of ratio-of-densities steering problems, explicit mathematical formulations, and fixes to several appendix issues, all of which we addressed in the revised manuscript (Intro, App. D.1, A.2) and response.|

While the system currently displays the pre-discussion scores, we hope the explicit post-discussion written endorsements by Reviewer naNy and Reviewer jeuE will be the primary context for your evaluation.

---

### 3. Resolution of Key Concerns

Across all four reviews, the main concerns fell into three recurring categories, and for each category at least one reviewer explicitly acknowledged that our revisions fully resolved their earlier concerns (most notably Reviewer naNy):

1. **Background and Clarity of the Framework.**
   Raised by both Reviewer gBQy and Reviewer naNy. We clarified how common steering methods (CFG, PoE, contrastive/reward-tilted sampling, Bayesian composition) instantiate ratio-of-densities paths, defined “experts” precisely, and made explicit the path-existence assumption underlying SDE/ODE/FKC methods (Introduction, Sec. 2.1, Appendix D.1, Appendix A.2).

2. **Empirical Relevance and Consequences of Marginal Path Collapse.**
   Raised by Reviewer TtEF and Reviewer naNy. We added systematic experiments measuring how often PEC is violated under standard schedules, realistic synthetic experiments using only widely used schedules (no adversarial constructions), and molecular studies showing that NR/FKC degrade exactly when PEC is violated, under identical hyperparameters to ACE (Sec. 3, App. C.2, E.2). We also analyzed the impact of collapse duration and included a homogeneous control regime where there is no collapse and ACE reduces to FKC.

3. **Experimental Design and Expert Definitions.**
   Raised primarily by Reviewer naNy and also touched on by other reviewers. We clarified the synthetic checker setup (exact checkerboard distribution, each expert’s role, and the Bayes factorization) and formalized heterogeneous conditioning structures that map directly to scaffold decoration (App. C.1, E.1). We also provided precise definitions of the DN/CONF/SBDD experts and added all missing citations and scheduler details (Sec. 2.5, App. E.2–E.4).

Additionally, **Appendix E.4** shows that ACE’s adaptive exponents are beneficial even in homogeneous, non-collapsing image-generation settings, highlighting that our generalized formulation (Theorem 2.3) has value beyond collapse prevention and supports more flexible steering strategies.

Reviewer naNy’s follow-up comment states that all previous questions on these key concerns have been fully addressed and that the logical clarity has substantially improved.

---

We hope this summary, together with the full revised manuscript and the reviewers’ follow-up comments, provides a clear picture of how the prior concerns have been fully addressed. Thank you again for your time and effort in handling this submission.

Sincerely,

The Authors

---

### Note · Program_Chairs · 2026-01-17
**Submission Desk Rejected by Program Chairs**

The following references in this submission do not refer to real documents and/or have major errors in bibliographic information:

 Jocelyn Sunseri and David R. Koes. Pharmacophore-constrained fragment linking. Journal of Chemical Information and Modeling, 60(3):1184-1193, 2020. doi: 10.1021/acs.jcim.9b00966.